# BEAR: Dissecting Embodied Abilities in Multimodal Language Models through Skill-level Evaluation and Diagnosis

Yu Qi [* 1]  Haibo Zhao [* 1]  Ziyu Guo [* 2]  Siyuan Ma [3 2]  Ziyan Chen [1]  Yaokun Han [2]  Renrui Zhang [2]
Zitiantao Lin [1]  Yizhe Zhu [1]  Shiji Xin [4]  Yijian Huang [1]  Boce Hu [1]  Kai Cheng [5]  Jiayi Zhang [1]  Peiheng Wang [6]
Jiazheng Liu [6]  Wenqing Wang [1]  Yiran Qin [7]  Haojie Huang [1]  Lawson L.S. Wong [1]

## Abstract

Understanding the capability bottlenecks of embodied multimodal large language models (MLLMs) is crucial for improvement. However, existing embodied benchmarks fail to provide actionable insights because they focus on task-level evaluation rather than discovering capability bottlenecks. To address this, we introduce BEAR, where we divide embodied tasks into 14 atomic skills for **skill-level evaluation**. BEAR comprises 4,469 interleaved image–video–text entries across 14 skills in 6 categories, ranging from low-level perception to high-level planning. We evaluate 20 MLLMs on BEAR under a **hierarchical skill-level diagnosis framework** and discover that *(1)* perceptual capabilities are major bottlenecks behind reasoning failures, and *(2)* models fail due to unstable spatiotemporal modeling which remain unexposed in previous benchmarks. Furthermore, building on these insights, we propose BEAR-AGENT, a multimodal conversable agent that augments MLLMs with visual and spatial tools. It substantially enhances MLLMs' performance across skills, yielding a relative improvement of *17.5%* on GPT-5 on BEARand outperform baselines by a large margin in both simulation and real-robot experiments across models. We provide our project website at **https://bear-official66.github.io/**.

---

[*]Equal contribution [1]Northeastern University, Boston, MA, USA [2]The Chinese University of Hong Kong, Hong Kong, China [3]Westlake University, Hangzhou, China [4]Harvard University, Cambridge, MA, USA [5]Purdue University, West Lafayette, IN, USA [6]Peking University, Beijing, China [7]University of Oxford, Oxford, United Kingdom. Correspondence to: Yu Qi <qi.yu2@northeastern.edu>.

*Proceedings of the 43rd International Conference on Machine Learning*, Seoul, South Korea. PMLR 306, 2026. Copyright 2026 by the author(s).

## 1. Introduction

Embodied abilities refer to a collection of perceptual and reasoning capabilities for an agent to interact with the physical world, spanning from low-level perception to high-level planning (Duan et al., 2022). As multimodal language models (MLLMs) are increasingly deployed as embodied agents, task-level performance alone is insufficient: without understanding which capabilities fail and why, progress in embodied intelligence remains largely unguided.

However, existing embodied benchmarks are fundamentally limited for diagnosis. Many of them focus on single domain (Yang et al., 2025a). Others treat each task as an indivisible unit and relying on final binary success (Yang et al., 2025b; Li et al., 2023) and they collapse long-horizon decision-making steps into a single outcome, making it challenging to attribute failures to specific perceptual or reasoning capabilities. Therefore, they rarely provide concrete guidance on how limitations should be addressed.

To overcome the shortcomings, we propose **skill-level evaluation** framework, where each skill is a atomic capability-oriented step for embodied task execution. Together, effective diagnosis of embodied MLLMs must satisfy three requirements: *(1)* skill-level evaluation and analysis beyond task success, *(2)* principled attribution of failures to underlying capabilities, and *(3)* actionable guidance for improving embodied performance.

Motivated by this, as shown in Fig. 2, *(1)* we propose **BEAR**, the first embodied benchmark **explicitly designed for capability diagnosis** rather than task success. In *Long-horizon* category (Fig. 3), BEAR inductively summarize embodied tasks into atomic skills, each skill is a structured perceptual and reasoning step. Beyond that, it enables 14 isolated skill-level evaluation through 4,469 interleaved image-video-text samples across 6 categories, covering capabilities from low-level perception to high-level planning.

Task-level evaluation inherently conflates many capabilities into a single success signal, making it difficult to pinpoint whether failures arise. Crucially, we propose a **hierarchical skill-level diagnosis framework** grounded in our struc-

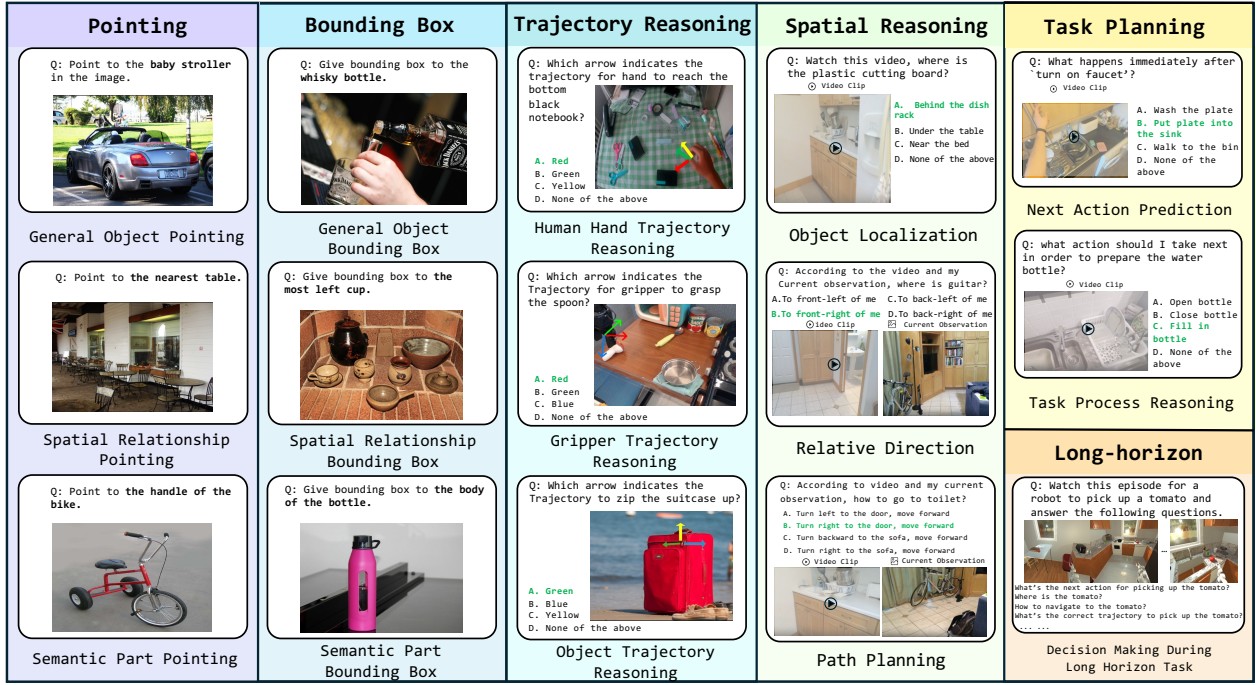

*Figure 1.* **BEAR** is the first diagnostic benchmark for evaluating MLLMs in embodied capabilities. It covers 6 categories and 14 atomic skills, comprising 4,469 interleaved image–video–text VQA samples curated from 13 diverse data sources and tailored to each category.

tured skill definition, enabling both **horizontal and vertical diagnosis** and providing a principled way to uncover fundamental capability bottlenecks. Horizontally, the *Long-horizon* category (Fig. 3) allows us to identify which skills consistently act as major bottlenecks during embodied task execution. Vertically, evaluating each atomic skill in isolation enables failures to be traced to fine-grained underlying capabilities. Beyond these two levels, **cross-skill failure attribution** aggregates failure patterns across skills to reveal which fundamental capabilities repeatedly give rise to errors in diverse contexts, moving beyond task-level symptoms to expose shared bottlenecks.

Extensive experiments and hierarchical diagnosis of 20 representative MLLMs on BEAR reveal bottlenecks that are uncovered by previous benchmarks: *(a)* **perceptual capabilities are major bottlenecks behind reasoning failures**, even for tasks such as planning and spatial reasoning that are more reasoning-oriented. *(b)* **unstable spatiaotemporal modeling**. The performance limitations can not be effectively addressed by Chain-of-thought (CoT) prompting or test time compute scaling. Additional methods should be designed to address limitations.

Motivated by these findings, we propose an actionable improvement: providing additional visual and spatial cues that is not readily accessible from original input to MLLMs. *(3)* To this end, we introduce **BEAR-AGENT**, a multimodal agent which interacts with an MLLM through dialogue and

leverages a set of external tools to supply additional visual and spatial information. Tools are implemented as modular Python functions, including trajectory visualization for motion understanding, calls to foundation models such as GroundingDINO for object grounding, and 3D scene graph construction for spatial relationships among objects. Experiments show that BEAR-AGENT improves GPT-5 (OpenAI, 2025a), the current state-of-the-art model on BEAR, by *9.12%*, corresponding to a relative performance gain of *17.5%*. Furthermore, we deploy the agent in simulation environment on three sets of representative manipulation tasks and deploy it in real-robot grasping. Experiment results show that it boosts performance of over *20.17%*. It demonstrates potential of BEAR-AGENT not only enhances offline evaluation but also online execution of embodied tasks, further highlighting diagnosis value of BEAR.

In summary, our contributions are threefold:

- We introduce **BEAR**, the first systematic embodied diagnostic benchmark with 4,469 interleaved image–video–text samples.

- We propose a hierarchical skill-level diagnosis and cross-skill failure attribution framework. Deploying it on BEAR reveals hidden capabilities bottlenecks and leads to actionable insights for improvement.

- We deploy insights to **BEAR-AGENT**, a multimodal agent that significantly enhances offline and online

evaluation across environments in different MLLMs.

**Conflict of Interest Disclosure.** The authors declare no financial conflicts of interest related to this work.

## 2. Related Work

Visual question answering has been extended to the embodied domain, with many benchmarks emphasizing specific categories such as pointing (Yuan et al., 2024; Zhou et al., 2025), spatial reasoning (Wang et al., 2025; Yang et al., 2025a; Du et al., 2024), and multi-agent collaboration (Kang et al., 2025). Comprehensive benchmarks (Dang et al., 2025; Du et al., 2024) such as EmbodiedBench (Yang et al., 2025b) focus primarily on task-level evaluation, and their analyses remain at the task level, limiting actionable insights into underlying capability bottlenecks. Embodied-Agent-Interface (Li et al., 2024d) decomposes agent decision making into modular functions for performance assessment (goal interpretation, subgoal decomposition), but they are not capability-oriented and it fails to provide capability improvement insights. In contrast, BEAR provides a structured skill-level evaluation and diagnostic framework with cross-skill failure attribution, enabling identification of fundamental bottlenecks and actionable directions, we also validate our diagnosis insights by providing an agent. We provide additional related work discussed in Appendix A.

## 3. The BEAR Benchmark

### 3.1. Overview of BEAR

In Fig. 1, BEAR is the first comprehensive benchmark for embodied capabilities, featuring 4,469 interleaved image-video-text samples. It includes five core categories, further decomposed into 14 fine-grained skills, along with a sixth *long-horizon* category to evaluate their integration in embodied tasks. Detailed statistics and category distributions are in Fig. 2a, 2b, and Appendix C, D.

**Five core categories are inductively summarized from task execution processes of embodied agents and humans.** Our categorization is derived from analyses of large-scale embodied household activity dataset such as BEHAVIOR-1K (Li et al., 2023) and ALFRED (Shridhar et al., 2020), together with insights from human cognitive processes for task execution. Using the activity of rinsing a cup as an example: *(1) Task Planning* involves questions about both past and future actions, including two skills, *Task Process Reasoning* (e.g., recognizing the agent is already picking up the cup) and *Next Action Prediction* (e.g., inferring the next step is to approach the faucet). *(2) Spatial Reasoning* captures the ability to localize objects and navigate within environment. It includes *Object Localization*, *Path Planning*, and *Relative Direction*. For instance, the agent must

locate the faucet relative to other landmarks (e.g., 'to the right of the stove'), plan a path to it (e.g., 'move forward'), and when near the faucet, identify its relative position (e.g., 'front-left'). This is followed by *(3) Bounding Box* for coarse localization by identifying region of the faucet. *(4) Pointing* for precise interaction (e.g., 'the handle of the faucet'), and *(5) Trajectory Reasoning* for motion execution (e.g., 'turn on faucet'). *Pointing* and *Bounding Box* and *Trajectory Reasoning* is further divided into three skills by perceptual concept and embodiment type.

*Long-horizon* category for the first time decomposes embodied tasks into skill-oriented steps. This category features 35 episodes collected from AI2-THOR (Ehsani et al., 2021), each decomposed into structured skill-oriented steps for **offline evaluation**. In Figure 3, an episode with high-level goal 'put the apple in the sink' is broken down into a chain of steps: the agent must first plan its next action, search for the sink's location, chart a path towards it, reason about its relative position, visually perceive the sink, and finally predict the trajectory to place the apple inside. Crucially, each step can be grounded to an atomic skill within BEAR. It indicates that our skill taxonomy is not only motivated by human cognitive processes but also practically applicable to embodied tasks.

### 3.2. Data Curation Process

**Diverse and category-specific data curation.** We curate our data using 13 distinct data sources spanning real-world images, videos, and simulation episodes, then employ category-tailored strategies to generate VQA pairs. For example, we use OpenImages (Kuznetsova et al., 2020) for *Pointing*, Open-X-Embodiment (O'Neill et al., 2024) for *Trajectory Reasoning*. Our multi-stage data generation pipeline combines automated semantic filtering via GPT-o3 (OpenAI, 2025b) with at least three rounds of rigorous **human verification**, conducted by a team of 10 trained annotators. We also apply strict **ethical filtering** to exclude sensitive or ambiguous content. This hybrid curation framework balances scale, accuracy, and ethical integrity. For full details, please see Appendix E.

**Distribution, quality, distractor and difficulty control.** **(1)** We ensure diverse question distribution within each category; for instance, the *Pointing* category spans over 100 image classes covering common indoor and outdoor objects for embodied interaction. **(2)** For multiple-choice questions, BEAR applies careful distractor design. Beyond semantically similar distractors, we add options like 'none of the above' to require MLLMs to thoroughly evaluate all candidates. **(3)** To mitigate response position bias, we balance the distribution of the correct answer key. **(4)** Difficulty levels are calibrated in each category. For example, in *Pointing*, we remove ground-truth masks that are too small or too

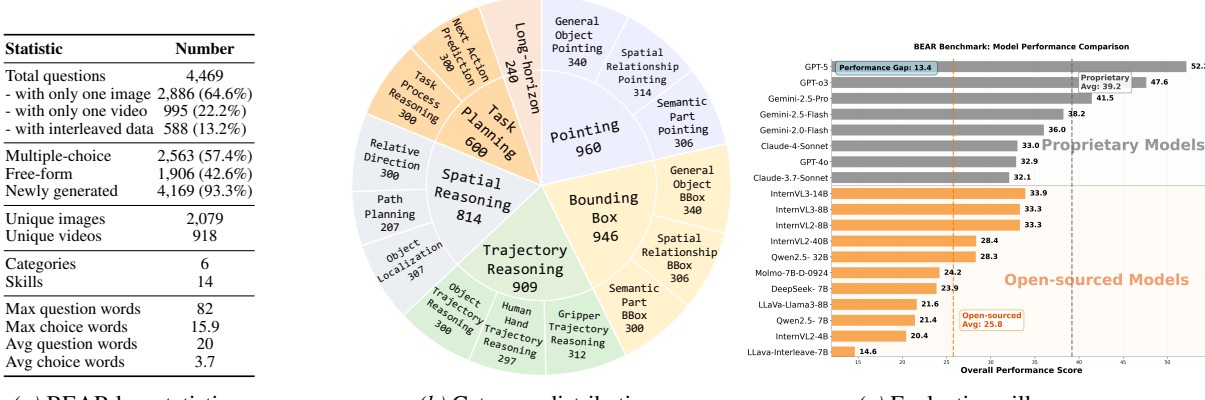

*(a)* BEAR key statistics.    *(b)* Category distribution.    *(c)* Evaluation pillar map.

*Figure 2.* **Statistics, category distribution and evaluation pillar map of the BEAR benchmark.**

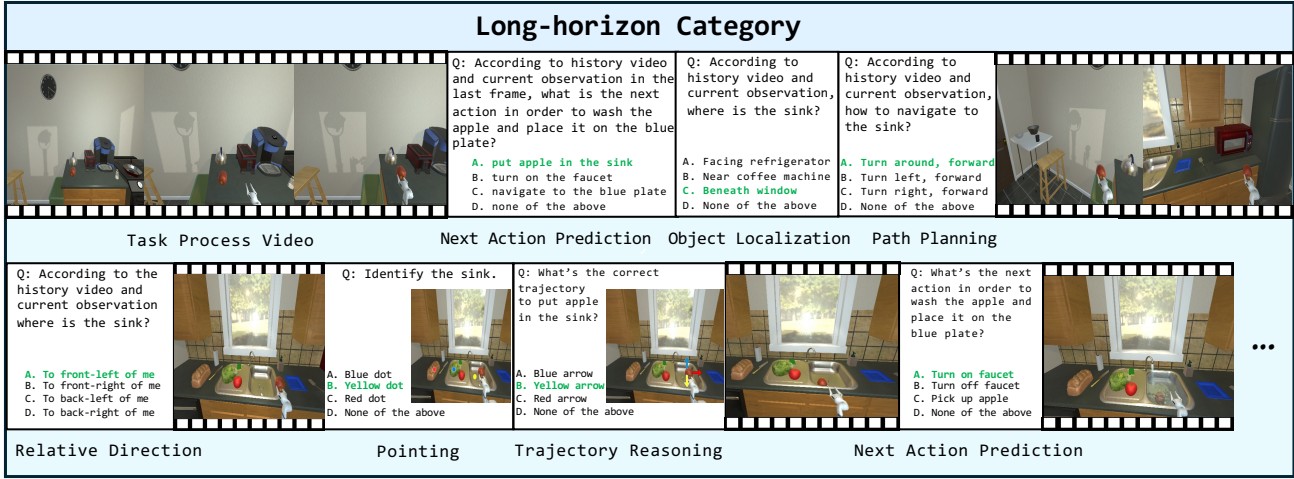

*Figure 3.* **Long-horizon category in BEAR.** The long-horizon category features 35 episodes collected from simulation environment. Each episode is decomposed into skill-oriented steps originate from five core categories and 14 skills in BEAR, ranging from perception to planning. Details in Appendix C.6.

large, and uniformly sample by both mask area and object category. **(5)** Only validation and test sets are used for data curation to reduce data contamination. Due to space limits, we refer readers to Appendix F for further details.

## 4. Experiment

### 4.1. Experiment Setup

**Models.** We evaluate 20 representative MLLMs on BEAR, with results shown in Table 1 and Figure 2c. We follow VLMEvalKit's (Duan et al., 2024) default evaluation protocol and provide our prompts in Appendix I.0.2. Depending on model design, inputs are processed either in a *Merged* setting (multiple frames combined) or a *Sequential* setting (frames processed individually). To establish a reference baseline, we report **human performance** of 5 human volunteers on *BEAR-mini*, a subset containing 40 samples per skill. Additional details are in Appendix G.

**Evaluation Metrics.** For *Pointing*, *Spatial Reasoning*, *Task Planning*, *Long-horizon*, we use success rate as evaluation metric. For *Long-horizon*, we report success rate over episodes, an episode is considered successful only if all steps are answered correctly. For *Bounding Box*, we report the average Intersection over Union (IoU).

### 4.2. Results and Analysis

**MLLMs exhibit limited embodied capabilities.** As shown in Table 1, most models achieve only 20–40% overall performance. Even GPT-5 (OpenAI, 2025a), the strongest model, reaches just 52%, far below human performance (89.40%). This gap persists across all skills. In Fig. 4, we evaluate **Chain-of-Thought (CoT)** prompting across all models and find inconsistent gains, with most improvements negligible and typically under 10%. Test-time compute scaling shows similarly limited effects, we refer readers to

Appendix G.0.4 and G.0.3 for more details.

**Proprietary models outperform open-sourced ones.**
Fig. 2(c) shows that proprietary models average 39.2%, outperforming open-source models by 13.4%. GPT-5 leads at 52.2%, exceeding the best open-source model InternVL-3 by 18.3%. Nevertheless, recent open-source models begin to surpass GPT and Claude variants, indicating their growing potential for embodied agents.

## 5. Failure Analysis

We propose a hierarchical skill-level diagnosis framework to uncover capability bottlenecks. The framework composes of both horizontal diagnosis and vertical diagnosis. In the end, cross-skill failure attribution aggregates failure patterns to identify shared limitations. We illustrate them below.

### 5.1. Horizontal Diagnosis: Bottlenecks in *Long-horizon Tasks*

In Fig. 5(d), we analyze failure distribution of GPT-4o during *long-horizon* task execution. Results indicate improving perceptual (e.g., *Pointing* and *Bounding Box*) and spatial reasoning skills, which account for 88% of failures, would yield greatest gains for embodied task performance.

### 5.2. Vertical Diagnosis: Fine-grained Failure Analysis

Vertically, we conduct fine-grained failure diagnosis for GPT-4o (Hurst et al., 2024) seperately in all 14 atomic skills. This analysis provides so far the most comprehensive roadmap of skill-level failure patterns in embodied MLLMs. We provide 14 detailed roadmaps in Appendix H, and summarize *Spatial Reasoning* and *Task Planning* failure patterns in Fig. 5(a)(b) in the next page.

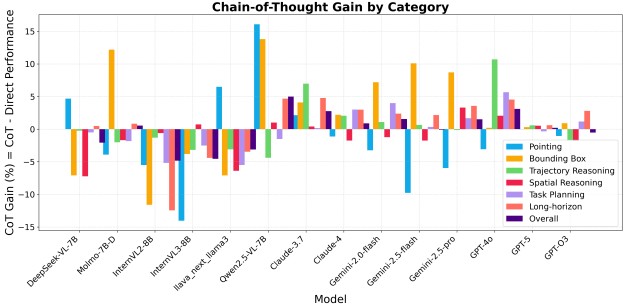

*Figure 4.* **Chain-of-thought performance versus direct prompting across models.** The effect of CoT varies across models and skills; overall, it yields mixed and often limited gains (less than 10%), and can even lead to performance degradation in some cases. We refer readers to Appendix G.0.3 for detailed results.

### 5.3. Cross-skill Capability Attribution

Aggregating failure patterns across skills reveals several consistent trends, providing actionable insights for future improvements. We highlight two of them below.

*(1)* **Perceptual limitations constitute major bottlenecks behind many reasoning failures.** In Fig. 5(c), based on the skill-level failure roadmaps, we categorize each error into perceptual, reasoning, or mixed types and compute their respective proportions. Statistics show that **majority of limitations happen in perceptual (54.8%) instead of reasoning (22.4%)**, moreover, in our detailed analysis, we find in reasoning-oriented skills, models often make incorrect decisions not because their reasoning logic is flawed, but because the visual information they rely on is inaccurately perceived or insufficiently grounded, as shown in Figure 5(a)(b), perception error make bigger portion instead of reasoning error in both categories.

In Fig. 6 in the next page, we present representative model responses for illustration. In *Pointing* and *Bounding Box*, errors are largely caused by localization failures, where models fail to correctly ground the target object. In *Trajectory Reasoning*, errors often stem from perceptual confusion, such as misinterpreting arrow colors, which subsequently leads to incorrect direction reasoning. In reasoning-related tasks like *Spatial Reasoning*, misidentifying the reference object results in incorrect 3D layout estimation and thus wrong inference of spatial relations (e.g., the model confuses the target journal with another poster, leading to an incorrect spatial prediction). Even in *Task Planning*, the dominant failure mode is action understanding errors, where the model observes motion but fails to correctly interpret the action. In Fig. 6(IV), we visualize the attention map of InternVL3-8B (Zhu et al., 2025) and find that, in frame 3, the model attends to the stirring motion but fails to recognize the correct action, which leads to incorrect action prediction.

Together, these results indicate that many high-level reasoning failures originate from upstream perceptual errors, where inaccurate grounding propagates through the reasoning process. Therefore, improving perceptual grounding can holistically enhance embodied capabilities across both perceptual and reasoning skills.

*(2)* **Another recurring cross-skill bottleneck lies in unstable spatial-temporal modeling.** This limitation consistently appears in skills requiring spatial grounding, where models struggle to maintain coherent object-centric or scene-centric coordinate frame. It also manifests in planning tasks, where models frequently omit prior steps or historical context (32.0%). More specifically, in perception-oriented skills such as *Pointing*, models often fail to determine relative spatial relations (e.g., identifying which teapot is closer to the

*Table 1.* **Evaluation results on BEAR.** We report performance of 15 MLLMs here due to space limits. We refer readers to Appendix G.0.2 for full results. GEN = *General Object (Pointing/Box)*; SPA = *Spatial Object (Pointing/Box)*; PRT = *Semantic Part (Pointing/Box)*; PRG = *Task Process Reasoning*; PRD = *Next Action Prediction*; GPR = *Gripper Trajectory Reasoning*; HND = *Human Hand Trajectory Reasoning*; OBJ = *Object Trajectory Reasoning*; LOC = *Object Localization*; PTH = *Path Planning*; DIR = *Relative Direction*. *BBox* scores are **scaled by 100** when computing overall average. We highlight highest scores among proprietary and open-source models.

| | Format | Pointing | | | Bounding Box | | | Task Planning | |
|---|---|---|---|---|---|---|---|---|---|
| | | GEN | SPA | PRT | GEN | SPA | PRT | PRG | PRD |
| Random Choice | | - | - | - | - | - | - | 25 | 25 |
| Human | | 95.50 | 92.00 | 93.50 | 0.830 | 0.770 | 0.820 | 87.50 | 92.00 |
| *Open-source Models* | | | | | | | | | |
| DeepSeek-VL-7B (Lu et al., 2024) | merged | 14.12 | 8.50 | 9.24 | 0.276 | 0.160 | 0.231 | 37.67 | 27.33 |
| Molmo-7B-D-0924 (Deitke et al., 2025) | merged | 23.53 | 19.28 | 25.48 | 0.109 | 0.082 | 0.109 | 37.67 | 31.00 |
| InternVL2-26B (Chen et al., 2024) | merged | 21.18 | 15.36 | 18.79 | 0.201 | 0.202 | 0.147 | 41.33 | 34.33 |
| InternVL2-40B (Chen et al., 2024) | merged | 23.24 | 21.24 | 22.29 | 0.329 | 0.269 | 0.268 | 40.00 | 33.67 |
| InternVL3-8B (Zhu et al., 2025) | merged | 52.65 | 42.48 | 43.95 | 0.369 | 0.275 | 0.297 | 43.00 | 33.67 |
| InternVL3-14B (Zhu et al., 2025) | merged | 37.94 | 27.78 | 32.80 | 0.304 | 0.258 | 0.276 | 41.00 | 33.00 |
| LLava-NeXT-Interleave-7B (Li et al., 2024b) | merged | 6.47 | 3.59 | 2.55 | 0.000 | 0.000 | 0.000 | 37.33 | 26.00 |
| LLaVa-NeXT-Llama3-8B (Li et al., 2024a) | merged | 2.94 | 1.31 | 0.96 | 0.320 | 0.246 | 0.205 | 36.67 | 29.67 |
| Qwen2.5-VL-7B-Instruct (Bai et al., 2025) | merged | 6.18 | 1.63 | 0.96 | 0.007 | 0.003 | 0.009 | 40.67 | 32.33 |
| Qwen2.5-VL-32B-Instruct (Bai et al., 2025) | merged | 27.35 | 27.78 | 42.68 | 0.020 | 0.018 | 0.017 | 42.67 | 42.33 |
| *Proprietary Models* | | | | | | | | | |
| Claude-4-Sonnet (Anthropic, 2025) | sequential | 39.12 | 40.86 | 45.54 | 0.221 | 0.173 | 0.197 | 44.00 | 37.67 |
| Gemini-2.5-Flash (Comanici et al., 2025) | sequential | 46.76 | 33.33 | 39.49 | 0.183 | 0.145 | 0.156 | 48.33 | 43.67 |
| Gemini-2.5-Pro (Comanici et al., 2025) | sequential | 55.00 | 42.48 | 55.41 | 0.144 | 0.103 | 0.177 | 52.00 | 49.00 |
| GPT-4o (Hurst et al., 2024) | sequential | 40.59 | 27.12 | 34.39 | 0.227 | 0.118 | 0.202 | 43.67 | 46.00 |
| GPT-5 (OpenAI, 2025a) | sequential | 70.00 | 63.69 | 54.90 | 0.411 | 0.326 | 0.352 | 59.67 | 61.00 |

| | Format | Trajectory Reasoning | | | Spatial Reasoning | | | Long-horizon | Avg |
|---|---|---|---|---|---|---|---|---|---|
| | | GPR | HND | OBJ | LOC | PTH | DIR | | |
| Random Choice | | 25 | 25 | 25 | 25 | 28 | 25 | 25 | - |
| Human | | 96.50 | 94.00 | 89.00 | 94.50 | 83.50 | 88.50 | 92.50 | 89.40 |
| *Open-source Models* | | | | | | | | | |
| DeepSeek-VL-7B (Lu et al., 2024) | merged | 41.03 | 38.72 | 22.67 | 42.02 | 37.68 | 32.00 | 20.00 | 23.89 |
| Molmo-7B-D-0924 (Deitke et al., 2025) | merged | 45.51 | 41.41 | 23.33 | 49.84 | 29.47 | 26.00 | 5.71 | 24.22 |
| InternVL2-26B (Chen et al., 2024) | merged | 53.21 | 43.77 | 30.33 | 26.06 | 26.57 | 22.00 | 11.29 | 25.66 |
| InternVL2-40B (Chen et al., 2024) | merged | 57.69 | 41.75 | 28.00 | 40.39 | 29.47 | 18.67 | 11.43 | 28.38 |
| InternVL3-8B (Zhu et al., 2025) | merged | 51.28 | 46.80 | 27.67 | 50.16 | 32.37 | 20.00 | 8.57 | 33.32 |
| InternVL3-14B (Zhu et al., 2025) | merged | 51.28 | 49.49 | 31.43 | 43.00 | 28.02 | 21.33 | 28.57 | 33.93 |
| LLaVa-NeXT-Interleave-7B (Li et al., 2024b) | merged | 37.18 | 37.04 | 20.67 | 37.79 | 27.54 | 19.67 | 5.71 | 14.64 |
| LLaVa-NeXT-Llama3-8B (Li et al., 2024a) | merged | 39.42 | 37.71 | 23.00 | 40.39 | 33.82 | 24.00 | 14.29 | 21.65 |
| Qwen2.5-VL-7B-Instruct (Bai et al., 2025) | merged | 54.49 | 48.15 | 30.00 | 38.44 | 31.40 | 21.00 | 22.86 | 21.44 |
| Qwen2.5-VL-32B-Instruct (Bai et al., 2025) | merged | 55.45 | 52.19 | 26.67 | 47.23 | 26.57 | 22.67 | 20.00 | 28.33 |
| *Proprietary Models* | | | | | | | | | |
| Claude-4-Sonnet (Anthropic, 2025) | sequential | 50.00 | 49.16 | 38.00 | 46.25 | 42.51 | 39.67 | 17.14 | 33.05 |
| Gemini-2.5-Flash (Comanici et al., 2025) | sequential | 64.42 | 63.97 | 45.00 | 61.24 | 43.00 | 44.67 | 31.43 | 38.24 |
| Gemini-2.5-Pro (Comanici et al., 2025) | sequential | 66.67 | 65.99 | 48.33 | 64.50 | 40.10 | 44.00 | 31.43 | 41.46 |
| GPT-4o (Hurst et al., 2024) | sequential | 41.99 | 35.35 | 30.67 | 60.91 | 33.33 | 31.00 | 31.43 | 32.90 |
| GPT-5 (OpenAI, 2025a) | sequential | 66.99 | 67.34 | 49.67 | 72.31 | 50.24 | 47.00 | 40.00 | 52.17 |

agent). In *Spatial Reasoning*, the absence of a stable reference frame leads to incorrect left–right judgments and distorted directional understanding, which further propagates into reasoning failures (30.3%). In *Task Planning*, this instability prevents models from reliably tracking action progress over time, leading to omitted steps. Actionable improvements includes precise spatial-temporal modeling using visual foundation models and other tools. We will futher illustrate them in the next section.

# 6. Diagnosis insights lead to BEAR-AGENT

In the previous section, we discover two major capability bottlenecks in MLLMs: *(1)* perceptual limitations and *(2)* unstable spatio-temporal modeling. Prior studies suggest that tool use (Hu et al., 2024) and visual prompting (Gupta & Kembhavi, 2023) can effectively improve the visual reasoning process of large models. Motivated by these works, we design BEAR-AGENT as a tool-augmented framework that provides both 2D structured perceptual grounding and 3D explicit world modeling tools to holistically improve MLLMs' embodied abilities.

## 6.1. Method

BEAR-AGENTis a multimodal conversable agent that provides tool support for perceptual grounding and explicit world modeling. As illustrated in Fig. 7 on page 8, the agent begins by initializing a conversation that guides MLLMs toward structured reasoning about the final answer. The

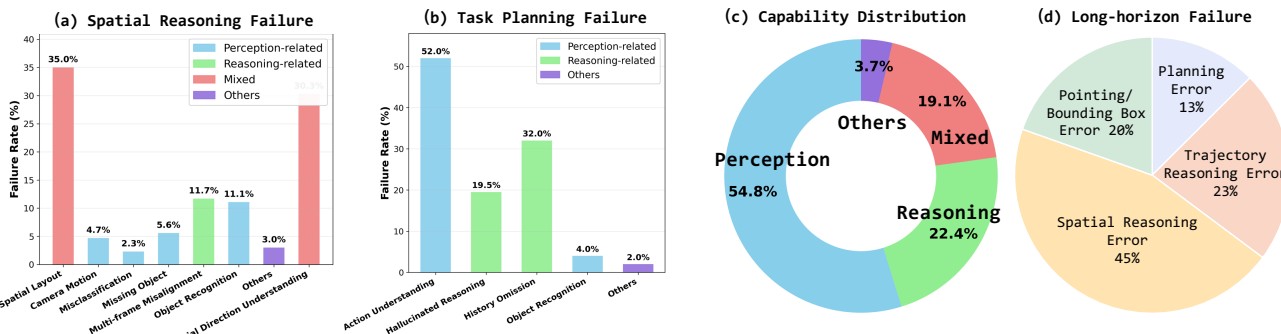

*Figure 5.* **Diganosis statistics.** We summarize failure patterns in spatial reasoning (a), task planning (b), and long-horizon tasks (d). We further quantify capability-specific error distributions in (c), showing perceptual errors form primary bottlenecks in BEAR.

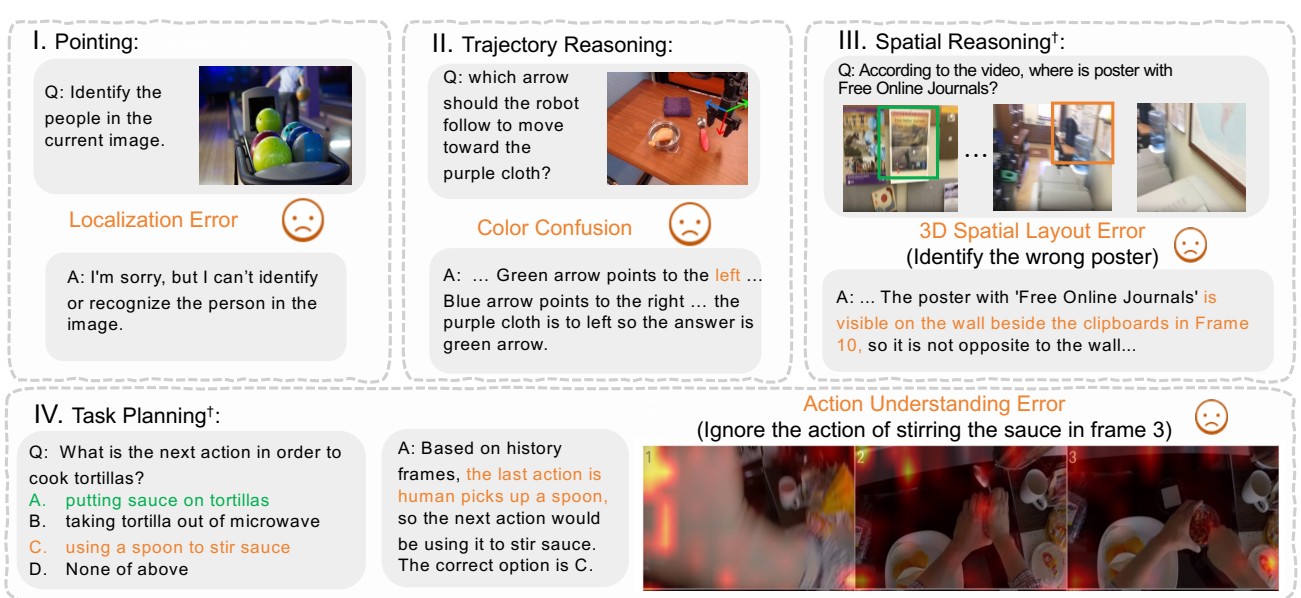

*Figure 6.* **Perceptual capability bottlenecks across skills.** We present representative model responses as failure cases. In (III) and (IV), perceptual errors propagate into reasoning traces, causing incorrect reasoning and answer. (III): Bounding box is only for visualizing wrong answer. (IV): Attention map and answer is visualized from InternVL3-8B.

initial prompts equip MLLMs with a set of tools wrapped as Python functions. *(1)*These tools include foundation vision models that provide explicit visual cues for object grounding, such as GroundingDINO (Liu et al., 2024b), Set-of-Mask (Yang et al., 2023a), and DepthAnything (Yang et al., 2024). Additional tools support trajectory extension and incorporate world knowledge about motion (e.g., the correct rotation direction of a handle). *(2)*For spatiotemporal world modeling, the agent provides functions to update a semantic scene graph, where nodes represent objects and edges encode their relations, along with a notebook module for recording temporal events to support long-horizon planning. After receiving the initial prompt, the MLLM can generate code to invoke these tools; the agent executes the code and returns the results. Once the model completes its reasoning and produces the answer, it signals the agent to terminate the conversation.

## 6.2. Offline evaluation on BEAR

**Experiment setup.** To evaluate the effectiveness of BEAR-AGENT, we conduct experiments shown in Figure 8(a). We use agent on both the best-performing proprietary and open-source model on BEAR: GPT-5 (OpenAI, 2025a) and InternVL3-14B (Zhu et al., 2025). For fair comparison, we establish three baselines: *One-shot*, *Few-shot*, and *Chain-of-thought*. Specifically, **One-shot** provides a single ground-truth question–answer pair as context before each question. **Few-shot** extends this with three question–answer pairs.

**Result analysis.** As shown in Figure 8(a), BEAR-Agent improves performance on BEAR for both GPT-5 and InternVL3-14B. In particular, it yields an average gain of **9.12%** for GPT-5, corresponding to a relative improvement of **17.5%**. Furthermore, BEAR-Agent enhances overall per-

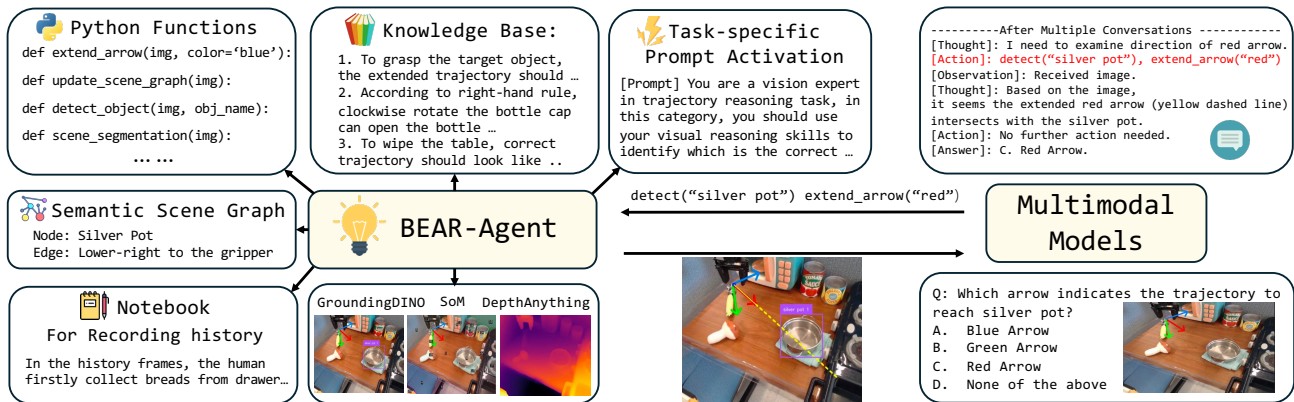

Figure 7. **BEAR-Agent.** BEAR-Agent is a multi-modal conversable agent that interacts with MLLMs through dialogues. It is equipped with category-specific knowledge base, necessary python functions as tools to enhance MLLMs' embodied reasoning abilities.

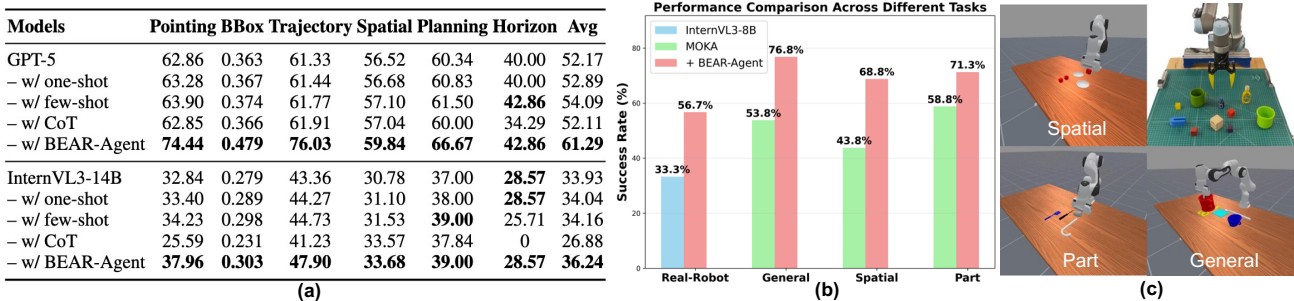

| Models | Pointing | BBox | Trajectory | Spatial | Planning | Horizon | Avg |
|---|---|---|---|---|---|---|---|
| GPT-5 | 62.86 | 0.363 | 61.33 | 56.52 | 60.34 | 40.00 | 52.17 |
| – w/ one-shot | 63.28 | 0.367 | 61.44 | 56.68 | 60.83 | 40.00 | 52.89 |
| – w/ few-shot | 63.90 | 0.374 | 61.77 | 57.10 | 61.50 | **42.86** | 54.09 |
| – w/ CoT | 62.85 | 0.366 | 61.91 | 57.04 | 60.00 | 34.29 | 52.11 |
| – w/ BEAR-Agent | **74.44** | **0.479** | **76.03** | **59.84** | **66.67** | **42.86** | **61.29** |
| InternVL3-14B | 32.84 | 0.279 | 43.36 | 30.78 | 37.00 | **28.57** | 33.93 |
| – w/ one-shot | 33.40 | 0.289 | 44.27 | 31.10 | 38.00 | **28.57** | 34.04 |
| – w/ few-shot | 34.23 | 0.298 | 44.73 | 31.53 | **39.00** | 25.71 | 34.16 |
| – w/ CoT | 25.59 | 0.231 | 41.23 | 33.57 | 37.84 | 0 | 26.88 |
| – w/ BEAR-Agent | **37.96** | **0.303** | **47.90** | **33.68** | **39.00** | **28.57** | **36.24** |

(a)

(b)

(c)

Figure 8. **Offline and online experiment results of BEAR-AGENT.** Due to space limits, we provide further details in Appendix K

formance across all categories, from low-level pointing to long-horizon reasoning, demonstrating its effectiveness on embodied tasks. Notably, the largest gains are observed in *Pointing*, *Bounding Box*, and *Trajectory Reasoning*, confirming that the integrated visual tools provide meaningful cues to support reasoning.

### 6.3. Online evaluation in simulation and real-robot

In Fig. 8(c), to validate the effectiveness of BEAR-AGENT and the diagnosis value of BEAR, We further conduct online evaluation including *(1)* **simulation experiments** in Maniskill (Gu et al., 2023) and *(2)* **real-robot experiment** with a UR5 with a custom 3D-printed fingertip.

**Experiment setup.** *(1)* For simulation experiments, we design three sets of basic manipulation tasks in the tabletop environment, each paired with four distinct language instructions that specify picking up a target object and placing it at a designated location. In Figure 8, *General task* requires picking up and placing objects by name, *Spatial task* involves grasping and placing objects at specified spatial locations, and *Part task* focuses on grasping functional parts. Instruction variants include commands such as 'pick up the top-right cube on the plate below' which direct the agent to attend to both object type and spatial relations. *(2)* For the real-robot experiments, we use a tabletop UR5 setup

with an agent-view camera. Tasks are specified through 10 different instructions that describe picking up a target object, e.g. 'pick up the smallest bottle on the table.'.

**Baseline.** *(1)* **In simulation**, we adopt MOKA (Liu et al., 2024a) as our baseline method. MOKA employs GPT-4v (Hurst et al., 2024) as its backbone to generate keypoints from top-down RGB observations and plan motions to complete the task. The keypoints include a grasp point for object picking, a target point for placement, and intermediate waypoints for motion planning. In our implementation, we integrate BEAR-Agent to support MOKA in the keypoint selection process. In Figure 8(b), we perform 20 rollouts for each language variation and report the task-level average success rate. *(2)* **In real-robot**, our baseline is InternVL3-8B. We prompt methods to give points under agent view in identifying the target objects, and the success rate is calculated by 30 rollouts. Further experiment details are provided in Appendix K.

**Result analysis.** *(1)* In Fig. 8(b), experiments demonstrate an average *20.17%* improvement in task performance when BEAR-Agent is integrated with MOKA, as long as *(2) 23.4%* improvement over InternVL3-8B. Results shows that BEAR-Agent effectively enhances the decision-making process of MLLMs in keypoint selection for manipulation tasks, further highlighting diagnosis value of BEAR.

# 7. Conclusion

In this work, we identify a key gap in embodied capability diagnosis and propose BEAR, the first benchmark for diagnosing MLLM embodied skills. Through hierarchical skill-level analysis, we uncover two fundamental limitations: perceptual bottlenecks and unstable spatiotemporal modeling. Based on these findings, we introduce an actionable solution, BEAR-AGENT, a multimodal conversable agent with tool augmentation. We demonstrate its effectiveness in both offline evaluation and real-world online settings, achieving over 20% performance gains across models such as InternVL and GPT-5, further highlighting the diagnostic value of BEAR. We hope our work provides actionable insights to development of more general embodied agents.

# 8. Limitations

BEAR-Agent is currently developed and evaluated within a limited scope. Future work could extend it by designing targeted training data for supervised fine-tuning and reinforcement learning, enabling stronger performance and broader applicability.

# Impact Statement

This paper presents work whose goal is to advance the field of machine learning. There are many potential societal consequences of our work, none of which we feel must be specifically highlighted here. In Appendix B, all data used in this study are collected and processed in accordance with ethical guidelines, and do NOT contain personally identifiable or sensitive private information.

# Acknowledgment

This material is based upon work supported by the National Science Foundation under Award No. 2107256.

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

# Contents

# A. More Related Work

## A.1. Multimodal Large Language Models

Multimodal large language models (MLLMs) have advanced significantly by integrating large language models (LLMs) with visual understanding. Early work focused on vision-language alignment (Chen et al., 2020; Li et al., 2020; Tan & Bansal, 2019), while recent approaches employ visual encoders and adapters to map features into linguistic space for joint reasoning (Radford et al., 2021; Yu et al., 2022; Zhang et al., 2021). This improves performance on tasks such as VQA and captioning, and enables zero-shot generalization in areas like robotics and autonomous driving. Representative MLLMs (Hu et al., 2024; Comanici et al., 2025; Zhu et al., 2025; Team, 2025; Lu et al., 2024; Li et al., 2024b; Dubey et al., 2024; Anthropic, 2025) exemplify the state of the art in cross-modal reasoning and extend the reach of multimodal learning to diverse applications.

## A.2. Benchmarking MLLMs in Embodied Capabilities

Embodied capabilities encompass an agent's ability to perceive, comprehend, and interact with the physical world. Existing benchmarks often target specific domains, such as pointing (Yuan et al., 2024; Zhou et al., 2025; Ji et al., 2025; Team et al., 2025; He et al., 2022; Fu et al., 2024), bounding box (Schulter et al., 2023; Chiang et al., 2024), spatial reasoning (Yang et al., 2025a; Rajabi & Kosecka, 2024; Zhang et al., 2025; Wang et al., 2025), motion understanding (Hong et al., 2025; Li et al., 2024c), task planning (Qiu et al., 2024; Chen et al., 2023; Ying et al., 2024), multi-agent collaboration (Qin et al., 2025), and embodied tasks in simulation (Yang et al., 2025b). To our knowledge, no comprehensive benchmark exists. We therefore introduce BEAR, the first fine-grained embodied reasoning benchmark with carefully designed category distributions, and *compare it against related benchmarks in Table 2*.

## A.3. MLLMs as Embodied Agents

Recently, MLLMs show promise as embodied agents, capable of perceiving multimodal inputs, reasoning over them, and generating actions for navigation, manipulation, and interactive tasks. Early systems such as PaLM-E (Driess et al., 2023) and SayCan (Ahn et al., 2022) connected language instructions to robotic actions through grounding and affordance-based planning. Generalist models (Reed et al., 2022) like Flamingo (Alayrac et al., 2022), GPT-4V (OpenAI, 2023), and InstructBLIP (Dai et al., 2023) demonstrated the ability to process interleaved modalities for diverse reasoning and action, and frameworks such as MM-ReAct (Yang et al., 2023b) and Voyager (Wang et al., 2023) further illustrate how LLMs can orchestrate external perception tools or acquire skills through open-ended exploration. In this work, we introduce BEAR-Agent, a conversable multimodal agent that integrates pretrained vision models to enhance perception, 3D understanding, and planning, offering a more targeted step toward robust multimodal embodied intelligence.

*Table 2.* **Category-level differences between BEAR and some existing benchmarks.** BEAR encompasses 6 categories, and we offer detailed descriptions of how each category differs from its most comparable counterpart in prior benchmarks.

| Benchmark | Category | Difference |
|---|---|---|
| Where2Place (Yuan et al., 2024), ReferBench (Zhou et al., 2025), BLINK (Fu et al., 2024) | Pointing | BEAR includes three different fine-grained pointing skills. An additional feature of our benchmark design is the integration of explicit difficulty control. *__In the meantime, BEAR also has other categories instead of only Pointing__*. |
| OmniLabel (Schulter et al., 2023), LocateBench (Chiang et al., 2024) | Bounding Box | BEAR includes three different fine-grained bounding box skills with thoughtfully designed difficulty control. *__In the meantime, BEAR also has other categories instead of only Bounding Box__*. |
| ERQA-Benchmark (Team et al.) | Trajectory Reasoning | For trajectory reasoning, BEAR includes three different embodiment, including human hands, gripper and object. Moreover, we include a broader range of dynamic motions and actions, such as *pick up*, *place*, *wipe*, and related manipulation skills. *__In the meantime, BEAR also has other categories instead of only Trajectory Reasoning__*. |
| VSI-Bench (Yang et al., 2025a) | Spatial Reasoning | Instead of general spatial understanding abilities, we emphasize atomic skills that are necessary for robot navigation, which include *Path Planning*, *Relative Direction*, *Object Localization*. *__In the meantime, BEAR also has other categories instead of only Spatial Reasoning__*. |
| Ego-Plan (Chen et al., 2023), Ego-Plan2 (Qiu et al., 2024) | Task Planning | We share the same motivation as Ego-Plan and Ego-Plan2 on *Next Action Prediction*, but extend the action space by incorporating necessary navigation actions, such as 'navigate to the toaster'. In the meantime, we introduce *Task Process Reasoning*, which focuses on assessing an agent's ability to understand and reason about the current stage and past activities of a task relative to its overall goal. *__In the meantime, BEAR also has other categories instead of only Task Planning__*. |
| EmbodiedBench (Yang et al., 2025b) | Long-horizon | *__EmbodiedBench__* provides valuable insights by introducing capability-oriented tasks, instead of other works only focusing on the overall success rate of each task. However, *__each task in EmbodiedBench includes multiple skill-oriented steps__*. for example, EmbodiedBench includes multiple navigation tasks, but each navigation task contain skills of *__path planning__* for navigation to the target object, *__pointing__* for target object recognition. EmbodiedBench evaluates the overall sucess rate without decomposing each task into atomic skill-oriented steps. However BEAR contains 14 atomic capability-oriented skills that can cover the execution steps of embodied tasks. |
| EmbodiedAgentInterface (Li et al., 2024d) | Long-horizon | *__EmbodiedAgentInterface__* provides a valuable framework for MLLM deployment to evaluate their decision-making abilities through symbolic representations. In contrast, our work focuses on a holistic evaluation and taxonomy of the perception and reasoning skills underlying embodied capabilities in MLLMs. Our approach serves as a diagnostic benchmark for comprehensively testing and analyzing model performance across different visual reasoning dimensions. |

# B. Ethics Statement

Our benchmark involves datasets collected from publicly available sources. All datasets used are either publicly released under appropriate licenses and have undergone ethical review by their respective publishers. We do not collect or distribute any personally identifiable information. We do not contain harmful or sensitive data. For human annotation and multi-stage verification, all annotators were recruited with informed consent and not exposed to harmful or sensitive content. Our benchmarks are intended for academic research purposes.

**Data Privacy and Consent.**    All data used in this study are either collected from publicly available open-source datasets or generated through simulation environments. We ensure that all datasets used comply with their respective licenses, which are listed as follows. No personally identifiable information (PII) is present in any data, and no real-world user data was collected for this work. Additionally, we manually removed any potentially sensitive visual content to ensure that all data used in our benchmark is anonymized, non-harmful, and ethically safe for public release.

**Datasets and Licenses**

- Ego4D (Grauman et al., 2022)    CC BY 4.0 License
- Epic-Kitchens (Damen et al., 2018)    CC BY 4.0 License
- OpenImages V7 (Kuznetsova et al., 2020)    CC BY 4.0 License
- PartImageNet (He et al., 2021)    No explicit license specified. The dataset and scripts are publicly released by the authors. We use it strictly for non-commercial academic research.
- AGD20K (Luo et al., 2022)    MIT License
- Open-X-Embodiment Dataset (O'Neill et al., 2024)    CC BY 4.0 License
- ScanNet (Dai et al., 2017)    Customized Terms of Use
- ScanNet++ (Yeshwanth et al., 2023)    Customized Terms of Use
- ArkitScene (Baruch et al., 2021)    Apple Custom Non-Commercial License
- TASTE-Rob (Zhao et al., 2025)    Customized Terms of Use
- AI2-THOR, RoboTHOR, ManipulaTHOR (Kolve et al., 2017; Deitke et al., 2020; Ehsani et al., 2021)    Apache License 2.0

**Annotators.**    15 human annotators are involved in labeling data or evaluating tasks, they are recruited voluntarily and provided informed consent prior to participation. The anotators are clearly informed about the purpose of the study, the nature of the data they will interact with, and their rights to withdrawal. The annotator pool primarily consisted of undergraduate, master's, and Ph.D. students from STEM-related fields, with distribution listed as follows in Figure 9. We ensure fair compensation and treated all annotator contributions ethically and respectfully.

**Human Studies.**    To establish a human performance baseline, we conduct user studies involving 5 human participants. All participants are above age 18 who are provided with informed consent prior to participation. They are briefed on the task goals, data usage policy, and their right to withdraw at any time. No PII is collected during the study. This study does not contain any harmful or sensitive data.

# C. Benchmark Category and Statistics

## C.1. Pointing

### C.1.1. OVERVIEW

**Question Format.**    Given an image and a natural language instruction, the *Pointing* category requires the Vision-Language Model (VLM) to predict a normalized 2D coordinate $(x, y)$ in the image, where $x, y \in [0, 1]$. Here, $x$ represents the horizontal position from left (0) to right (1), and $y$ represents the vertical position from top (0) to bottom (1). $x$ is the This coordinate indicates the target pixel location corresponding to the instruction.

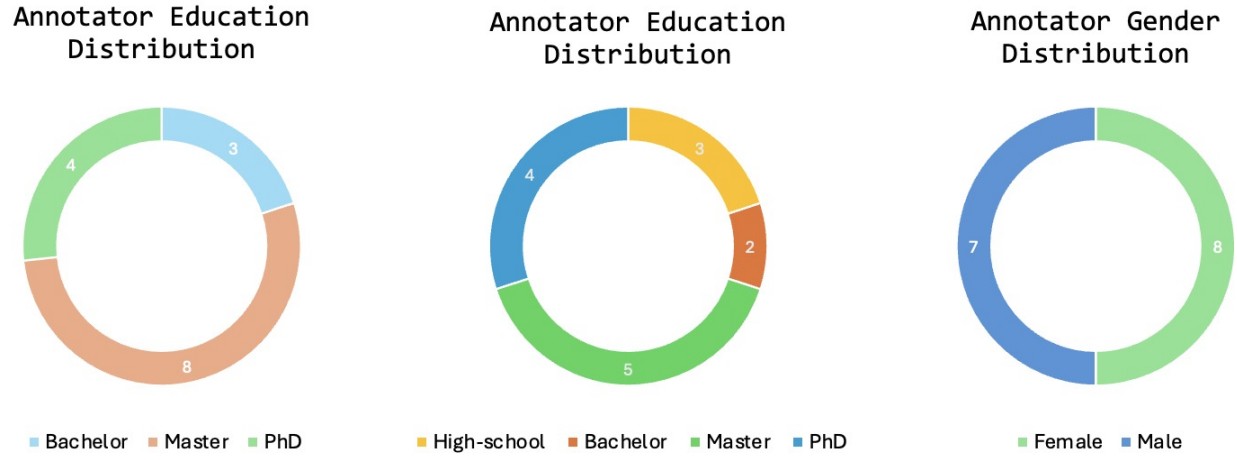

*Figure 9.* **Annotator and Human Participant Distribution.**

**Category.**    The pointing category comprises three sub-category: *General Object Pointing, Spatial Relationship Pointing*, and *Semantic Part Pointing*.

**Significance.**    Pointing is a core embodied reasoning skill, bridging perception, language understanding, and action planning. In real-world embodied scenarios, agents must resolve ambiguous references, comprehend spatial relations, and localize object parts for tasks.

### C.1.2. GENERAL OBJECT POINTING

**Definition.**    Given an image as input, the task requires the VLM to identify an object based on a detailed linguistic description and to localize it by pointing to its pixel-level coordinates in the image. The description may include fine-grained semantic attributes such as color, type, and specific identifiers. For example, the instruction may specify: 'Identify the red Audi car with the blue and red '1' on its body.'

**Significance.**    General object pointing is a fundamental embodied reasoning task that requires grounding natural language descriptions into object identification in the visual scene. It tests the ability of the MLLMs to align perception and language for fine-grained object recognition. This capability is essential for daily human interactions and serves as a basis for subsequent visual reasoning and embodied actions such as object tracking, grasping, manipulation, or navigation.

### C.1.3. SPATIAL RELATIONSHIP POINTING

**Definition.**    Given an image and relational cues, such as 'closest', 'nearest to', 'behind', 'to the left of', the model must give point to the correct target object. This sub-task evaluates the VLM's capacity to interpret and reason about spatial relationships between objects. For instance: 'Point to the farthest chair in the second column from left to right', 'Point to the object on top of the microwave', 'Point to the nearest car in the image'

**Significance.**    Spatial relationship pointing is a fundamental component of embodied intelligence. In both real-world and simulated environments, objects are often arranged in complex spatial configurations. Therefore, it is essential for models to accurately interpret spatial relationships such as 'in front of', 'behind', 'on top of' or 'to the left of'. Furthermore, object category information alone is often insufficient for disambiguation—for example, scenes may contain multiple instances of the same object type, such as several chairs or cups. In these cases, correctly identifying the target object requires understanding its relative position with respect to other reference objects. Mastering this capability is critical for tasks where instructions frequently rely on spatial references rather than absolute object descriptions.

*Figure 10.* **Example of General Object Pointing.**

### C.1.4. SEMANTIC PART POINTING

**Definition.** Given an image as input, the VLM must identify and point to specific semantic parts of an object, based on natural language descriptions. This task focuses on fine-grained localization of object parts rather than whole objects. For example, 'Point to the handle of the ax.' or 'Point to the string area of the badminton racket.'

**Significance.** Part-level perception is essential for fine-grained interaction and embodied decision-making. Many real-world tasks require not only recognizing an object but also understanding its semantic components. For example, effective tool use, object manipulation, or human-robot collaboration often depends on identifying specific parts such as handles, switches, buttons, or spouts. By evaluating a model's ability to localize and point to object parts based on natural language instructions, this task assesses the VLM's capacity for fine-grained visual understanding beyond object-level recognition. It moves beyond simple object detection, requiring nuanced perception that is critical for downstream tasks such as grasp planning, part-based affordance reasoning, and interactive instruction following.

### C.2. Bounding Box

#### C.2.1. OVERVIEW

**Question Format.** Given an image and a natural language instruction, the *Bounding Box* category requires the Vision-Language Model (VLM) to predict a 2D bounding box in the image, specified by $(x_{\min}, y_{\min}, x_{\max}, y_{\max})$, $x_{min}, y_{min}, x_{max}, y_{max} \in [0, 1]$. Here, $x$ represents the horizontal position from left (0) to right (1), and $y$ represents the vertical position from top (0) to bottom (1). The predicted bounding box should precisely localize the target object or region described in the instruction.

**Category.** This category includes three sub-tasks: *General Bounding Box*, *Spatial Relationship Bounding Box* and *Part-level Bounding Box*.

**Significance.** 2D bounding box prediction is a fundamental capability for embodied vision and reasoning. Unlike simple point-based localization, this task requires the model to infer both the position and the spatial extent of the target object or semantic part. Accurately estimating not only where an object is but also its precise spatial localization is critical for downstream tasks such as manipulation, grasp planning, affordance understanding, and object tracking in interactive environments.

*Figure 11.* **Example of Spatial Relationship Pointing.**

**Data Source.** We reuse the *Pointing* category while removing samples with ambiguous bounding box ground truth.

### C.2.2. GENERAL OBJECT BOUNDING BOX

**Definition.** Given an image as input, the task requires the VLM to give 2D bounding box to an object based on a detailed linguistic description and to localize it by pointing to its pixel-level coordinates in the image. Similar to General Object Pointing, the description may include fine-grained semantic attributes such as color, type, and specific identifiers. For example,

**Significance.** General Object 2D Bounding Box Prediction evaluates the ability of a multi-modal large language model to localize and delineate specific objects in space based on detailed natural language descriptions.

### C.2.3. SPATIAL RELATIONSHIP BOUNDING BOX

**Definition.** Given an image and relational cues—such as 'closest' 'nearest to' 'behind'—the model must give the correct 2D bounding box corresponding to the target object. This task evaluates the VLM's ability to interpret and reason about spatial relationships between objects. For example: 'Identify the farthest chair in the second column from left to right', 'Select the bounding box of the object on top of the microwave'.

**Significance.** Spatial relationship-based 2D bounding box prediction is essential for embodied intelligence. In complex scenes, models must interpret cues like 'in front of' or 'next to' to select the correct object, especially when multiple instances of the same category exist. This ability is critical for tasks where instructions rely on relative positioning, not just object labels.

### C.2.4. SEMANTIC PART BOUNDING BOX

**Definition.** Given an image as input, the VLM must identify and predict the boundingbox of specific semantic parts of an object based on natural language descriptions. Unlike whole-object localization, this task targets fine-grained part-level understanding. For example, 'Point to the handle of the toothbrush' or 'Point to the lid of the kettle'.

**Significance.** Part-level bounding box prediction is important for fine-grained interaction in embodied tasks. Real-world activities, such as tool use and object manipulation, require not only recognizing objects but also understanding their functional parts. This task evaluates a model's ability to ground language to semantic components, supporting affordance

*Figure 12.* **Example of Semantic Part Pointing.**

reasoning, grasping, and decision-making.

## C.3. Trajectory Reasoning

### C.3.1. OVERVIEW

**Definition.** In *trajectory reasoning*, the model is required to infer the expected direction or path of motion based on the type of action (for example, opening, lifting, picking up, placing, pushing) and the spatial and interaction context. The trajectory may involve movements of different embodiment, such as human hands and robot gripper, towards specific objects or locations, or manipulations of objects such as opening a drawer, lifting an item.

**Question Format.** **This is a single-choice question out of four different choices.** Each question presents three arrows, randomly selected from four possible colors (red, green, yellow, and blue), along with a fourth option: 'None of the above' indicating that none of the arrows represents the correct direction. By default, all arrows are assumed to have the correct origin point. The VLM is required to select the single option corresponding to the correct directional cue.

**Category.** This category includes three subtasks: *Object Trajectory Reasoning*, *Human Hand Trajectory Reasoning*, *Gripper Trajectory Reasoning*.

**Challenges.** *Trajectory Reasoning* requires the model to integrate multiple factors to infer accurate motion patterns. First, the model must account for object geometry, as different shapes afford different directions of movement. Second, it must consider viewpoint variations. For example, the trajectory for opening a door differs depending on whether the handle is viewed from the front or side. Third, the model must understand the action semantics and physical regularities of motion, such as knowing that pulling and pushing a door result in opposite trajectories, bottle caps typically open via counterclockwise rotation, or zippers move along the fastening track. Finally, the model must exhibit precise visual reasoning ability to judge whether a given direction leads toward a functional goal or causes it to veer off-course.

**Significance.** *Trajectory Reasoning* bridges the gap between identifying where to act and understanding how to act. While object localization tasks such as pointing or predicting a bounding box reveal static spatial intent, embodied agents must further infer the dynamic process of interaction, which is how an object, hand, or gripper moves through space to accomplish a task. This reasoning capability is essential for modeling continuous, goal-directed behavior in real-world environments, such as opening a drawer, pouring water. It reflects a deeper level of embodiment, where agents not only locate affordances,

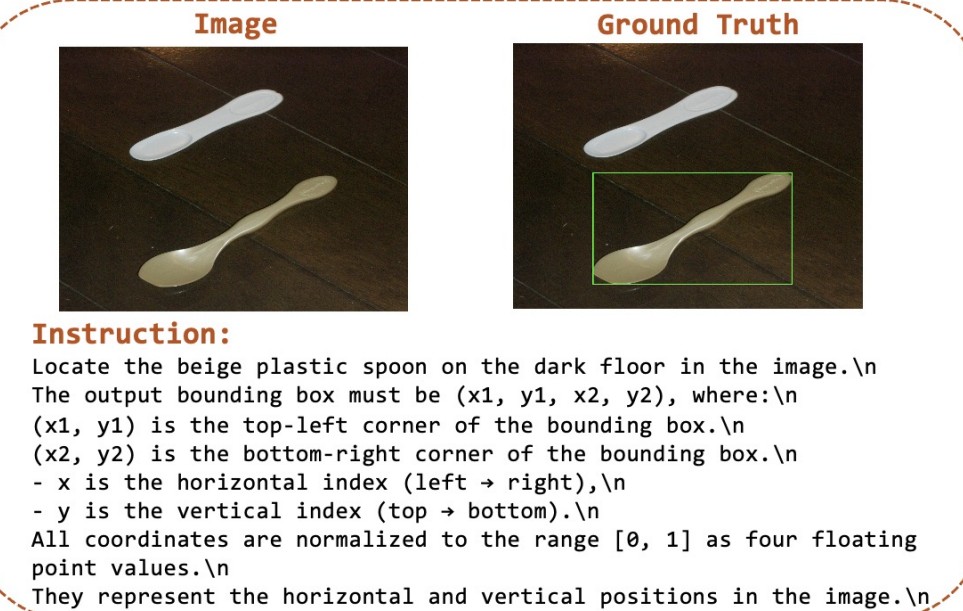

*Figure 13.* **Example of General Object Bounding Box Prediction.**

but also anticipate and align with the temporal and kinematic structure of actions.

### C.3.2. OBJECT TRAJECTORY REASONING

**Definition.**   Given an image as input, the model is required to infer the expected direction or path of motion for an object or a part that is being acted upon, for example, the object is being opened, lifted, or pushed, based on the type of action (for example, opening, lifting, picking up, placing, pushing) and the spatial and interaction context. This task focuses solely on object motion, without involving any embodiment. All arrows are assumed to originate from the correct starting position. The model only needs to reason about whether the arrow direction aligns with the motion of the intended object.

**Significance.**   *Object Trajectory Reasoning* enables models to understand how various objects or components move in response to different actions. This ability is essential for interpreting and predicting the physical dynamics of interactions across diverse objects and contexts. Furthermore, it provides actionable guidance for embodied agents to interact effectively with different objects.

**Challenges.**   *Trajectory Reasoning* involves two key challenges. First, the model must infer the underlying **object dynamics**, which often follow physical regularities. For example, it should understand that pulling and pushing a door produce opposite trajectories, bottle caps are typically opened via counterclockwise rotation, or zippers move along a predefined fastening track. These dynamics are closely tied to the object's geometry—different shapes afford different types or directions of motion. Second, the model must be robust to variations in different **viewpoint**. The perceived motion path can differ depending on where the object is viewed from. For instance, opening a door looks different when seen from the front versus the side. The model must reason across frames and viewpoints to accurately infer the intended direction of motion and distinguish between goal-directed and off-course trajectories.

### C.3.3. GRIPPER TRAJECTORY REASONING

**Definition.**   Given an image as input, the model is required to infer the expected direction or path of motion of a robotic gripper in order to reach and grasp a specific object. The gripper trajectory depends on the spatial layout of the scene, the shape, position and orientation of the target object. This task focuses solely on the gripper's motion, without requiring reasoning about the object's subsequent motion. All arrows are assumed to originate from the current gripper TCP, short for Tool Center Point. The model only needs to decide whether the arrow direction aligns with a feasible and purposeful motion for reaching and grasping the target object.

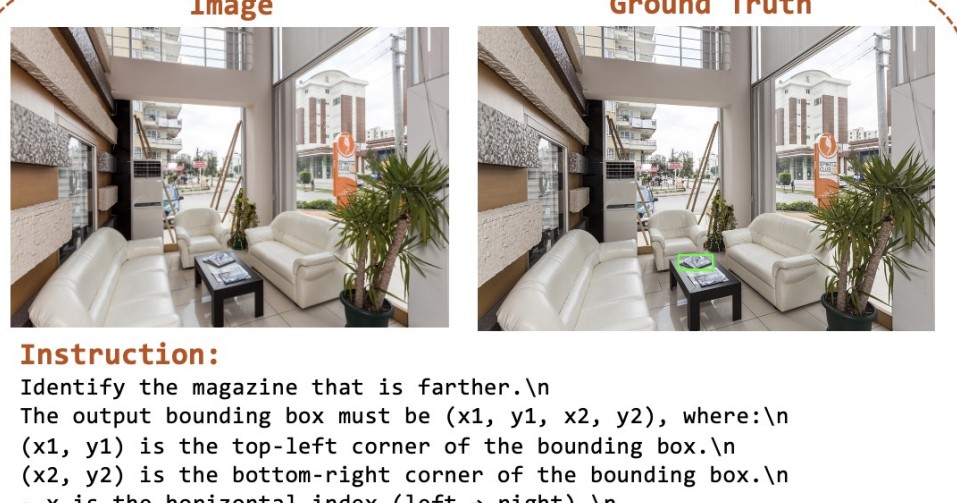

*Figure 14.* **Example of Spatial Relationship Bounding Box Prediction.**

**Significance.** This task evaluates the model's ability to reason about spatial relations and motion planning for goal-directed robotic actions.

**Challenges.** The key challenges lie in identifying the correct object among many in the scene and reasoning about its spatial position in relation to the trajectory direction.

### C.3.4. HUMAN HAND TRAJECTORY REASONING

**Definition.** Given an image as input, the model is required to infer the expected direction or path of motion of a human hand in order to reach and grasp, place, open a certain object. The gripper trajectory depends on the spatial layout of the scene, the shape, position and orientation of the target object. This task focuses solely on the gripper's motion, without requiring reasoning about the object's subsequent motion. All arrows are assumed to originate from the current gripper TCP, short for Tool Center Point. The model only needs to decide whether the arrow direction aligns with a feasible and purposeful motion for reaching and grasping the target object.

**Significance.** Understanding the trajectory of a human hand for fundamental skills like reaching, grasping and placing is crucial for VLMs to infer human intent, anticipate interactions with objects, and build embodied understanding from visual scenes. This task enables models to reason about early-stage physical interactions, which is foundational for downstream applications such as action prediction, affordance understanding, and instruction following.

**Challenges.** The key challenges lie in identifying the correct object among many in the scene and reasoning about its spatial position, such as 'on top of', 'to the left of', or 'to the right of', in relation to the hand trajectory direction.

### C.4. Spatial Reasoning

### C.4.1. OVERVIEW

**Definition.** Spatial reasoning refers to an agent's ability to understand 3D space. For an embodied agent, it requires a basic comprehension of the environment—such as what objects are present and how they relate to each other. Additionally, the agent needs a sense of self-location and orientation in order to effectively plan a path. This category takes as input either

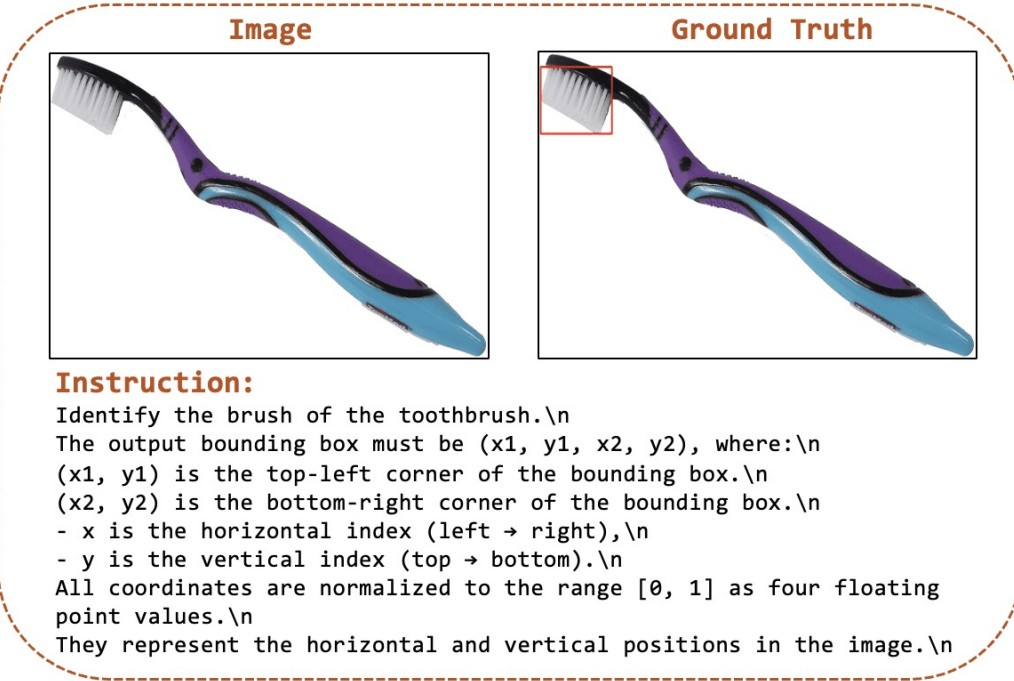

*Figure 15.* **Example of Semantic Part Bounding Box Prediction.**

a video or a video plus the current observation (which can be an image or the last frame of the video). The question format is multiple-choice with four options.

**Question Format.** **This is a single-choice question with four options labeled A, B, C, and D.** Each option describes a possible spatial relation, and potential path about the question. The model is required to select the one and only correct answer based on the given descriptions.

**Category and Significance.** There are three sub-tasks under this task, including object localization, relative direction and path reasoning. These tasks are closely interrelated and each plays a crucial role in embodied navigation and spatial reasoning. *Object Localization* is a foundational skill, requiring the agent to process a video segment and determine the spatial relationships between itself, various objects, and key landmarks in the environment. This is essential because accurate object localization allows the agent to build a mental map of its surroundings, which supports more effective decision-making and goal-directed behavior. *Path Planning* evaluates the agent's ability to perform coarse-level navigation. Given the spatial understanding of object and landmark positions, the agent must estimate an approximate path toward the target. This involves high-level decisions such as selecting a general direction or identifying intermediary waypoints. The task focuses not on precise control but on whether the agent can reason about the environment to avoid major obstacles and select a feasible route. It reflects the agent's competence in integrating object localization and relative direction information to form a navigational strategy that is both efficient and safe. *Relative Direction* builds on this by enabling the agent to understand its position relative to a specific object. This finer-grained spatial awareness allows for more precise planning, such as aligning with or approaching a target from a particular angle.

### C.4.2. OBJECT LOCALIZATION

**Definition.** Based on given video as input, *Object Localization* refers to the agent's ability to identify the spatial positions of relevant objects and landmarks for a given object. The potential options could be 'on the top of the sink', 'near the refridgerator', Given a video segment as input, the agent must perceive and understand the positional relationships between various objects, including key reference points (e.g., tables, doors) that may serve as navigation landmarks. This task lays the foundation for downstream reasoning, enabling the agent to build a mental map of the scene and prepare for actions such as navigation or interaction.

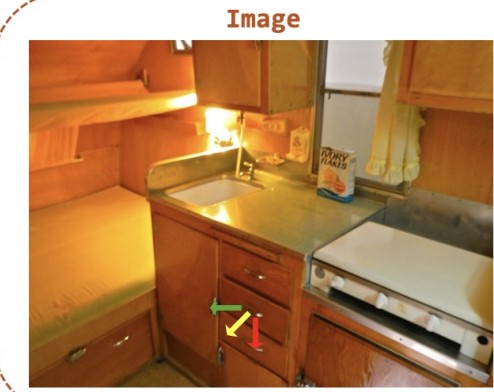

Image

**Instruction:**
Please directly output the selected
option.\n

Question: Which arrow in the image best
indicates the direction in which the
drawer will move when open it\n

Options:\n
A. Red Arrow.\n
B. Green Arrow.\n
C. Yellow Arrow.\n
D. None of the above.

**Ground Truth: C**

*Figure 16.* **Example of Object Trajectory Reasoning.**

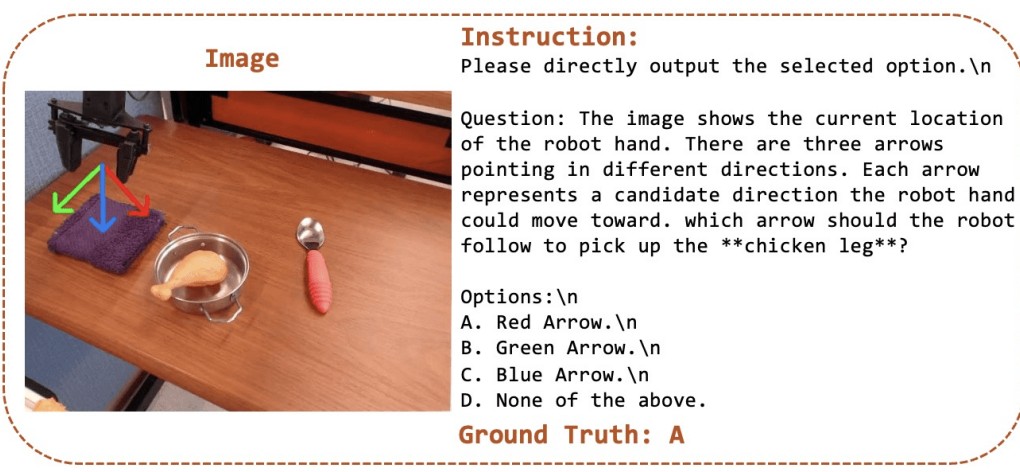

Image

**Instruction:**
Please directly output the selected option.\n

Question: The image shows the current location
of the robot hand. There are three arrows
pointing in different directions. Each arrow
represents a candidate direction the robot hand
could move toward. which arrow should the robot
follow to pick up the **chicken leg**?

Options:\n
A. Red Arrow.\n
B. Green Arrow.\n
C. Blue Arrow.\n
D. None of the above.

**Ground Truth: A**

*Figure 17.* **Example of Gripper Trajectory Reasoning.**

**Significance.** This task lays the foundation for downstream reasoning, enabling the agent to build a mental map of the scene and prepare for actions such as navigation or interaction. Landmarks and other objects in the scene play a critical role by anchoring the agent's spatial understanding, helping it to orient itself within the environment and reason about where to search for task-relevant objects. By grounding object positions relative to stable, easily recognizable features in the environment, the agent can more effectively generalize across scenes and plan robust behaviors in novel layouts.

**Challenge.** *Object Localization* is challenging due to partial observability, requiring the agent to accumulate spatial cues over time. It must interpret references like "near the table" by combining spatial relations with scene semantics, track stable landmarks, and convert egocentric views into a global map. These demands make localization critical for robust navigation and spatial understanding.

### C.4.3. PATH PLANNING

**Definition.** Given a video as context and a current observation image, this task assesses the agent's ability to plan a route to a target object by understanding the scene layout and identifying key landmarks. It requires spatial reasoning to determine feasible directions, such as 'turn left at the refrigerator' or 'walk past the couch.'

**Significance.** Path planning is essential for MLLMs to perform effective navigation and interaction. It enables the model to build a coarse map of the environment, avoid obstacles, and plan routes toward task-relevant objects.

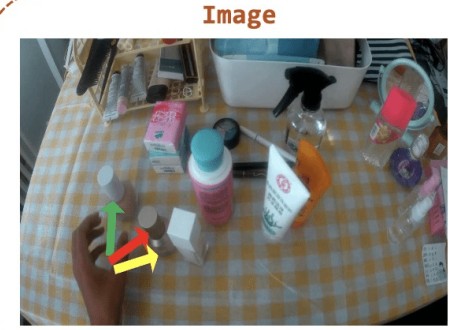

*Figure 18.* **Example of Human Hand Trajectory Reasoning.**

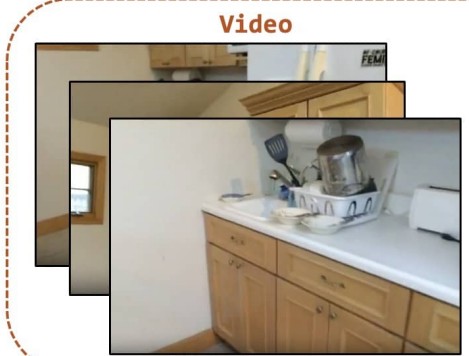

*Figure 19.* **Example of Object Localization.**

### C.4.4. RELATIVE DIRECTION

**Definition.**   Given a video as context and a current observation image, this task assesses the agent's ability to plan a route to a target object by understanding the scene layout and identifying key landmarks. It requires spatial reasoning to determine feasible directions, such as 'turn left at the refrigerator' or 'walk past the couch.'

**Significance.**   *Path Planning* is essential for MLLMs to perform effective navigation and interaction. It enables the model to build a coarse map of the environment, avoid obstacles, and plan routes toward task-relevant objects.

### C.5. Task Planning

### C.5.1. OVERVIEW

**Definition.**   *Task planning* refers to an agent's ability to understand tasks that have already occurred and to predict the next appropriate action required to achieve a high-level goal. For example, in the context of making coffee, after placing the cup on the coffee machine, one would typically proceed to turn on the machine to begin brewing. An intelligent agent must be capable of interpreting past actions, such as recalling the sequence in which they occurred, and reasoning about what action should follow to accomplish the overarching task.

**Question Format.**   **This is a single-choice question with four options labeled A, B, C, and D.** Each option describes a possible answer about the question. The model is required to select the one and only correct answer based on the given descriptions.

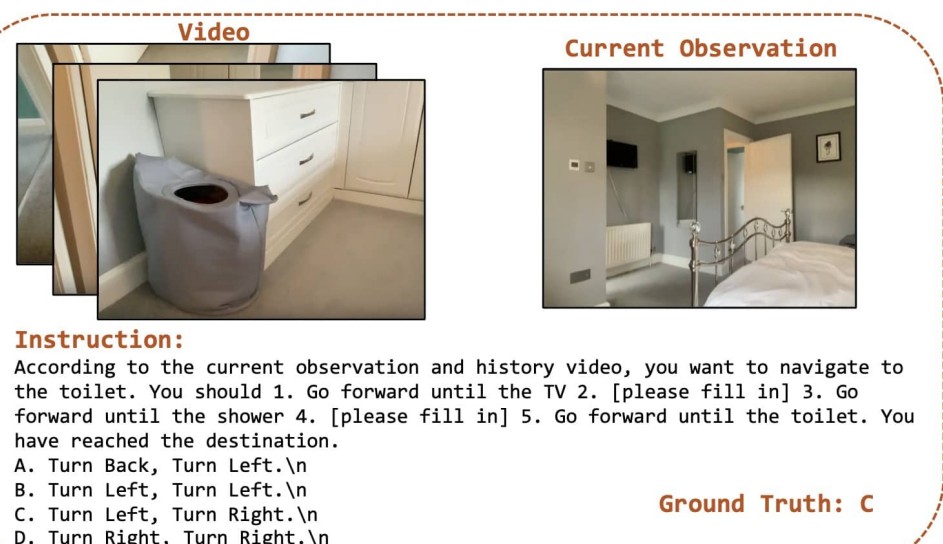

*Figure 20.* **Example of Path Planning.**

**Category and Significance.** The benchmark comprises two categories: task process reasoning and next action prediction. In task process reasoning, the input is a video segment, and the primary goal is to assess whether the model can recall the sequence of previously executed actions and their temporal order. For example, a representative question might be: 'Which of the following actions is not performed after picking up plate?', 'What action occurs immediately after drying the pot?' This task is crucial because it evaluates the model's temporal reasoning ability, which is essential for understanding the progression of multi-step activities and maintaining coherence in long-horizon decision making. In next action prediction, the model is required to infer the most plausible subsequent action based on a given high-level goal. This task is essential for evaluating the model's capacity for goal-conditioned reasoning and anticipatory planning, which are critical for intelligent agents to operate effectively in dynamic environments by selecting actions that align with long-term objectives. One core challenge in this task lies in the need for the agent to retain and reason over temporally ordered past events while also anticipating future actions. This requires a strong capacity for temporal understanding, long-term memory, and goal-directed inference. For embodied agents, the ability to integrate historical context with future planning is fundamental to executing coherent, multi-step tasks in real-world environments. Achieving robust performance on such tasks demands not only accurate perception, but also a deep understanding of causality, task structure, and temporal dependencies.

C.5.2. TASK PROCESS REASONING

**Definition.** In *Task Process Reasoning*, the model is given a video segment and must recall past actions and their temporal order. The goal is to assess whether the model understands the sequence of events, for example: 'Which action is not performed after picking up the plate?' or 'What occurs after drying the pot?' This tests the model's temporal reasoning, essential for understanding and executing complex tasks.

**Significance.** *Task Process Reasoning* is critical for embodied understanding, as it requires the model to comprehend temporal information and accurately recall previously observed events. This ability to track and interpret the sequence of past actions is essential for intelligent agents to make informed decisions, reason about ongoing tasks, and maintain situational awareness in dynamic environments. Understanding what has already occurred lays the foundation for anticipating future steps and ensuring coherent task execution.

**Examples.** For detailed examples, we refer reders to Figure 91.

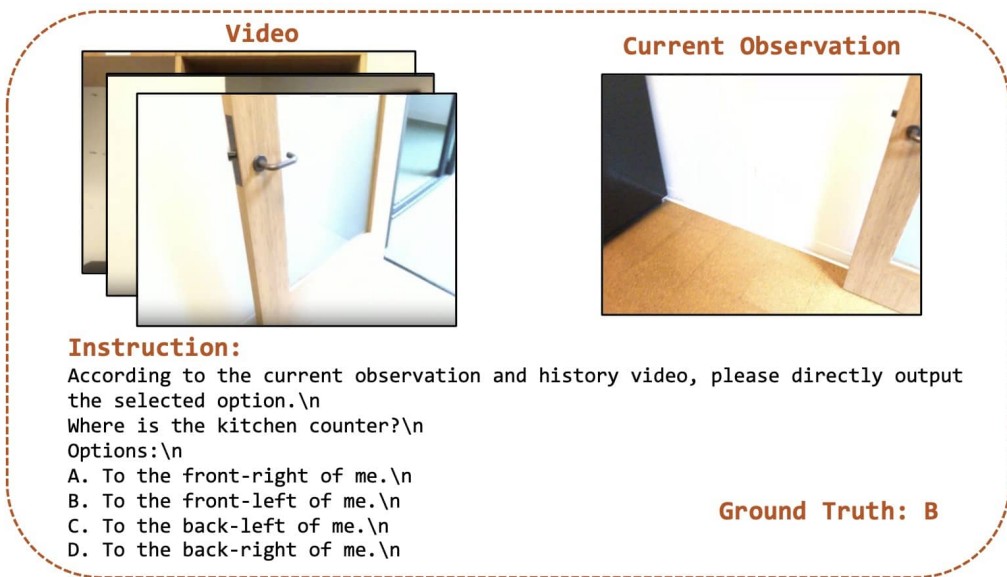

*Figure 21.* **Example of Relative Direction.**

### C.5.3. NEXT ACTION PREDICTION

**Definition.** *Next Action Prediction* involves providing the model with a video segment and a high-level goal, and assessing whether it can accurately anticipate the next action required to complete that goal. For example, given the question: 'Considering the progress shown in the video and my current observation in the last frame, what action should I take next for preparing the hot water and testing its temperature?', the model must choose from options such as 'put down kettle', 'pick up kettle', 'pour more hot water into glass' or 'none of the above' This task evaluates the model's ability to integrate temporal context and goal-directed reasoning to generate plausible next steps.

**Significance.** Humans possess an innate ability for planning, enabling them to decompose high-level goals into manageable steps and anticipate subsequent actions based on current progress. For embodied agents, the ability to break down complex tasks and predict the next appropriate action is a critical component of intelligent and goal-directed behavior.

**Examples.** For detailed examples, we refer reders to Figure 92.

### C.6. Long-horizon

**Definition.** In the *long-horizon category*, we use ManipulaTHOR (Ehsani et al., 2021) and RoboTHOR (Deitke et al., 2020) to collect 35 task-oriented episodes. Each episode is with a different high-level goal (e.g., 'pick up the apple and place it on the blue plate', 'open the drawer', 'wash the apple and place it on the blue plate', and each episode is decomposed into necessary reasoning steps, with each step posed as a VQA question. Each reasoning step can correspond to one of our 14 embodied reasoning skills, from *Next Action Prediction* for predicting next high-level action for accomplishing the task, to *Object Localization* to identify the location of the target object, *Path Planning* to plan the path to navigate to the location to be around the target object, *Relative Direction* for the agent to finely adjust its position beside the sink. After the agent is in front of the sink, and then predict the *Bounding Box* for coarsely recognize the object, and then predict the *Point* for interaction with it.

**Motivation for *Long-horizon* Task.** The long-horizon task is introduced to assess whether our taxonomy of embodied reasoning skills is not only theoretically grounded but also practically composable and operational. Each high-level instruction (e.g., 'wash the apple and place it on the blue plate') is broken into a sequence of VQA-style reasoning steps, where each step maps explicitly to one of BEAR's 14 atomic skills. This compositional structure allows us to: (i) demonstrate that complex tasks can be expressed as ordered chains of atomic abilities (*compositionality*); (ii) verify that our skill taxonomy covers the decision-making space of common household tasks (*completeness*); and (iii) enable fine-grained

diagnosis of model failure by pinpointing which sub-skill failed in the chain (*diagnosticity*). Thus, the long-horizon category serves as an empirical testbed to validate the utility, sufficiency, and interpretability of our skill taxonomy in realistic, multi-step task settings.

**Evaluation for *Long-horizon* Category.**   We adopt an offline evaluation protocol, where each collected episode is decomposed into a sequence of VQA-style structured reasoning steps. An episode is deemed successful only if the model correctly answers all associated questions. The final evaluation metric is the success rate, computed as the proportion of episodes solved successfully by the model.

**Examples.**   We provide examples of *Long-horizon* category in Figure 93 and Figure 94.

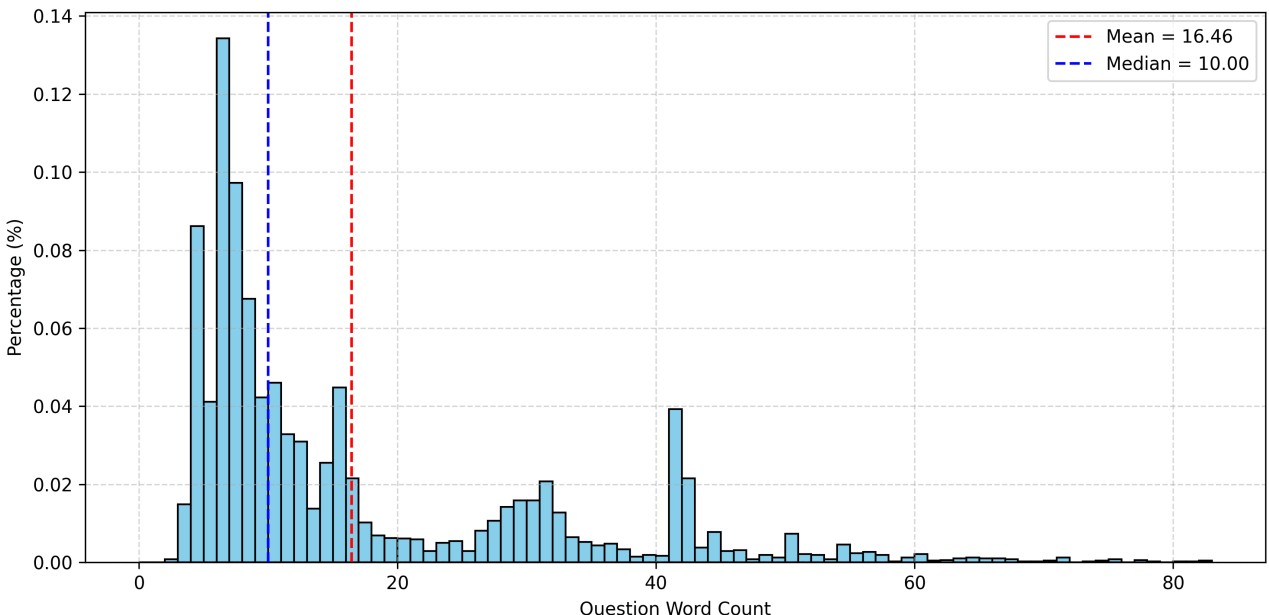

*Figure 22.* **The distribution of the number of words per question in BEAR.**

## D. Benchmark Distribution and Visualization Analysis

### D.1. Global statistics

**Question distribution.** Figure 22 illustrates the distribution of word counts in questions, showcasing the diversity and complexity within the dataset. The median number of words per question is 10, with the maximum question length reaching 82 words. The majority of the questions fall between 5 to 11 words. Questions exceeding the maximum threshold are grouped under the final bin for visualization clarity.

**Option distribution.** Figure 23 shows the distribution of word counts for individual options (excluding the choice letter, e.g., 'A.'). The median word count per option is 4, and the longest option contains 20 words.

**Image and video resolution.** Figure 24 displays the resolution distribution of images. A large proportion (79.5%) of images lie in the 512–1024 resolution range. Only a small fraction (0.2%) fall below 256 pixels, while high-resolution images above 2048 pixels are also rare (0.1%). Figure 25 presents video resolution statistics. The vast majority (93.4%) of videos fall in the 512–1024 resolution range, while only 6.6% reach the 1024–2048 range. No videos exceed 2048 pixels in resolution.

**Video frame number and video duration.** Figure 26 visualizes the distribution of frame counts per video. An overwhelming 95.8% of videos contain more than 60 frames, indicating long video clips. Only a small portion (4.2%) of videos have fewer than 60 frames. Figure 27 illustrates the duration distribution, revealing that only 3.4% of videos exceed 120 seconds. The majority of videos are shorter than 60 seconds, indicating that BEAR primarily consists of short-horizon tasks.

**Question word cloud and word frequency.** As shown in Figure 28 and Figure 29, our dataset contains a high frequency of concrete, action-related terms like "hand", "identify", "move", "arrow", and "robot", which reflects the emphasis on spatial reasoning and agent behavior. Our data highlights physical concepts and directional cues, which may help explain the performance gap in related tasks.

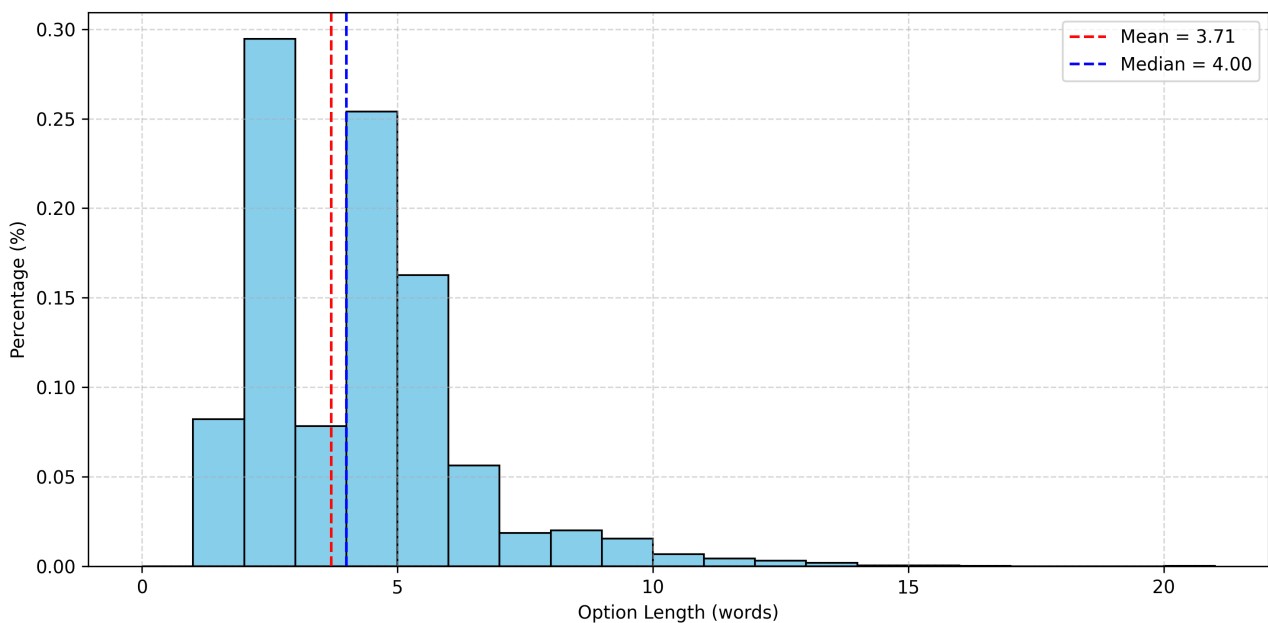

*Figure 23.* **The distribution of the number of words per option in BEAR.**

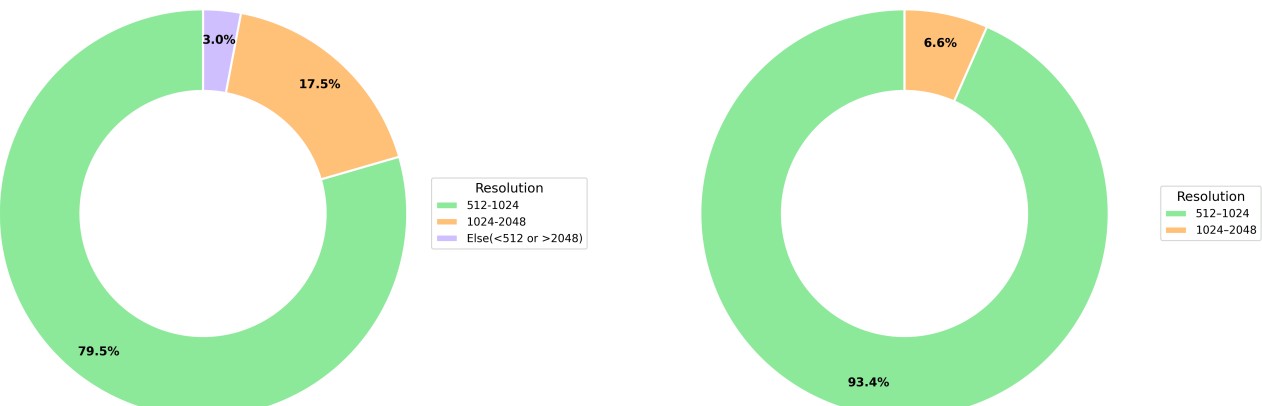

*Figure 24.* **BEAR image resolution distribution.**          *Figure 25.* **BEAR video resolution distribution.**

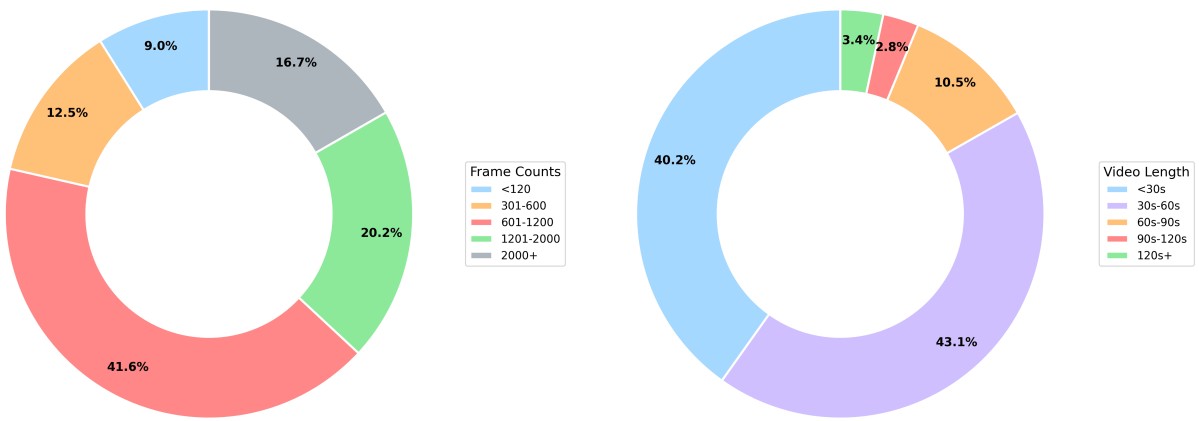

*Figure 26.* **BEAR video frame count distribution.**     *Figure 27.* **BEAR video duration distribution.**

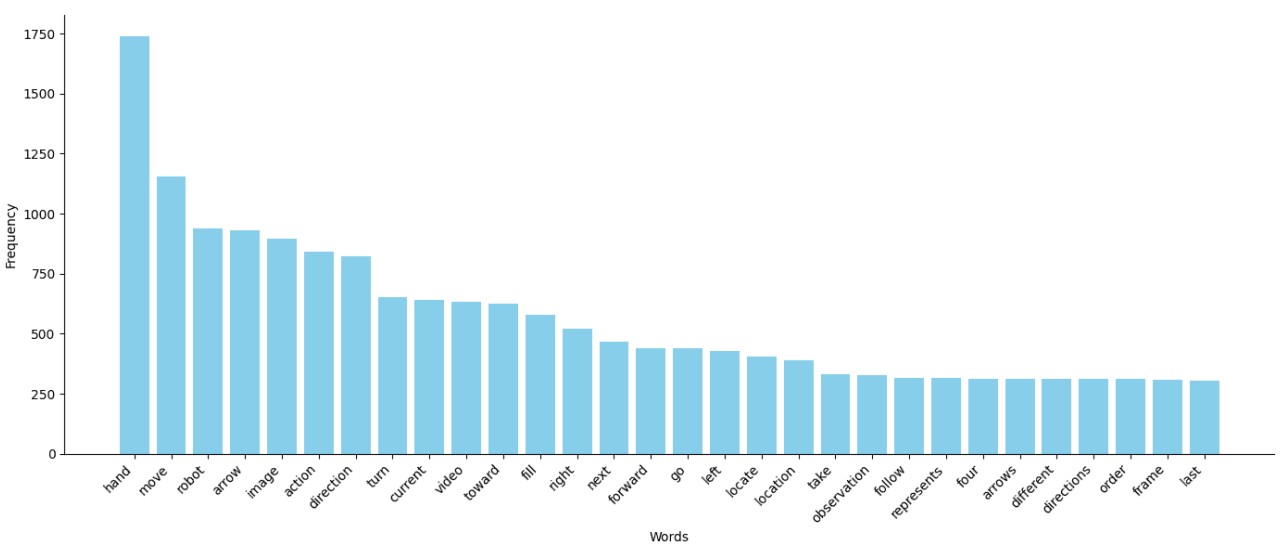

*Figure 28.* **BEAR keyword histogram.**

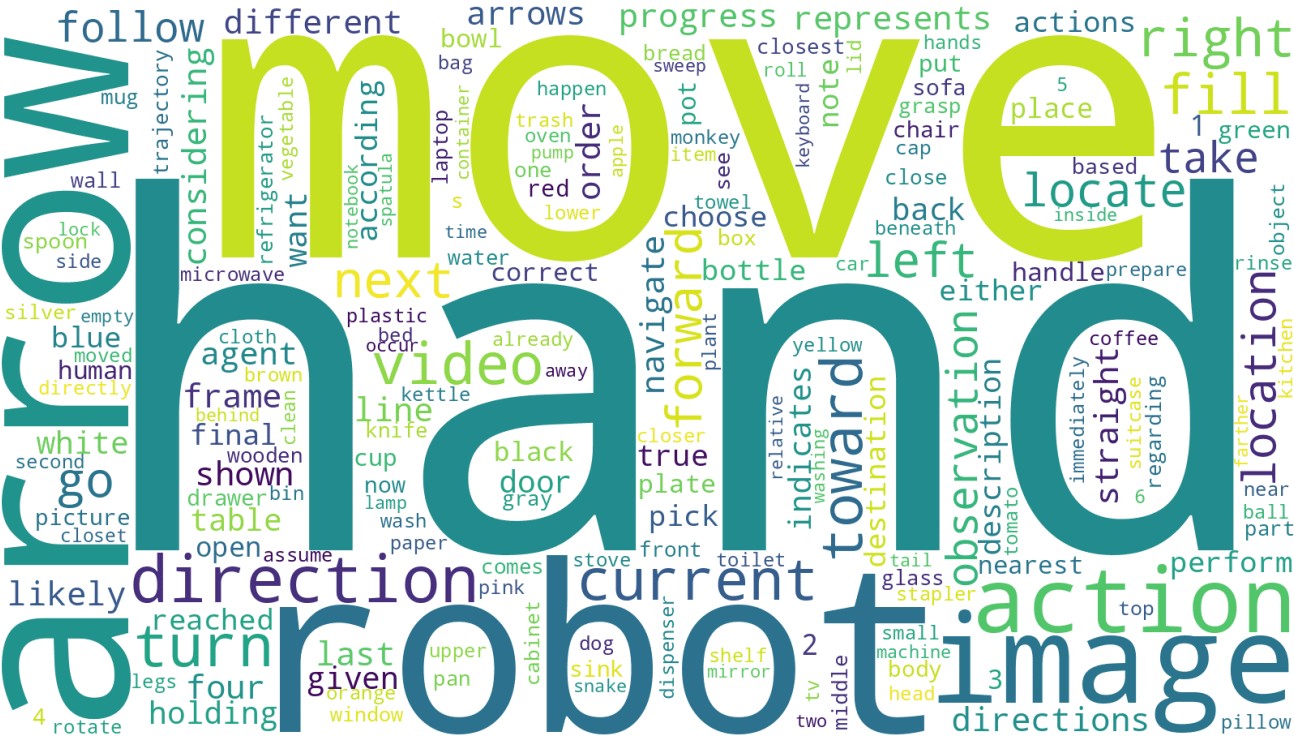

*Figure 29.* **BEAR word cloud.**

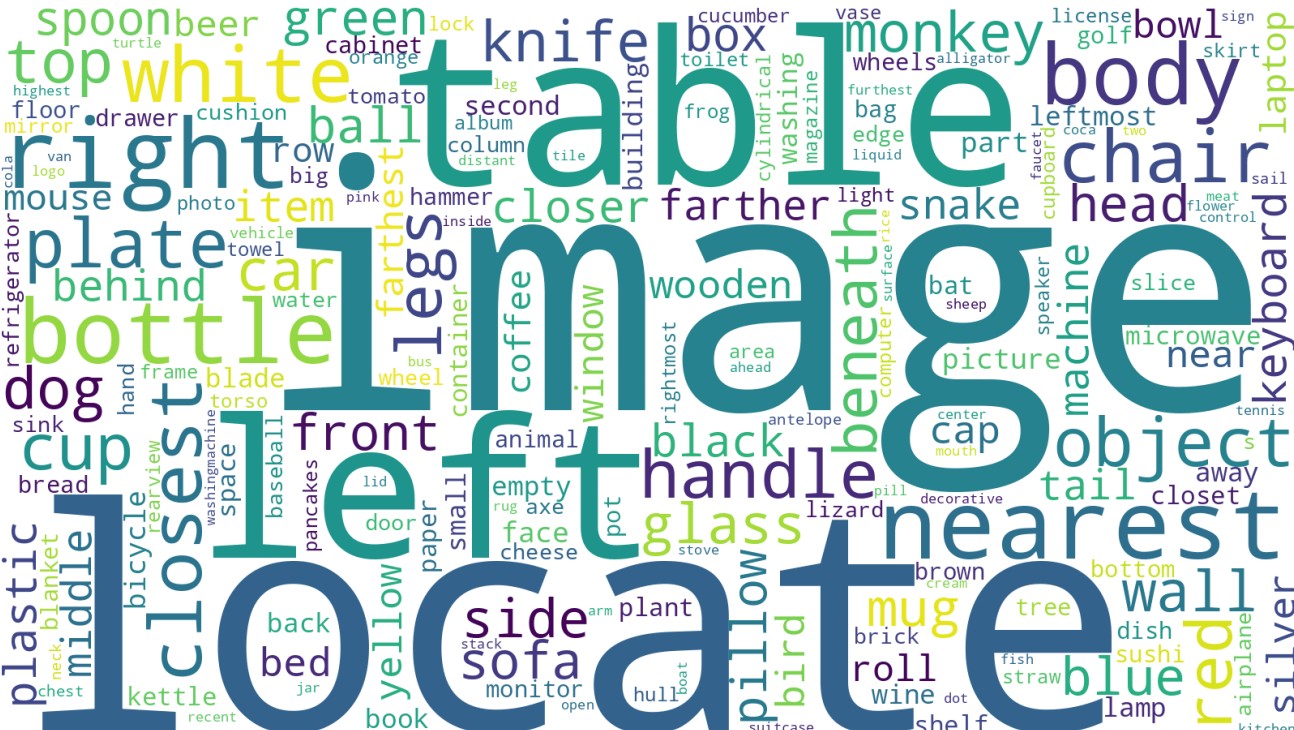

*Figure 30.* **Pointing word cloud.**

## D.2. Category-specific statistics

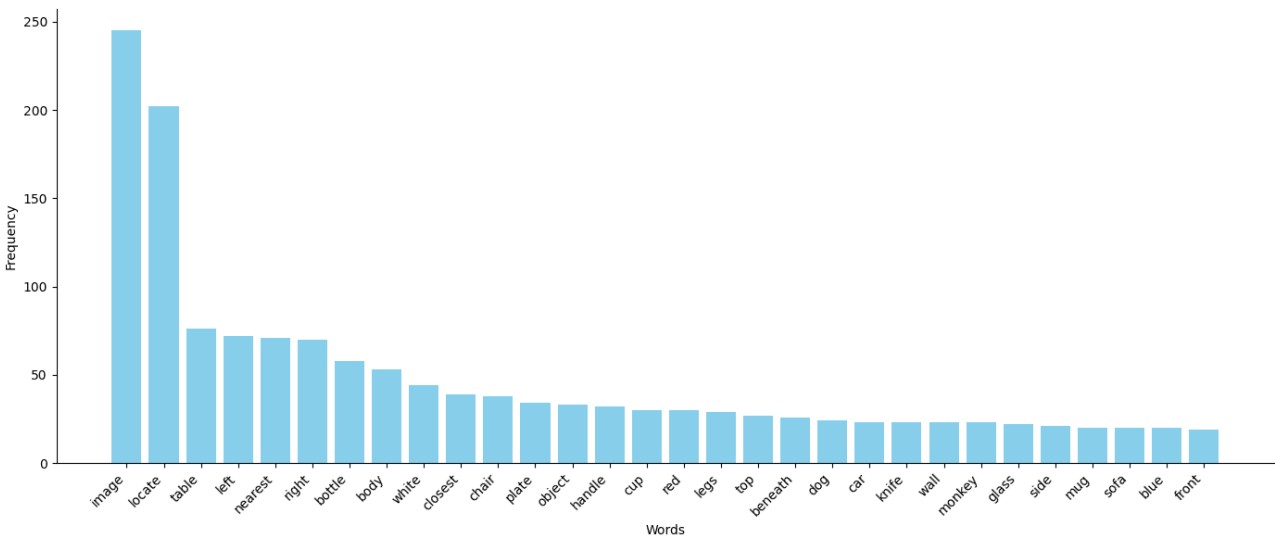

*Figure 31.* **Pointing keyword histogram.**

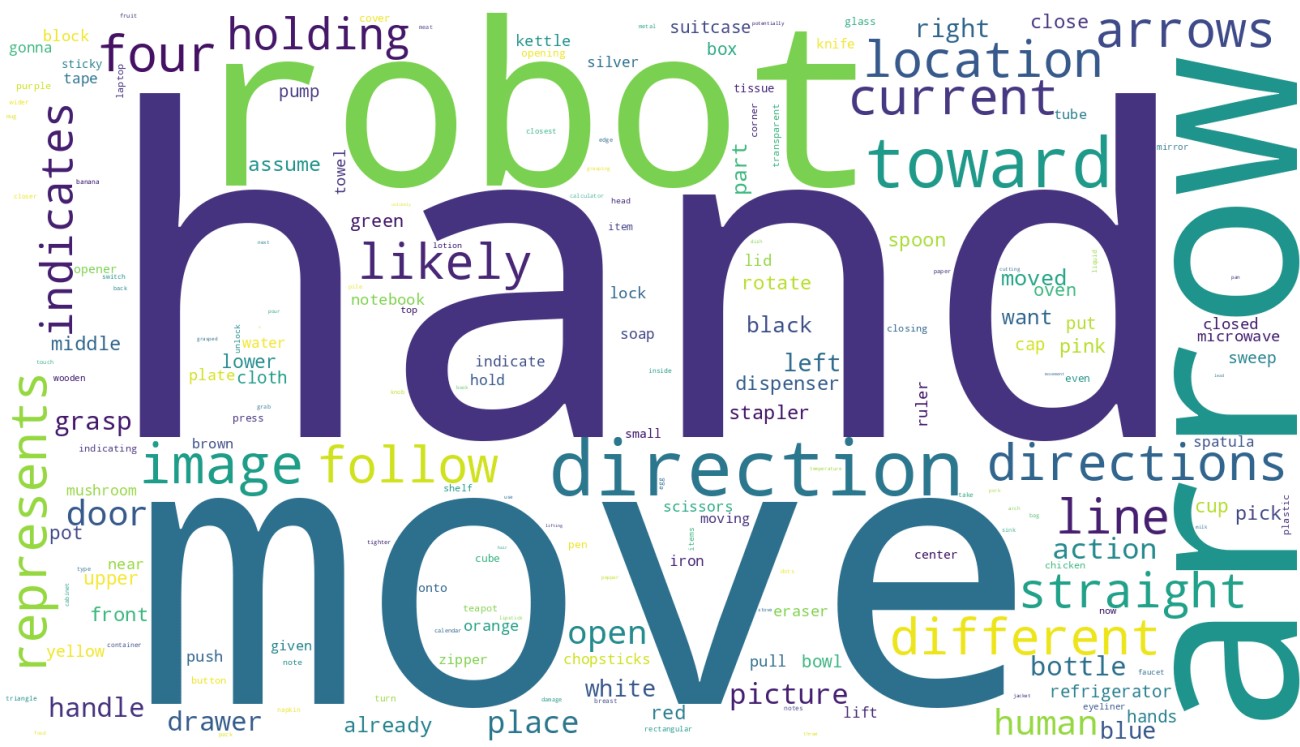

*Figure 32.* **Trajectory Reasoning word cloud.**

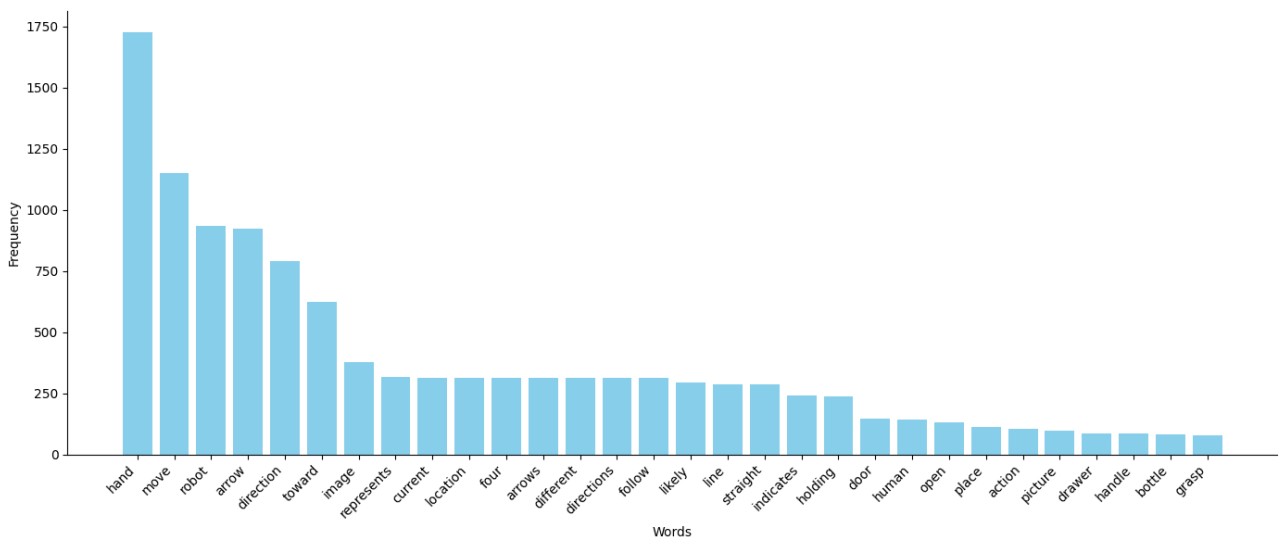

*Figure 33.* **Trajectory Reasoning keyword histogram.**

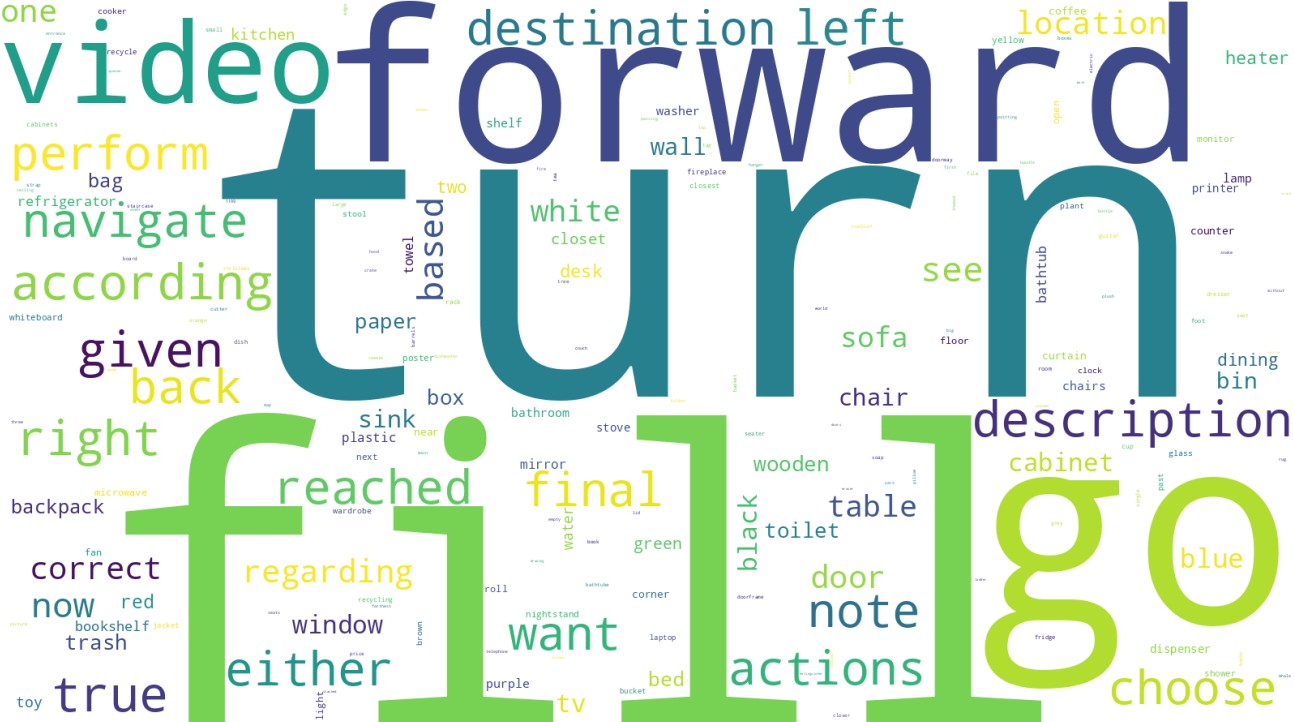

*Figure 34.* **Spatial Reasoning word cloud.**

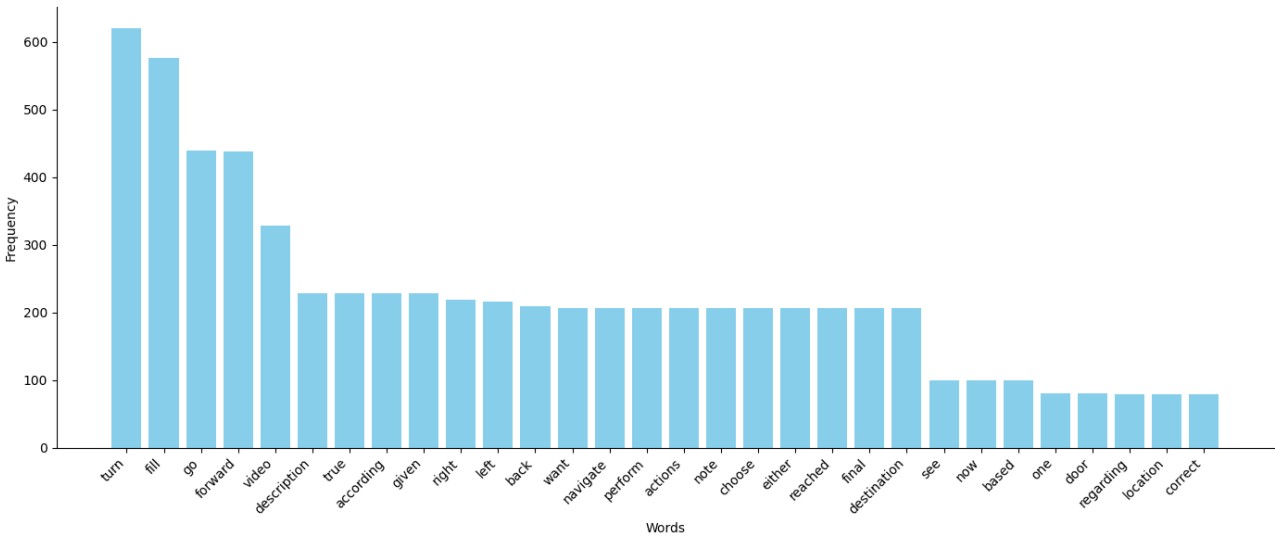

*Figure 35.* **Spatial Reasoning keyword histogram.**

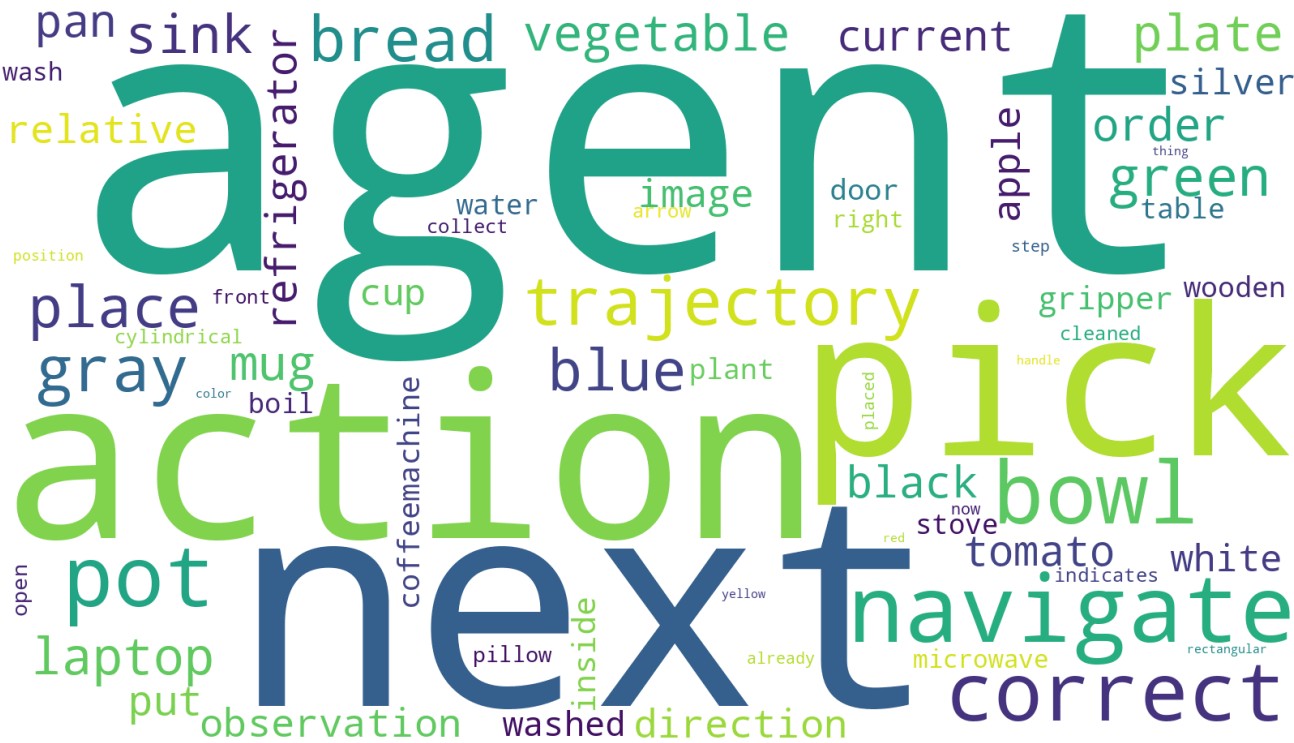

*Figure 36.* **Long-horizon word cloud.**

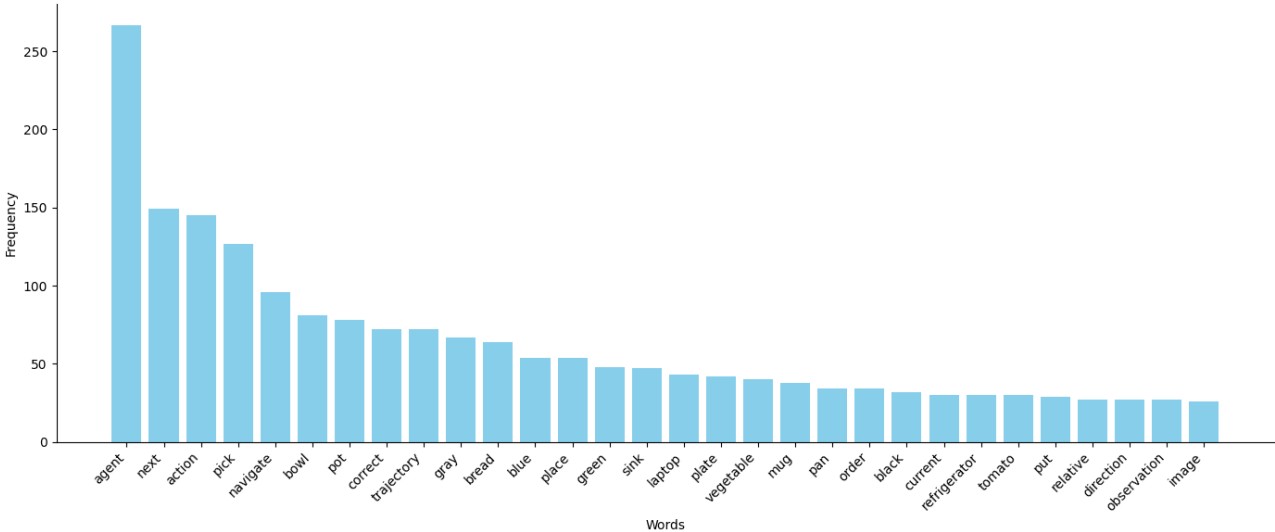

*Figure 37.* **Long-horizon keyword histogram.**

# E. Benchmark Curation Process

## E.1. Data Source Overview

Overall, our dataset is highly diverse, with each category composed of distinct types of data. The dataset is primarily built upon OpenImages (Kuznetsova et al., 2020), and is supplemented by the test sets from AGD20K (Luo et al., 2022) and PartImageNet (He et al., 2021). We also include BridgeData-v2 (Walke et al., 2023), a large-scale and diverse dataset of robotic manipulation behaviors that provides demonstration videos spanning a wide range of everyday manipulation tasks across various environments and object types. Images used for Human Hands Trajectory Reasoning are sourced from TASTE-Rob (Zhao et al., 2025). Additional video data is drawn from Egoplan-bench (Chen et al., 2023), Egoplan-bench2 (Qiu et al., 2024), EPIC-Kitchens (Damen et al., 2018), and Ego4D (Grauman et al., 2022). We also incorporate environments from ManipulaTHOR (Ehsani et al., 2021) and RoboTHOR (Deitke et al., 2020). All data acquisitions strictly comply with the licensing requirements outlined in Section B.

### E.1.1. POINTING AND 2D BOUNDING BOX PREDICTION

**Image Source.** The raw images for Pointing and 2D Bounding Box tasks are primarily sourced from the validation and test sets of OpenImages (Kuznetsova et al., 2020), supplemented by the test sets of AGD20K (Luo et al., 2022) and PartImageNet (He et al., 2021), resulting in over 30K images across 51 object categories. These categories are chosen to reflect common objects in embodied environments. We curate ground truth and questions via automated pipelines with human verification for accuracy, as detailed in Section E.2.1.

**Image Category.** We choose our data from each category of raw images to reflect objects commonly encountered in embodied environments. While the majority of categories represent indoor household or workspace items (e.g., microwave, spoon, toilet, soap dispenser), we also include a small number of outdoor relevant objects (e.g., car, traffic light, bench, Vehicle) to promote scene diversity and test model generalization beyond indoor settings.

*Note that the notion of image category here refers specifically to the image label used to download and filter source images, and is distinct from task types or question categories used in our benchmark.*

Object categories for General Object Pointing and Bounding Box Prediction, Spatial Relationship Pointing and Bounding Box Prediction are listed below:

- **Indoor**: Mailbox, Hot plate, Spoon, Drip coffee maker, Wallet, Kitchen & dining room table, Computer desk, Cup, Mixing bowl, Kitchen knife, Chopsticks, Bottle, Toaster, Microwave oven, Laptop, Computer keyboard, Computer mouse, Corded phone, Remote control, Tomato, Sofa bed, Filing cabinet, Door handle, Bottle opener, Soap dispenser, Coffee, Toilet paper, Pillow, Teapot, Measuring cup, Hammer, Wrench, Milk, Pancake, Doughnut, Bread, Spatula, Tap, Box, Zipper, Toilet, Facial tissue holder, Bottled Water

- **Outdoor**: Outdoor Umbrella Base, Bush, Bench, Mailbox, Car, Building, Path, Traffic light, Ball, Street Lamp, Sidewalk

For Semantic Part Pointing and 2D Bounding Box Prediction, we only download images which are likely to have meaningful part-level semantics, resulting in the following indoor and outdoor raw image categories:

- **Indoor**: Furniture, Kitchenware, Electronic appliance, Home appliance, Office supply, Container, Tableware, Personal item, Cleaning tool, Decoration, Pet supply, Food, Clothing, Medical item.

- **Outdoor**: Animal, Vehicle, Boat, Aircraft, Sports equipment, Tool, Outdoor equipment, Construction tool, Garden tool, Industrial machine, Plant.

### E.1.2. TRAJECTORY REASONING

**Image Source.** The images for Object Trajectory Reasoning come from articulated objects of OpenImages (Kuznetsova et al., 2020). The images from Gripper Trajectory Reasoning come from BridgeData-v2 (Walke et al., 2023), a large and diverse dataset of robotic manipulation behaviors, which provides demonstration videos covering a wide range of everyday manipulation tasks across multiple environments and object types. The images for Human Hands Trajectory Reasoning come from TASTE-Rob (Zhao et al., 2025), which is a large-scale dataset of 100K egocentric hand-object interaction videos.

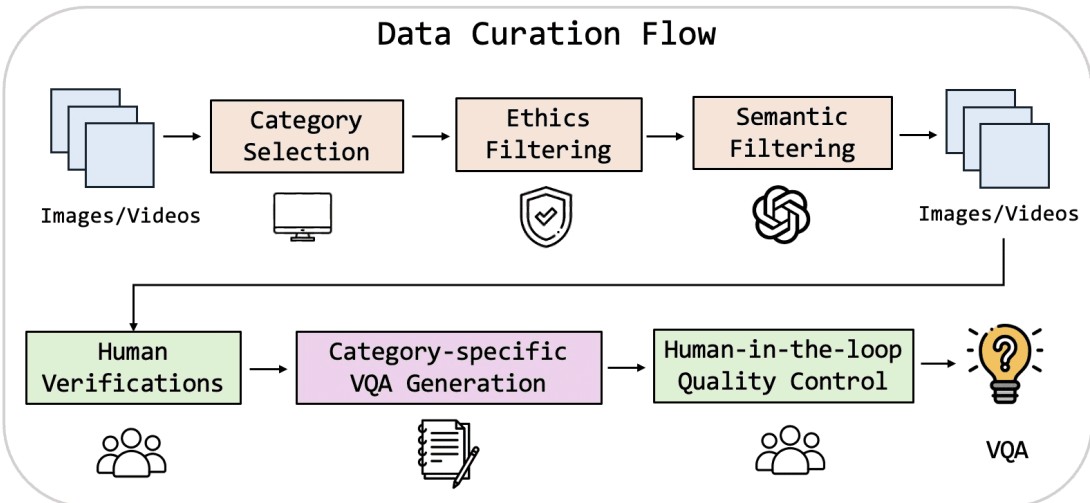

*Figure 38.* **Data Curation Work Flow.**

**Scene Category.** The scene categories covered in our benchmark include Kitchen, Bathroom, Living Room, Bedroom, Office, Study Room, Laboratory, Workspace, Dining Room, Storage Room, Closet, Hallway, Corridor, and Laundry Room.

### E.1.3. SPATIAL REASONING

**Data Source.** Videos in Spatial Reasoning category are sourced from the validation set and test set of ScanNet (Dai et al., 2017), ScanNet++(Yeshwanth et al., 2023), and ARKitScenes(Baruch et al., 2021). All of which are a large-scale indoor RGB-D dataset.

### E.1.4. TASK PLANNING

**Data Source.** Our source of Task Planning category covers from Egoplan-bench (Chen et al., 2023), Egoplan-bench2 (Qiu et al., 2024) and videos from EPIC-Kitchens (Damen et al., 2018) and Ego4D (Grauman et al., 2022).

### E.2. Data Filtering and VQA Generation

**General Principle.** As a general principle, we applied different data curation methods tailored to each category of the dataset combined with content safety filtering, human-in-the-loop filtering and human-in-the-loop correction, as shown in Figure 38. For every task category, we ensured balanced distributions across data instances, task types, and answer choices. In addition, each data point was subjected to at least two rounds of human verification to correct errors and eliminate low-quality or unreasonable samples. For details on distractor selection, as well as difficulty and quality control, refer to Section F.

### E.2.1. POINTING DATA CURATION.

**Download Source Images.** The pointing data curation process follows several key steps. We begin by collecting over 30K images sourced from OpenImages (Kuznetsova et al., 2020), supplemented by the test set of AGD20K (Luo et al., 2022) and PartImageNet (He et al., 2021), while ensuring a balanced distribution across both image categories and data sources. We then employ GPT-4o (Hurst et al., 2024) to classify these images into three categories, which is General Object Pointing, Spatial Relationship Pointing, and Semantic Part Pointing. For more details on the category balancing process, please refer to Appendix E.1.1.

**Ground Truth Generation.** For each selected image, we follow a structured data curation workflow. As illustrated in Figure 39, we first employ GPT-4o to perform object captioning, extracting object names from the image. These names are then passed to Grounded-Segment-Anything (Ren et al., 2024) to generate segmentation masks corresponding to the identified objects. For the *Semantic Part Pointing* category, we utilize Segment-Anything (Kirillov et al., 2023) to perform

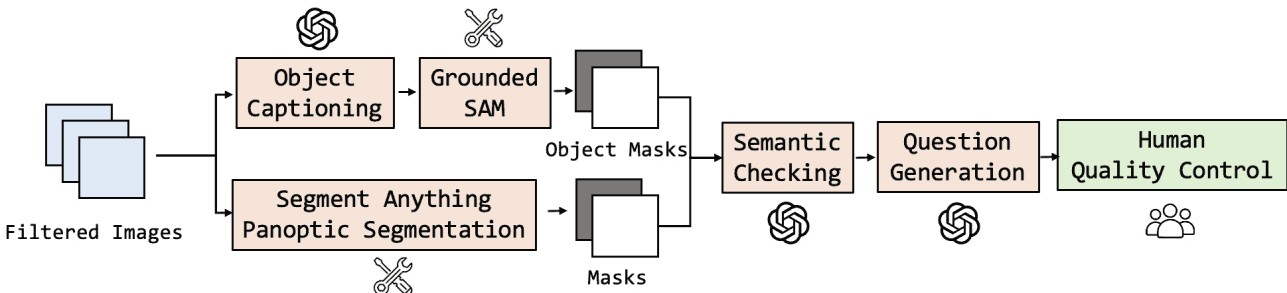

*Figure 39.* **Pointing Data Curation Work Flow.**

panoptic segmentation and generate all possible masks within the given scene.

After acquiring segmentation masks for each scene, we employ GPT-4o ([Hurst et al., 2024](#)) to perform semantic filtering, discarding masks that do not correspond to semantically meaningful regions. Subsequently, GPT-4o is used to generate natural language questions based on the location of each retained mask and the corresponding image. These questions are designed to contain rich descriptive cues, including color, shape, spatial position, and size, to reduce ambiguity and ensure accurate ground truth alignment. Examples include: 'Identify the blue and red dotted mug','Identify the nearest album on the table', and 'Identify the red car in the rightmost lane of the road'. To ensure the quality and safety of the generated content, two rounds of human verification are conducted by annotators, focusing on both quality assurance and ethical compliance.

**Evaluation Metrics.** The Multimodal Large Language Model is tasked with predicting a normalized $(x, y)$ coordinate, where $x \in (0, 1)$ and $y \in (0, 1)$, representing a pixel location within the image. A prediction is deemed correct if the indicated pixel lies within the ground truth mask; otherwise, it is considered incorrect.

### E.2.2. 2D BOUNDING BOX PREDICTION DATA CURATION

**Image Source.** The images used for the 2D Bounding Box Prediction task are selected similarly to those in the Pointing task. We filter out images containing multiple ground truth masks of the same category (e.g., two or more legs) to avoid ambiguity. The instruction is modified to prompt the Multimodal Large Language Model to output a bounding box instead of a pixel location.

**Ground Truth Generation.** The 2D bounding box ground truth is derived from segmentation masks curated through the pipeline illustrated in Figure [39](#). Specifically, each ground truth annotation is first represented as a binary mask, from which the corresponding bounding box is subsequently computed.

**Evaluation Metrics.** The Multimodal Large Language Model is tasked with predicting a normalized 2D bounding box, denoted as $(x_1, y_1, x_2, y_2)$, where $x \in (0, 1)$ and $y \in (0, 1)$. To evaluate performance, we compute the average Intersection over Union (IoU) between the ground truth bounding box $GT$ and the model-predicted bounding box $P$.

$$\text{IoU} = \frac{|GT \cap P|}{|GT \cup P|}$$

### E.2.3. TRAJECTORY REASONING DATA CURATION

**Image Source.** The images are sourced from TASTE-Rob ([Zhao et al., 2025](#)), OpenImages-v7 ([Kuznetsova et al., 2020](#)), and BridgeData V2 ([Walke et al., 2023](#)). A quality control process is applied to filter and retain only reasonable and relevant images for use.

**Question and Ground Truth Generation.** For Human Hand Trajectory Reasoning and Gripper Trajectory Reasoning, we utilize the demonstration trajectories and the annotated language instructions provided by TASTE-Rob ([Zhao et al., 2025](#)) and BridgeData V2 ([Walke et al., 2023](#)). We employ CoTracker3 ([Karaev et al.](#)) to generate ground truth trajectory arrows.

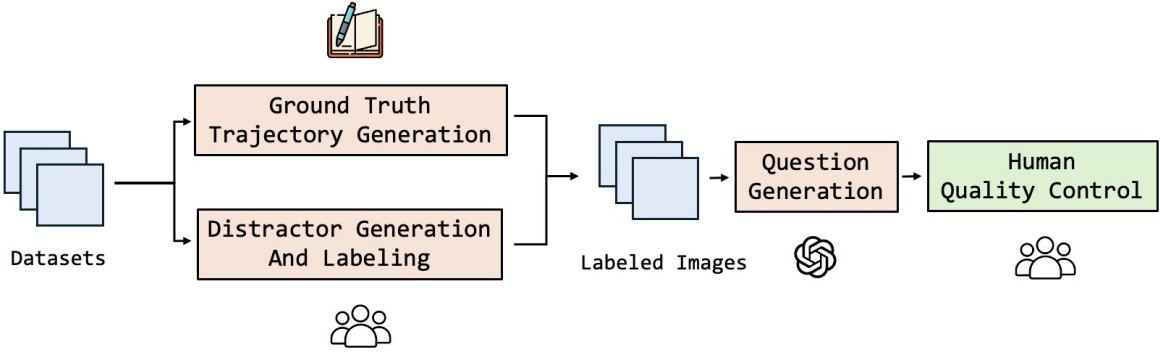

*Figure 40.* **Trajectory Reasoning Data Curation Work Flow.**

Additionally, we manually verify, correct, and annotate a portion of the data to ensure the accuracy and overall quality of the dataset. We provide a brief overview of our data curation pipeline in Figure 40.

**Distractor Generation.**    To generate ground truth and corresponding distractor options, we follow a carefully controlled selection process. Initially, multiple candidate trajectories are randomly sampled as distractors, prioritizing those directed toward alternative target objects. During data curation, we manually filter these trajectories to ensure semantic clarity and eliminate ambiguity. Additionally, we verify that all distractors are visually distinct from the background and do not conflict with the image's overall color composition.

### E.2.4. SPATIAL REASONING DATA CURATION

**Video Source.**    The video source comes from ScanNet (Dai et al., 2017), ScanNet++ (Yeshwanth et al., 2023), and ARKitScenes (Baruch et al., 2021).

**Question and Ground Truth Generation.**    We leverage camera annotations from selected videos in ScanNet (Dai et al., 2017), ScanNet++ (Yeshwanth et al., 2023), and ARKitScenes (Baruch et al., 2021), along with object labels and nearby object metadata, as inputs to GPT-4o (Hurst et al., 2024) for automatic question generation. To accommodate the diversity of question types in our benchmark, we design dedicated question-generation scripts tailored to each category. Due to the imperfect quality of automatically generated questions, we adopt a human-in-the-loop pipeline, where annotators manually revise both the questions and their corresponding answer choices. To further ensure the accuracy and consistency of the dataset, an additional round of quality verification is conducted by independent volunteers. We provide a brief overview of our spatial reasoning data curation pipeline in Figure 41.

**Distractor Generation.**    We adopt a multiple-choice question-answer format, where each question is accompanied by four options labeled A, B, C, and D. Among these, one represents the ground truth description. Option D is consistently set to either 'none of the above' or 'all of the above', which serves to evaluate the model's capacity for critical reasoning and its ability to assess the validity of multiple alternatives rather than relying solely on pattern recognition. The remaining distractor options are content-aligned with the correct answer and are crafted to be of comparable length, ensuring a fair comparison. Additionally, we apply strict control over the phrasing and specificity of the ground truth option to minimize ambiguity and bias.

### E.2.5. TASK PLANNING DATA CURATION

**Benchmark Source and Ground Truth Generation.**    The benchmark sources are derived from EgoPlan-Bench (Chen et al., 2023), EgoPlan-Bench2 (Qiu et al., 2024), as well as videos from EPIC-Kitchens (Damen et al., 2018) and Ego4D (Grauman et al., 2022). For the Next Action Planning category, we utilize the narration context provided in EgoPlan-Bench (Chen et al., 2023) and EgoPlan-Bench2 (Qiu et al., 2024), and augment the answer options with navigation-related content (e.g. 'walk to the sink'), as navigation is a fundamental component of embodied tasks. For the Task Process Reasoning category, we again leverage narration context from the two EgoPlan benchmarks and employ GPT-4o (Hurst

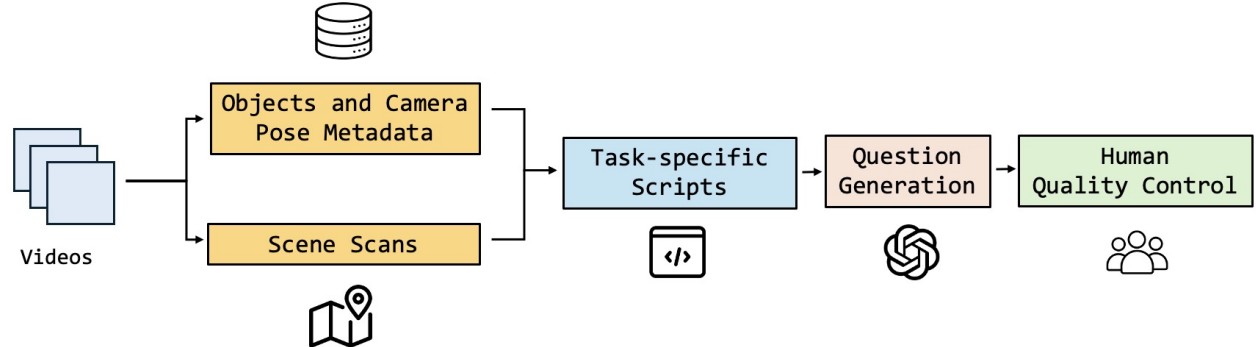

*Figure 41.* **Spatial Reasoning Data Curation Work Flow.**

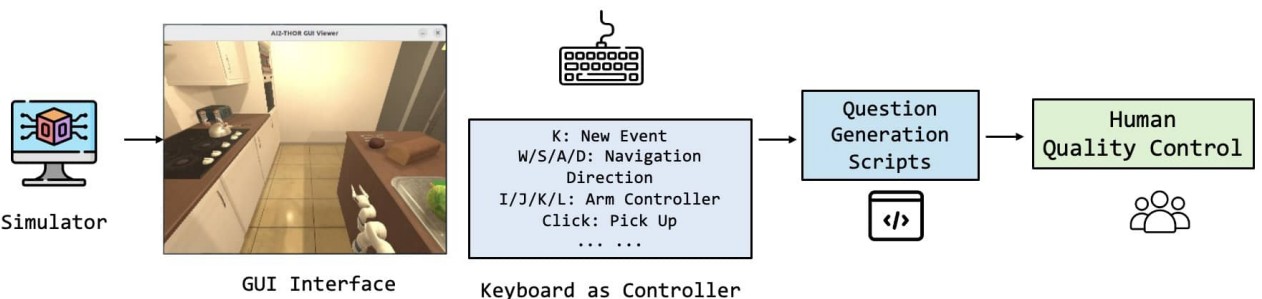

*Figure 42.* **Long-horizon Data Curation Work Flow.**

et al., 2024) to automatically generate questions targeting temporal reasoning. To ensure the accuracy and overall quality of the benchmark, all questions and answers undergo human verification by trained annotators.

### E.2.6. LONG-HORIZON TASK DATA CURATION

**Demo Source.** The demo is collected in the environment using ManipulaTHOR (Ehsani et al., 2021) and RoboTHOR (Deitke et al., 2020). We design a custom GUI for demo collection and data labeling, as shown in Figure 42.

**Question and Ground Truth Generation.** We design a graphical user interface (GUI) that enables human annotators to interact directly with the simulated environment using the keyboard as a controller. Custom scripts allow annotators to control navigation and actions via keyboard inputs, including the triggering of key events. Using this interface, we collected over 50 robot demonstrations, during which we record the state of each object in the simulation along with other critical environment information. After manual review, we retain 35 high-quality demonstrations based on criteria such as completeness, clarity of intent, and consistency. Based on these curated interactions, we develop automated scripts to generate question–answer pairs. To ensure data quality and semantic validity, we further recruit human annotators to manually annotate and verify the generated data.

## F. Benchmark Distractor, Quality and Difficulty Control

A central principle guiding the design of our benchmark is the deliberate control of task difficulty, enabling fine-grained evaluation of model capabilities across varying levels of reasoning complexity. To achieve this, we design category-specific difficulty control strategies, ensure the validity and informativeness of distractor options, and implement a multi-stage quality assurance process.

More specific, our general for question generation is listed as follows:

- All questions must contain one or more images.

- All questions should be written in English.

- All questions are clearly categorized.

- All fine-grained categories is under difficulty control.

- All fine-grained categories goes through at least two round of human verification process.

- All questions are clearly categorized.

- All questions are uniformly formatted to ensure clarity and consistency across the benchmark.

- All questions have very clear ground truth.

- All distractors are contextually reasonable and intentionally designed to function as meaningful and challenging alternatives.

F.0.1. DIFFICULTY CONTROL

For different benchmark categories, we adopt category-specific strategies for difficulty control, which is listed as follows:

- **Pointing and Bounding Box Category**: We control task difficulty by regulating the size of the ground truth masks. Specifically, we sort all candidate masks by area and remove those that are either too small or too large, as such extremes may lead to questions that are overly difficult or trivial. From the remaining masks, we perform uniform sampling to ensure a balanced distribution of difficulty levels and object categories.

- **Trajectory Reasoning Category**: Difficulty is controlled through a combination of manual design and heuristic variation. Human annotators are instructed to generate both ground truth answers and distractor options with varying degrees of ambiguity. Specifically, we manipulate factors such as:
  - Trajectory proximity: Harder samples place distractor arrows spatially closer to the ground truth, increasing visual confusion.
  - Object similarity: Distractors are selected to point toward objects that are visually or semantically similar to the target object (e.g., bottles of similar shape or color).
  - Language ambiguity: We vary the clarity of the question (e.g., specifying 'the bottle' vs. 'the bottle with white rectangular lid ') to test the model's ability to resolve referential expressions.

- **Spatial Reasoning Category**: we adopt task-specific strategies to control the difficulty of spatial reasoning questions:
  - Object Localization: Difficulty is determined by the spatial distance and visibility between the target object and the anchor. Easy examples feature clearly visible objects near salient anchors, while hard examples involve occluded objects or non-trivial spatial relations.
  - Path Planning: Difficulty is defined by the complexity of navigation steps. Easy samples require only a single forward movement, whereas hard samples involve multiple turns, changes in direction, or room transitions.
  - Relative Direction: We control difficulty by manipulating the ambiguity of direction options and the need for temporal reasoning.

- **Task Planning Category**: We adopt task-specific strategies to control the difficulty of task planning questions:
  - Task Process Reasoning: Difficulty is governed by the complexity of temporal dependencies between actions. Easy samples involve short sequences with clearly separable events. Hard samples contain longer temporal chains, event ambiguity, or distractor choices that are temporally plausible but incorrect (e.g., 'wipe the table' vs. 'dry the plate').
  - Next Action Prediction: Easy instances correspond to goals that can be completed with a single or short sequence of atomic actions (e.g., "pick up the cup and place it on the table"), requiring minimal reasoning. In contrast, harder instances involve longer-horizon goals that demand multiple reasoning steps.

```
{
  "idx": 302,
  "image": "",
  "video": "video/video_302.mp4",
  "question": "Which description of following about the water dispenser is true according to the
video given?",
  "options": {
    "A": "there is a blue water bottle on the top of the water dispenser",
    "B": "there are no water bottles on the top of the water dispenser",
    "C": "there is a fully filled blue water bottle on the top of the water dispenser",
    "D": "None of above"
  },
  "gt": "A",
  "mask": "",
  "bbox": "",
  "category": "object localization",
}
```

*Figure 43.* **Unified Benchmark Data Format.** All our data adheres to a consistent format across tasks. For example, in an object localization instance, fields that are not applicable are left blank.

### F.0.2. DISTRACTOR CONTROL

Effective distractor control is essential in multiple-choice question design to assess a model's true reasoning ability rather than its reliance on superficial cues. Distractors should be plausible but incorrect, semantically coherent, and contextually relevant to the question. They must match the correct answer in style, length, and structure, avoiding easy elimination through obvious differences. All options should be mutually exclusive and collectively comprehensive. A well-designed distractor set compels the model to engage in genuine reasoning. Moreover, distractor strategies should be tailored to the specific task type, whether temporal, spatial, visual, or goal-directed, to effectively probe the intended reasoning skill.

More specifically, our distractor design follows these principles:

- **Inclusion of 'None of the above' as Option D.** The fourth option is consistently set as 'None of the above' to promote deeper reasoning. A model must correctly reject all three distractors and affirm that none is correct. It ensures the model isn't simply selecting the most similar option but performing holistic understanding and elimination.

- **Semantic and Structural Consistency.** The other distractor options are carefully crafted to match the correct answer in semantic content, sentence structure, and length. This prevents models from exploiting surface-level cues and encourages reliance on true comprehension.

- **Category-Specific Distractor Strategies.** We adopt tailored distractor generation methods for different question categories. Each strategy is designed to meaningfully challenge the reasoning ability most relevant to the task.
  - Trajectory Reasoning Category: Distractors are designed by assigning arrows that point in incorrect directions—toward irrelevant or misleading objects while maintaining similar visual and spatial plausibility.
  - Spatial Reasoning Category: Distractors are crafted as misleading spatial descriptions, such as references to other objects or plausible yet incorrect localization cues and movement paths.
  - Planning Category: Distractors include semantically similar but incorrect actions, often with misleading or incorrect temporal dependencies to challenge causal and sequential reasoning.

### F.0.3. QUALITY CONTROL

To ensure the reliability and validity of our benchmark, we implement multiple layers of quality control:

- **Format Unification.** To ensure consistency and facilitate seamless evaluation, we unify the input and output formats across all tasks and modalities in our benchmark. This standardization simplifies data preprocessing, enables reusable

evaluation pipelines, and ensures that models are tested under comparable conditions. Moreover, a unified format makes it easier for researchers to adopt, reproduce, and extend the benchmark, reducing ambiguity and implementation overhead. We illustrate this with an example from the object localization category, as shown in Figure 43.

- **Ethics Checking.** We filter out any content that may violate ethical standards or pose risks, ensuring that all data is safe for both human and machine use.

- **Quality Verification.** We assess each instance for linguistic clarity, logical soundness, and relevance to the target reasoning skill.

- **Manual Filtering.** Two final rounds of human-in-the-loop verification ensures that only high-quality, task-aligned examples remain, removing any residual noise or inconsistency.

These steps collectively help eliminate annotation artifacts, ensure fairness, and maintain the integrity of the reasoning challenges across categories.

# G. Experiment

### G.0.1. MODEL NAME AND INFERENCE SET UP

**Model Name and Parameters.** We evaluate both proprietary and open-source models on our benchmark. The model names and corresponding inference settings are summarized in Table 3. For models whose parameters are not explicitly specified, we adopt their default configurations as provided by their respective APIs or official implementations. We provide detailed descriptions of the models as follows.

- **DeepSeek-VL** (Lu et al., 2024) is a real-world vision-language model proposed by DeepSeek-AI. It integrates a hybrid vision encoder for efficient high-resolution image processing (1024×1024) and adopts a staged VL pretraining strategy that gradually balances textual and visual modalities. DeepSeek-VL achieves strong performance across a wide range of visual-language tasks while maintaining robustness on language-centric benchmarks.

- **Molmo** (Deitke et al., 2025)Molmo series of multimodal vision–language models developed by the Allen Institute for AI (AI2). It is built on the Qwen2-7B architecture and uses OpenAI's CLIP model as the vision encoder, enabling it to process both image and text inputs simultaneously.

- **InternVL 2.0** (Chen et al., 2024) is a next-generation vision-language model developed by OpenGVLab. It features a strong visual encoder that supports high-resolution image understanding (up to 896×896) and a vision-language training paradigm optimized for both alignment and generation. By introducing high-quality pretraining data and fine-grained vision-language alignment techniques, InternVL 2.0 demonstrates superior performance on various visual reasoning and grounding benchmarks.

- **InternVL 3.0** (Zhu et al., 2025) is a next-generation multimodal model proposed by OpenGVLab. It is trained from scratch with a unified vision-language pretraining paradigm. Unlike models that adapt text-only LLMs, it jointly learns from both multimodal and text data. Key innovations include Variable Visual Position Encoding (V2PE), Supervised Fine-Tuning (SFT), and Mixed Preference Optimization (MPO). InternVL 3-78B sets a new open-source SOTA on MMMU, rivaling GPT-4o and Claude 3.5 Sonnet.

- **LLaVa-NeXT-Interleave** (Li et al., 2024b) is a multimodal large language model proposed by ByteDance. It expands visual instruction tuning beyond single-image settings by supporting multi-image, multi-frame, multi-view, and multi-patch inputs. It introduces the M4-Instruct dataset and the LLaVA-Interleave Bench to evaluate multi-modal reasoning. The model achieves SOTA on multi-image, video, and 3D tasks while preserving single-image performance and demonstrating emergent cross-modal capabilities.

- **LLaVa-NeXT-Llama3** (Li et al., 2024a) is a next-generation open-source large multimodal model built upon Meta's LLaMA3 (Dubey et al., 2024), integrating high-resolution vision encoders with strong language backbones. It leverages interleaved visual instruction tuning to handle complex multi-image inputs and achieves competitive performance across a broad range of visual-language tasks. The model is notable for its strong instruction-following abilities and efficient training pipeline.

- **Qwen2.5-VL** (Bai et al., 2025) is the latest flagship vision-language model from Alibaba's Qwen series, showcasing strong capabilities in object localization, document parsing, and long-video understanding. It introduces dynamic resolution processing and absolute time encoding for native perception of spatial and temporal cues. Built with a dynamic-resolution ViT and Window Attention, Qwen2.5-VL achieves high efficiency while maintaining fine-grained detail. The 72B version particularly excels in document and diagram comprehension, while preserving strong language abilities.

- **Claude-3.7-Sonnet** (Anthropic, 2024), which is released by Anthropic in February 2025, is a multimodal large language model that demonstrates improved performance over its predecessor in natural language understanding, code generation, and multimodal reasoning. It is well-suited for complex tasks and high-quality dialogue generation.

- **Claude-4-Sonnet** (Anthropic, 2025), introduced in May 2025, further advances the model's capabilities in multi-turn dialogue consistency, visual reasoning, and tool use. Its overall performance is comparable to state-of-the-art models such as GPT-4o and Gemini 2.5 Pro.

- **Gemini-2.0-Flash** (Team, 2024) released by Google DeepMind in late 2024, is a lightweight and efficient variant of the Gemini multimodal series. Designed for real-time applications, Gemini 2.0 Flash focuses on fast inference while retaining strong performance in core tasks such as visual question answering, image captioning, and basic reasoning.

- **Gemini-2.5-Flash** (Comanici et al., 2025) builds upon its predecessor with improved architectural refinements and a broader training corpus. It enhances the model's capabilities in visual grounding, spatial reasoning, and multilingual understanding, while maintaining low-latency response suitable for deployment in interactive systems.

- **Gemini-2.5-Pro** (Comanici et al., 2025) represents the most powerful version of the 2.5 series, offering state-of-the-art performance across a wide range of multimodal benchmarks. With support for long-context understanding, fine-grained visual localization, and complex task execution, Gemini 2.5 Pro competes closely with leading models such as GPT-4o and Claude 4 in both accuracy and versatility.

- **GPT-4o** (Hurst et al., 2024) is OpenAI's flagship omnimodal model, capable of processing and reasoning over text, images, audio, and video inputs within a unified architecture. Unlike its predecessors, GPT-4o achieves native multimodal understanding without relying on modality-specific adapters, enabling seamless integration across vision, language, and speech. The model supports real-time audio interactions with low latency and exhibits enhanced spatial, temporal, and perceptual grounding. With strong performance on visual question answering, object localization, audio comprehension, and long-context reasoning, GPT-4o sets a new benchmark for general-purpose multimodal intelligence and serves as the backbone for OpenAI's latest ChatGPT systems.

- **GPT-5** (OpenAI, 2025a) is officially released by OpenAI on August 7, 2025. It is most advanced AI system of OpenAI, achieving state-of-the-art performance in coding, math, writing, health, and visual perception. As a unified model, it adapts between fast responses and extended reasoning to deliver expert-level answers, with a Pro version offering deeper reasoning capabilities.

- **GPT-o3** (OpenAI, 2025b) represents the most advanced reasoning model in OpenAI's o-series, designed for deep, step-by-step problem-solving across coding, math, science, and visual perception. Released on April 16, 2025, GPT-o3 introduces multimodal 'image-aware' chain-of-thought reasoning and integrated tool usage such as web browsing, file analysis, and image manipulation

**Inference Format.** In Table 3, *Merged* means that for the video and interleaved question answer pairs, we merge the uniformly sampled frames from the video into one image and then send it as the input of the multimodal large language model. *Sequential* means that we sequentially send frames and prompt accordingly.

*Table 3.* **Inference parameters for models. All other parameters not specified here use the default model configurations.**

| Model | Format | Inference Setup |
|---|---|---|
| DeepSeek-VL-7B (Lu et al., 2024) | Merged | dtype = torch.bfloat16, max_new_tokens = 512, do_sample = False, temperature=0, seed=42 |
| Molmo-7B-D-0924 (Deitke et al., 2025) | Merged | dtype = torch.bfloat16, max_new_tokens = 512, do_sample = False, temperature=0, seed=42 |
| InternVL2-4B (Chen et al., 2024) | Merged | dtype = torch.bfloat16, max_new_tokens = 512, do_sample = False, temperature=0, seed=42, top_p=None, num_beams=1) |
| InternVL2-8B (Chen et al., 2024) | Merged | dtype = torch.bfloat16, max_new_tokens = 512, do_sample = False, temperature=0, seed=42, top_p=None, num_beams=1) |
| InternVL2-26B (Chen et al., 2024) | Merged | dtype = torch.bfloat16, max_new_tokens = 512, do_sample = False, temperature=0, seed=42, top_p=None, num_beams=1) |
| InternVL2-40B (Chen et al., 2024) | Merged | dtype = torch.bfloat16, max_new_tokens = 512, do_sample = False, temperature=0, seed=42, top_p=None, num_beams=1) |
| InternVL3-8B (Zhu et al., 2025) | Merged | dtype = torch.bfloat16, max_new_tokens = 512, do_sample = False, temperature=0, seed=42, top_p=None, num_beams=1) |
| InternVL3-14B (Zhu et al., 2025) | Merged | dtype = torch.bfloat16, max_new_tokens = 512, do_sample = False, temperature=0, seed=42, top_p=None, num_beams=1) |
| LLava-NeXT-Interleave-7B (Li et al., 2024b) | Merged | dtype = torch.bfloat16, max_new_tokens = 2048, do_sample = False, seed=42, temperature=0, top_p=None, num_beams=1 |
| LLaVa-NeXT-Llama3-8B (Li et al., 2024a) | Merged | dtype = torch.bfloat16, max_new_tokens = 2048, do_sample = False, seed=42, temperature=0, top_p=None, num_beams=1 |
| Qwen2.5-VL-7B-Instruct (Bai et al., 2025) | Merged | dtype = torch.bfloat16, max_new_tokens = 512, do_sample = False, max_num = 1, seed=42 |
| Qwen2.5-VL-32B-Instruct (Bai et al., 2025) | Merged | dtype = torch.bfloat16, max_new_tokens = 512, do_sample = False, max_num = 1, seed=42 |
| Claude-3.7-Sonnet-20250219 (Anthropic, 2024) | Sequential | - |
| Claude-4-Sonnet-20250514 (Anthropic, 2025) | Sequential | - |
| Gemini-2.0-Flash (Team, 2024) | Sequential | - |
| Gemini-2.5-Flash (Comanici et al., 2025) | Sequential | - |
| Gemini-2.5-Pro (Comanici et al., 2025) | Sequential | - |
| GPT-4o (Hurst et al., 2024) | Sequential | - |
| GPT-5 (OpenAI, 2025a) | Sequential | - |
| GPT-o3 (OpenAI, 2025b) | Sequential | - |

G.0.2. BENCHMARK EVALUATION RESULTS

**General Principle**

- We use different evaluation metrics for different categories. For *Bounding Box* category, we report IoU (Intersection over Union) for evaluation, while for other categories, we report success rate(%). Details of the evaluation metrics are provided below in each category.

- All models are evaluated using the direct format, where the model is prompted to directly output the final answer.

- We conduct human studies on BEAR-mini, a subset of our benchmark constructed by randomly sampling 40 questions from each skill. The detailed procedure for the human evaluation is described below.

- In order to calculate the overall average performance of MLLMs on 6 categories, we multiply *Bounding Box* by 100 and then take average of 6 categories.

**Human Studies.** To establish a human performance baseline, we conduct user studies on BEAR-mini, a subset of our benchmark created by randomly sampling 40 questions from each skill. Five adult participants, all of whom provided informed consent, took part in the study. They were briefed on the task goals, data usage, and their right to withdraw at any time. No personally identifiable information (PII) was collected. The study was carried out solely for research purposes in compliance with institutional human subjects guidelines.

**Pointing.** The evaluation results for the Pointing category are reported in Table 4. The evaluation metric is success rate, defined as the average over all questions, where a score of 1 is assigned if the predicted point is correct and 0 otherwise. The MLLM is tasked with predicting a normalized $(x, y)$ coordinate on a single image, where $x \in (0, 1)$ and $y \in (0, 1)$, representing a pixel location within the image. A prediction is deemed correct if the indicated pixel lies within the ground truth mask; otherwise, it is considered incorrect.

**Bounding Box.** The evaluation results for the Bounding Box category are reported in Table 4. The evaluation metric is IoU, short for Intersection over Union. We report the average IoU on all questions. The MLLM is tasked with predicting a normalized 2D bounding box, denoted as $(x_1, y_1, x_2, y_2)$, where $x \in (0, 1)$ and $y \in (0, 1)$. To evaluate performance, we compute the IoU between the ground truth bounding box $GT$ and the model-predicted bounding box $P$.

$$\text{IoU} = \frac{|GT \cap P|}{|GT \cup P|}$$

**Trajectory Reasoning.** The evaluation results for Trajectory Reasoning category is reported in Table 4. The evaluation metric is success rate (%). A question is considered correct if the model selects the ground-truth option, otherwise it is counted as incorrect.

**Spatial Reasoning.** The evaluation results for Spatial Reasoning category is reported in Table 4. The evaluation metric is success rate (%). A question is considered correct if the model selects the ground-truth option, otherwise it is counted as incorrect.

**Task Planning.** The evaluation results for Task Planning category is reported in Table 4. The evaluation metric is success rate (%). A question is considered correct if the model selects the ground-truth option, otherwise it is counted as incorrect.

**Long-horizon Reasoning.** The evaluation results for the long-horizon category are reported in Table 4. The evaluation metric is success rate (%) over all 35 episodes. Each episode records an agent performing a common task in simulation, which we manually decompose into a sequence of necessary decision-making steps. A question is considered correct if the model selects the ground-truth option, otherwise it is counted as incorrect. An episode is deemed successful only if the MLLM answers all questions within that episode correctly.

*Table 4.* **Evaluation results on BEAR.** We report performance of 20 MLLMs. GEN = *General Object (Pointing/Box)*; SPA = *Spatial Object (Pointing/Box)*; PRT = *Semantic Part (Pointing/Box)*; PRG = *Task Process Reasoning*; PRD = *Next Action Prediction*; GPR = *Gripper Trajectory Reasoning*; HND = *Human Hand Trajectory Reasoning*; OBJ = *Object Trajectory Reasoning*; LOC = *Object Localization*; PTH = *Path Planning*; DIR = *Relative Direction*. *Bounding Box* scores are scaled by 100 when computing overall average. We also report *Random Choice* for multi-choice questions.

| | Format | Pointing | | | | Bounding Box | | | | Task Planning | | |
|---|---|---|---|---|---|---|---|---|---|---|---|---|
| | | GEN | SPA | PRT | Avg | GEN | SRA | PRT | Avg | PRG | PRD | Avg |
| Random Choice | | - | - | - | - | - | - | - | - | 25 | 25 | 25 |
| Human | | - | - | - | - | - | - | - | - | - | - | - |
| *Open-source Models* | | | | | | | | | | | | |
| DeepSeek-VL-7B (Lu et al., 2024) | merged | 14.12 | 8.50 | 9.24 | 10.62 | 0.276 | 0.160 | 0.231 | 0.222 | 37.67 | 27.33 | 32.50 |
| Molmo-7B-D-0924 (Deitke et al., 2025) | merged | 23.53 | 19.28 | 25.48 | 22.76 | 0.109 | 0.082 | 0.109 | 0.100 | 37.67 | 31.00 | 34.34 |
| InternVL2-4B (Chen et al., 2024) | merged | 18.53 | 10.78 | 12.42 | 13.91 | 0.117 | 0.082 | 0.107 | 0.102 | 37.33 | 32.33 | 34.83 |
| InternVL2-8B (Chen et al., 2024) | merged | 21.18 | 21.90 | 21.97 | 21.68 | 0.294 | 0.194 | 0.179 | 0.222 | **44.00** | 31.67 | 37.84 |
| InternVL2-26B (Chen et al., 2024) | merged | 21.18 | 15.36 | 18.79 | 18.44 | 0.201 | 0.202 | 0.147 | 0.183 | 41.33 | 34.33 | 37.83 |
| InternVL2-40B (Chen et al., 2024) | merged | 23.24 | 21.24 | 22.29 | 22.25 | 0.329 | 0.269 | 0.268 | 0.289 | 40.00 | 33.67 | 36.84 |
| InternVL3-8B (Zhu et al., 2025) | merged | **52.65** | **42.48** | **43.95** | **46.36** | **0.369** | **0.275** | **0.297** | **0.314** | 43.00 | 33.67 | 38.34 |
| InternVL3-14B (Zhu et al., 2025) | merged | 37.94 | 27.78 | 32.80 | 32.84 | 0.304 | 0.258 | 0.276 | 0.279 | 41.00 | 33.00 | 37.00 |
| LLava-NeXT-Interleave-7B (Li et al., 2024b) | merged | 6.47 | 3.59 | 2.55 | 4.20 | 0.000 | 0.000 | 0.000 | 0.000 | 37.33 | 26.00 | 31.67 |
| LLaVa-NeXT-Llama3-8B (Li et al., 2024a) | merged | 2.94 | 1.31 | 0.96 | 1.73 | 0.320 | 0.246 | 0.205 | 0.257 | 36.67 | 29.67 | 33.17 |
| Qwen2.5-VL-7B-Instruct (Bai et al., 2025) | merged | 6.18 | 1.63 | 0.96 | 2.92 | 0.007 | 0.003 | 0.009 | 0.007 | 40.67 | 32.33 | 36.50 |
| Qwen2.5-VL-32B-Instruct (Bai et al., 2025) | merged | 27.35 | 27.78 | 42.68 | 32.60 | 0.020 | 0.018 | 0.017 | 0.018 | 42.67 | **42.33** | **42.50** |
| *Proprietary Models* | | | | | | | | | | | | |
| Claude-3.7-Sonnet (Anthropic, 2024) | sequential | 47.94 | 36.27 | 37.58 | 40.60 | 0.195 | 0.132 | 0.187 | 0.171 | 32.67 | 44.33 | 38.50 |
| Claude-4-Sonnet (Anthropic, 2025) | sequential | 39.12 | 40.86 | 45.54 | 41.84 | 0.221 | 0.173 | 0.197 | 0.197 | 44.00 | 37.67 | 40.84 |
| Gemini-2.0-Flash (Team, 2024) | sequential | 51.76 | 34.97 | 40.13 | 42.29 | 0.270 | 0.167 | 0.224 | 0.220 | 38.67 | 40.00 | 39.34 |
| Gemini-2.5-Flash (Comanici et al., 2025) | sequential | 46.76 | 33.33 | 39.49 | 39.86 | 0.183 | 0.145 | 0.156 | 0.161 | 48.33 | 43.67 | 46.00 |
| Gemini-2.5-Pro (Comanici et al., 2025) | sequential | 55.00 | 42.48 | **55.41** | 50.96 | 0.144 | 0.103 | 0.177 | 0.141 | 52.00 | 49.00 | 50.50 |
| GPT-4o (Hurst et al., 2024) | sequential | 40.59 | 27.12 | 34.39 | 34.04 | 0.227 | 0.118 | 0.202 | 0.182 | 43.67 | 46.00 | 44.84 |
| GPT-5 (OpenAI, 2025a) | sequential | **70.00** | **63.69** | 54.90 | **62.86** | **0.411** | **0.326** | **0.352** | **0.363** | 59.67 | 61.00 | 60.34 |
| GPT-o3 (OpenAI, 2025b) | sequential | 59.12 | 44.44 | 55.41 | 52.99 | 0.348 | 0.278 | 0.313 | 0.313 | 57.67 | 55.33 | 56.50 |

| | Format | Trajectory | | | | Spatial Reasoning | | | | Long-horizon | Avg |
|---|---|---|---|---|---|---|---|---|---|---|---|
| | | GPR | HND | OBJ | Avg | LOC | PTH | DIR | Avg | | |
| Random Choice | x | 25 | 25 | 25 | 25 | 25 | 50 | 25 | 25 | 25 | |
| Human | - | - | - | - | - | - | - | - | - | - | - |
| *Open-source Models* | | | | | | | | | | | |
| DeepSeek-VL-7B (Lu et al., 2024) | merged | 41.03 | 38.72 | 22.67 | 34.14 | 42.02 | **37.68** | **32.00** | **37.23** | 20.00 | 23.89 |
| Molmo-7B-D-0924 (Deitke et al., 2025) | merged | 45.51 | 41.41 | 23.33 | 36.75 | 49.84 | 29.47 | 26.00 | 35.10 | 5.71 | 24.22 |
| InternVL2-4B (Chen et al., 2024) | merged | 44.55 | 34.01 | 25.67 | 34.74 | 40.07 | 33.82 | 26.33 | 33.41 | 8.57 | 20.45 |
| InternVL2-8B (Chen et al., 2024) | merged | 41.67 | 38.38 | 22.33 | 34.13 | 39.41 | 29.95 | 25.33 | 31.56 | 11.49 | 33.32 |
| InternVL2-26B (Chen et al., 2024) | merged | 53.21 | 43.77 | **30.33** | 42.44 | 26.06 | 26.57 | 22.00 | 24.88 | 11.29 | 25.66 |
| InternVL2-40B (Chen et al., 2024) | merged | **57.69** | 41.75 | 28.00 | 42.48 | 40.39 | 29.47 | 18.67 | 29.51 | 11.43 | 28.38 |
| InternVL3-8B (Zhu et al., 2025) | merged | 51.28 | 46.80 | 27.67 | 41.92 | **50.16** | 32.37 | 20.00 | 34.18 | 8.57 | 33.32 |
| InternVL3-14B (Zhu et al., 2025) | merged | 51.28 | 49.49 | 31.43 | 43.36 | 43.00 | 28.02 | 21.33 | 30.78 | **28.57** | **33.93** |
| LLaVa-NeXT-Interleave-7B (Li et al., 2024b) | merged | 37.18 | 37.04 | 20.67 | 31.63 | 37.79 | 27.54 | 19.67 | 28.33 | 5.71 | 14.64 |
| LLaVa-NeXT-Llama3-8B (Li et al., 2024a) | merged | 39.42 | 37.71 | 23.00 | 33.38 | 40.39 | 33.82 | 24.00 | 32.74 | 14.29 | 21.65 |
| Qwen2.5-VL-7B-Instruct (Bai et al., 2025) | merged | 54.49 | 48.15 | 30.00 | 44.21 | 38.44 | 31.40 | 21.00 | 30.28 | 22.86 | 21.44 |
| Qwen2.5-VL-32B-Instruct (Bai et al., 2025) | merged | 55.45 | **52.19** | 26.67 | **44.77** | 47.23 | 26.57 | 22.67 | 32.16 | 20.00 | 28.33 |
| *Proprietary Models* | | | | | | | | | | | |
| Claude-3.7-Sonnet (Anthropic, 2024) | sequential | 52.88 | 48.82 | 31.33 | 44.34 | 38.76 | 33.33 | 34.67 | 35.59 | 20.00 | 32.11 |
| Claude-4-Sonnet (Anthropic, 2025) | sequential | 50.00 | 49.16 | 38.00 | 45.72 | 46.25 | 42.51 | 39.67 | 42.81 | 17.14 | 33.05 |
| Gemini-2.0-Flash (Team, 2024) | sequential | 61.54 | 59.60 | 31.33 | 50.82 | 54.07 | 33.82 | 39.67 | 42.52 | 25.71 | 36.03 |
| Gemini-2.5-Flash (Comanici et al., 2025) | sequential | 64.42 | 63.97 | 45.00 | 57.80 | 61.24 | 43.00 | 44.67 | 49.64 | 31.43 | 38.24 |
| Gemini-2.5-Pro (Comanici et al., 2025) | sequential | 66.67 | 65.99 | 48.33 | 60.33 | 64.50 | 40.10 | 44.00 | 49.53 | 31.43 | 41.46 |
| GPT-4o (Hurst et al., 2024) | sequential | 41.99 | 35.35 | 30.67 | 36.00 | 60.91 | 33.33 | 31.00 | 41.75 | 31.43 | 32.90 |
| GPT-5 (OpenAI, 2025a) | sequential | **66.99** | 67.34 | 49.67 | 61.33 | **72.31** | **50.24** | 47.00 | **56.52** | **40.00** | **52.17** |
| GPT-o3 (OpenAI, 2025b) | sequential | **66.99** | **68.35** | **53.67** | **63.00** | 70.36 | 49.28 | **49.67** | 56.44 | 34.29 | 47.62 |

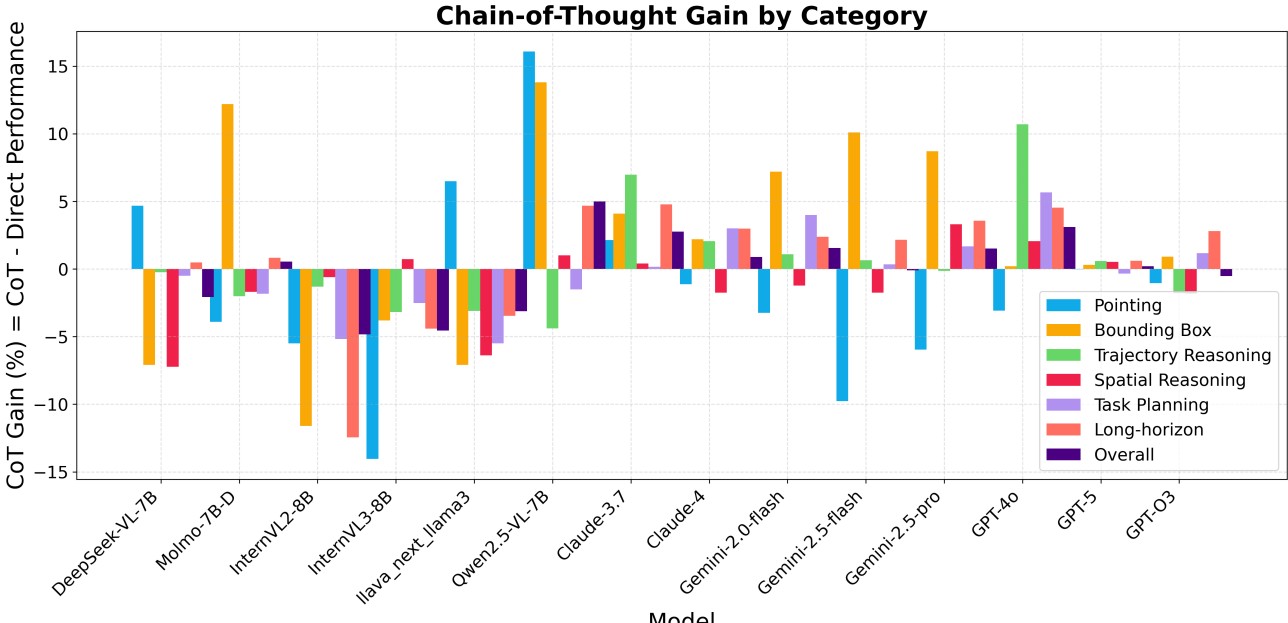

*Figure 44.* **Chain-of-thought gains across categories and models.** We evaluate the impact of Chain-of-Thought prompting on 13 state-of-the-art MLLMs and report the performance gain. Overall, CoT offers only marginal benefits and, in some cases, leads to negative effects.

### G.0.3. PERFORMANCE WITH COT

We analyze the impact of Chain-of-Thought(CoT) prompting on 6 categories of embodied reasoning tasks:*Pointing*, *Bounding Box*, *Trajectory Reasoning*, *Spatial Reasoning*, *Task Planning* and *Long-horizon*. We present the result of Chain-of-Thought(CoT) prompting in Table 5. And we visualize the performance gain in Figure 44.

Overall, CoT yields mixed results, with its effectiveness varying by skill and model type. But there are the following observations:

1. Overall, CoT offers limited and sometimes even negative overall performance, with its impact being highly category-specific and model-dependent.

2. For complex reasoning tasks such as *Trajectory Reasoning* and *Task Planning*, CoT tends to improve the performance of proprietary models, though the gains remain modest. We hypothesize that this improvement arises because these tasks inherently demand multi-step reasoning, where CoT provides an explicit structure for organizing intermediate decisions.

3. For low-level perception tasks such as *Pointing* and *Bounding Box*, the effect of CoT varies considerably across open-source models, as shown in Figure 44. In contrast, proprietary models show a consistent pattern: CoT improves performance on *Bounding Box* but reduces accuracy on *Pointing*. We hypothesize that CoT is beneficial for *Bounding Box* because it encourages structured reasoning and enforces a consistent output format (e.g., $(x_1, y_1, x_2, y_2)$). However, for *Pointing*, where the answer format is already simple (a single coordinate pair $(x, y)$), additional reasoning steps may introduce unnecessary complexity and disrupt direct visual grounding, ultimately degrading performance

4. For *Spatial Reasoning*, CoT prompting proves largely ineffective across models. We hypothesize that spatial understanding is inherently intuitive and often relies on non-verbal perceptual cues, whereas standard CoT enforces a sequential, language-based decomposition. This mismatch is likely to introduce errors into the reasoning chain, thereby degrading performance rather than enhancing it. For a more detailed analysis, we refer readers to the Appendix.

We conduct additional analyses to further verify our observations:

**Image**

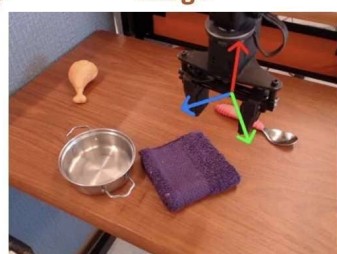

**Instruction:**
The image shows the current location of the robot
hand. There are four arrows pointing in different
directions. Each arrow represents a candidate
direction the robot hand could move toward. which
arrow should the robot follow to move toward the
**silver vessel**?
Options:\n
A. Green Arrow.
B. Blue Arrow.\n
C. Red Arrow.\n
D. None of the above.

**GPT-4o:** To determine which direction the robot hand should move to reach the
silver vessel, we can analyze the image as follows:\n\n1. **Silver Vessel
Location**: The silver vessel is on the left side of the image.\n\n2. **Arrow
Directions**:\n   - **Red Arrow**: Points upwards and slightly right.\n   - **Green
Arrow**: Points downward and left.\n   - **Blue Arrow**: Points to the left.\n\n3.
**Desired Movement**: To move the robot hand toward the silver vessel, the hand
should move to the left.\n\n4. **Correct Arrow**: The blue arrow correctly points
to the left toward the silver vessel.\n\nTherefore, the robot hand should follow
the **Blue Arrow**.\n\nOption B: Blue.

*Figure 45.* **The reasoning process of GPT-4o on** *Trajectory Reasoning.* The necessary reasoning step includes recognizing object location and then analyze the direction of trajectory arrows.

**Gemini-2.0-Flash:**

Okay, let's analyze the video to determine the order of actions and find the one
that doesn't happen before putting away the raisins.\n\n1.  The person puts a fork
into a case then puts a rubber band around it.\n2.  The person cleans the fork.\n3.
The person put the fork back to the case and put the rubber band around it.\n4.
The person opens a drawer to grab a cereal box (Shreddies).\n5.  The person pours
cereal into a bowl.\n6.  The person gets a green bag from a cupboard.\n7.  The
person opens the fridge.\n8.  The person gets a yogurt.\n9.  The person puts a
raisin into a bowl.\n10.The person adds the yogurt to the bowl, and the mix is
ready.\n11.The person cleans up around the table and puts away the yogurt.\n12.The
person puts away the cereal.\n13.The person puts away the green bag.\n14.The person
puts away raisins.\n\nNow let's check the options:\n\n*    **A: open drawer:** The
person opens a drawer before they get the cereal, before putting away raisins.\n*
**B: pour cereal:** The person pours cereal into a bowl, which is before they put
away raisins.\n*    **C: open fridge:** The person opens the fridge before they get
the yogurt, before putting away raisins.\n\nThus, all the actions A, B, and C
happen before the person puts away the raisins.\n\n**Final Answer: D**

*Figure 46.* **The reasoning process of Gemini-2.0-Flash on** *Task Process Reasoning.* The necessary reasoning step includes step-by-step analysis of the video and then checking the options.

1. As shown in Figure 44, all models achieve less than a 5% overall performance gain, with many showing improvements close to zero or even negative.

2. As shown in Figure 44, all the proprietary models receive positive performance gain. As shown in Table 5, for example, Gemini-2.0-Flash receive 3.99% improvement in *Task Planning* and 1.08% improvement in *Trajectory Reasoning*, and GPT-4o receive 5.66% improvement in *Task Planning* and 10.71% in *Trajectory Reasoning*. In order to analyze that why CoT can improve the performance in *Trajectory Reasoning* and *Task Planning*, we observe the Chain-of-Thought Reasoning process of Gemini-2.5-Flash, GPT-4o, and other models, and observe there is a very clear structure in the reasoning process of *Trajectory Reasoning* and *Task Planning* that can motivate the MLLMs to correctly answer the questions, as shown in Figure 45 and Figure 46.

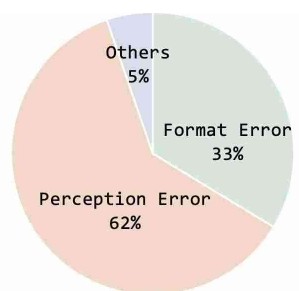 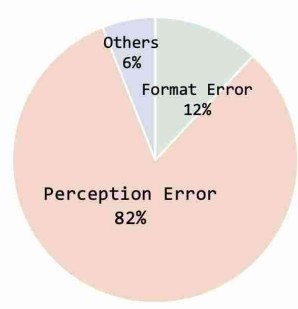

*Figure 47.* **(a) Error distribution of Gemini-2.5-Pro with *direct* prompting format. (b) Error distribution of Gemini-2.5-Pro with *CoT* prompting format.** CoT can significantly reduce format error.

3. For hypotheses on the effect of Chain-of-Thought on *Bounding Box* and *Pointing*, we perform further analysis to verify our following hypothesis:

   (a) CoT prompting beneifts the structured output of *Bounding Box* and format alignment. As shown in Figure 47, CoT can significantly reduce the format error of Gemini-2.5-Pro. Moreover, for other proprietary models, CoT also reduces format errors, with 6% for GPT-4o and 20% for Claude-4. As shown in Figure 48, our observations indicate that models consistently perform coordinate normalization as the final step of their output, which greatly improves format alignment.

   (b) For *Pointing*, we conduct a detailed analysis and find that enforcing reasoning in the model output can sometimes introduce incorrect information, which can possibly interfere with the final pointing decision and leads to incorrect conclusions, as shown in Figure 49 and Figure 50.

4. For *Spatial Reasoning*, we observe that errors often arise when the visual information cues provided to the model are misleading or incomplete. For example, within the *Object Localization* category, the model may mistakenly predict the presence of an object that is in fact absent, or it may fail to correctly identify the object's true location. Similarly, in the *Relative Direction* subcategory, models sometimes misinterpret the spatial relationship between objects, leading to incorrect judgments about orientation or relative position. In *Path Planning*, errors often occur when the model builds its reasoning on an incorrect spatial map, such as misidentifying obstacles or misplacing landmarks, which in turn results in invalid or suboptimal navigation paths. These examples highlight that spatial reasoning errors are typically not due to a lack of reasoning steps, but rather to incorrect grounding in the visual scene, which propagates through the reasoning process and ultimately leads to wrong conclusions.

**Gemini-2.5-Pro:**

 **Step 1:** The user wants me to identify the rearview mirror of the van in the provided image.\n\* **Step 2:** I will first examine the original image to locate the van. The van is the large white vehicle that dominates the center of the image.\n\* **Step 3:** Rearview mirrors, also known as side-view mirrors, are located on the sides of a vehicle, typically near the front doors. I will look at the front part of the van, which is on the right side of the image.\n\* **Step 4:** On the right side of the van, attached to the front passenger door just below the window, there is a black object protruding outwards. This is the rearview mirror.\n\* **Step 5:** I can use the provided crop images to confirm this. The third crop, in particular, gives a clear, close-up view of the front section of the van's side. In this crop, the black rearview mirror is clearly visible.\n\* **Step 6:** I will now define a bounding box that encloses the entire rearview mirror assembly, including the arm that attaches it to the van's door. I will estimate the normalized coordinates for this box based on its position in the original image. The mirror is in the upper-right quadrant of the image.\n\n(0.702788, 0.554378, 0.781682, 0.652462)

*Figure 48.* **CoT output of Gemini-2.5-Pro in** *Bounding Box*. The necessary reasoning step includes step-by-step analysis of object recognition and then normalization of coordinates.

**Claude-3.7-Sonnet**

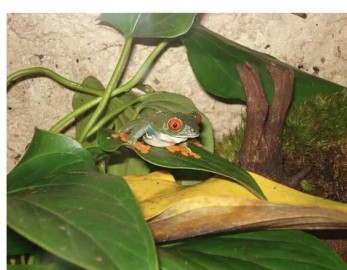

I need to identify the legs of the red-eyed tree frog in this image.\n\nLet me analyze the image step by step:\n\n1. First, I can see a red-eyed tree frog in the image - it has the characteristic red eyes, green body, and orange/yellow coloring on its feet.\n\n2. The frog is positioned among green leaves and appears to be resting on what looks like a yellow/golden colored surface (possibly a banana or similar fruit).\n\n3. Looking at the frog's body position, I can identify its legs:\n   - The frog is facing toward the camera/slightly to the right\n

- I can see its front legs/arms positioned in front of its body\n   - The back legs are less visible but can be seen supporting the frog's body\n\n4. The most clearly visible legs are the front legs, which show the characteristic orange/yellow coloring on the feet/toes that red-eyed tree frogs are known for.\n\n5. The front legs appear to be positioned in the lower-center portion of where the frog is located, with the distinctive orange-colored toes/feet being quite prominent.\n\nFocusing on the most visible and characteristic part of the frog's legs (the orange-colored feet/toes of the front legs), these appear to be located in the center-lower portion of the frog's body.\n\n(0.42, 0.65).

*Figure 49.* **CoT output of Claude-3.7-Sonnet in** *Pointing*. Red font indicates incorrect visual information.

**Claude-4-Sonnet**

I need to identify the rearview mirror of the white van in the image.\n\nLet me examine the van carefully:\n\n1. This is a white commercial van photographed from the side\n2. I can see the driver's side of the vehicle\n3. The rearview mirror would typically be mounted on the driver's side door or A-pillar area\n4. Looking at the front portion of the van, near the driver's door area\n\nI can see a dark-colored rearview mirror mounted on the driver's side of the van. It appears to be positioned on the door, extending outward from the vehicle. The mirror is visible as a dark rectangular/oval shape protruding from the side of the van.\nThe rearview mirror is located approximately in the upper-middle portion of the driver's side door area.\n\n(0.73, 0.47)"

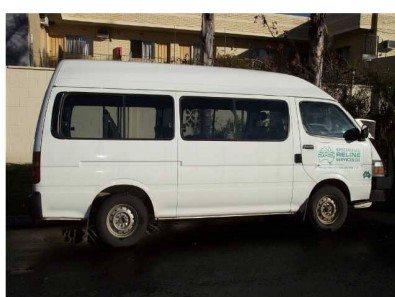

*Figure 50.* **CoT output of Claude-4-Sonnet in** *Pointing*. Red font indicates incorrect visual information.

*Table 5.* **CoT results on BEAR.** We use *Chain-of-Thought* prompting strategy for model inference, and present the performance below.

| | Format | Pointing | | | | Bounding Box | | | | Task Planning | | |
|---|---|---|---|---|---|---|---|---|---|---|---|---|
| | | GEN | SPA | PRT | Avg | GEN | SRA | PRT | Avg | PRG | PRD | Avg |
| Random Choice | | - | - | - | - | - | - | - | - | 25 | 25 | 25 |
| Human | | - | - | - | - | - | - | - | - | - | - | - |
| *Open-source Models* | | | | | | | | | | | | |
| DeepSeek-VL-7B (Lu et al., 2024) | merged | 22.06 | 10.13 | 13.69 | 15.29 | 0.167 | 0.132 | 0.153 | 0.151 | 34.67 | 29.33 | 32.00 |
| Molmo-7B-D (Deitke et al., 2025) | merged | 20.88 | 12.75 | 22.93 | 18.85 | 0.276 | 0.160 | 0.231 | 0.222 | 37.67 | 27.33 | 32.50 |
| InternVL2-8B (Chen et al., 2024) | merged | 17.35 | 12.09 | 19.11 | 16.18 | 0.123 | 0.084 | 0.112 | 0.106 | 39.33 | 26.00 | 32.67 |
| InternVL3-8B (Zhu et al., 2025) | merged | **37.65** | **28.43** | **30.89** | **32.32** | **0.311** | **0.237** | **0.281** | **0.276** | 38.33 | 33.33 | 35.83 |
| LLaVa-NeXT-Llama3-8B (Li et al., 2024a) | merged | 11.47 | 4.58 | 8.60 | 8.22 | 0.214 | 0.207 | 0.138 | 0.186 | 29.00 | 26.33 | 27.67 |
| Qwen2.5-VL-7B-Instruct (Bai et al., 2025) | merged | 17.35 | 17.97 | 21.66 | 18.99 | 0.158 | 0.133 | 0.145 | 0.145 | 36.00 | 34.00 | 35.00 |
| *Proprietary Models* | | | | | | | | | | | | |
| Claude-3.7-Sonnet (Anthropic, 2024) | sequential | 44.41 | 39.22 | 44.59 | 42.74 | 0.247 | 0.180 | 0.209 | 0.212 | 31.00 | 46.33 | 38.67 |
| Claude-4-Sonnet (Anthropic, 2025) | sequential | 42.35 | 37.42 | 42.36 | 40.71 | 0.263 | 0.173 | 0.221 | 0.219 | 47.00 | 40.67 | 43.84 |
| Gemini-2.0-Flash (Team, 2024) | sequential | 41.47 | 31.37 | 44.27 | 39.04 | 0.348 | 0.255 | 0.273 | 0.292 | 45.33 | 41.33 | 43.33 |
| Gemini-2.5-Flash (Comanici et al., 2025) | sequential | 32.94 | 23.86 | 33.44 | 30.08 | 0.290 | 0.244 | 0.252 | 0.262 | 52.67 | 40.00 | 46.34 |
| Gemini-2.5-Pro (Comanici et al., 2025) | sequential | 46.76 | 38.24 | 50.00 | 45.00 | 0.252 | 0.209 | 0.224 | 0.228 | 53.33 | 51.00 | 52.17 |
| GPT-4o (Hurst et al., 2024) | sequential | 39.41 | 22.22 | 31.21 | 30.95 | 0.224 | 0.128 | 0.200 | 0.184 | 50.67 | 50.33 | 50.50 |
| GPT-5 (OpenAI, 2025a) | sequential | **67.35** | **57.19** | **64.01** | **62.85** | **0.406** | **0.321** | **0.370** | **0.366** | 58.67 | 61.33 | 60.00 |
| GPT-o3 (OpenAI, 2025b) | sequential | 58.82 | 44.77 | 52.23 | 51.94 | 0.348 | 0.278 | 0.339 | 0.322 | 58.33 | 57.00 | 57.67 |

| | Format | Trajectory | | | | Spatial Reasoning | | | | Long-horizon | Avg |
|---|---|---|---|---|---|---|---|---|---|---|---|
| | | GPR | HND | OBJ | Avg | LOC | PTH | DIR | Avg | | |
| Random Choice | x | 25 | 25 | 25 | 25 | 25 | 50 | 25 | 25 | 25 | |
| Human | - | - | - | - | - | - | - | - | - | - | - |
| *Open-source Models* | | | | | | | | | | | |
| DeepSeek-VL-7B (Lu et al., 2024) | merged | 41.67 | 36.70 | 23.33 | 33.90 | 39.09 | 26.57 | 24.33 | 30.00 | **20.00** | 24.38 |
| Molmo-7B-D (Deitke et al., 2025) | merged | 44.55 | 34.01 | 25.67 | 34.74 | 40.07 | **33.82** | 26.33 | **33.41** | 8.57 | 25.05 |
| InternVL2-8B (Chen et al., 2024) | merged | 38.78 | 36.70 | 23.00 | 32.83 | 39.41 | 30.43 | 23.00 | 30.95 | 0 | 20.87 |
| InternVL3-8B (Zhu et al., 2025) | merged | 44.87 | 36.03 | **35.33** | 38.74 | **47.23** | **33.82** | 23.67 | 34.91 | 0 | 28.90 |
| LLaVa-NeXT-Llama3-8B (Li et al., 2024a) | merged | 36.86 | 32.32 | 21.67 | 30.28 | 26.71 | 27.05 | 25.33 | 26.36 | 0 | 18.19 |
| Qwen2.5-VL-7B-Instruct (Bai et al., 2025) | merged | **48.40** | **40.07** | 31.00 | **39.82** | 29.64 | 31.88 | **32.33** | 31.28 | 17.14 | **26.12** |
| *Proprietary Models* | | | | | | | | | | | |
| Claude-3.7-Sonnet (Anthropic, 2024) | sequential | 57.05 | 55.89 | 41.00 | 51.31 | 36.16 | 38.16 | 33.67 | 35.99 | 31.42 | 36.89 |
| Claude-4-Sonnet (Anthropic, 2025) | sequential | 55.24 | 44.78 | 43.33 | 47.78 | 44.63 | 41.55 | 37.00 | 41.06 | 22.86 | 36.03 |
| Gemini-2.0-Flash (Team, 2024) | sequential | 63.46 | 56.57 | 35.67 | 51.90 | 53.75 | 33.82 | 36.33 | 41.30 | 25.71 | 38.41 |
| Gemini-2.5-Flash (Comanici et al., 2025) | sequential | 65.06 | 62.29 | 48.00 | 58.45 | 57.65 | 42.03 | 44.00 | 47.89 | 31.43 | 40.40 |
| Gemini-2.5-Pro (Comanici et al., 2025) | sequential | **68.91** | 65.32 | 46.33 | 60.19 | 64.17 | 47.34 | 47.00 | 52.84 | 37.14 | 45.02 |
| GPT-4o (Hurst et al., 2024) | sequential | 51.28 | 50.51 | 38.33 | 46.71 | 56.03 | 41.06 | 34.33 | 43.81 | 34.29 | 37.44 |
| GPT-5 (OpenAI, 2025a) | sequential | 65.38 | **68.35** | **52.00** | **61.91** | **73.29** | **47.83** | **50.00** | **57.04** | 34.29 | **52.78** |
| GPT-o3 (OpenAI, 2025b) | sequential | 67.95 | 66.33 | 49.33 | 61.20 | 68.40 | **47.83** | 47.67 | 54.63 | **42.86** | 50.42 |

*Table 6.* **Results of different test-time scaling (TTS) strategies on BEAR-mini.**

| Model | Method | Reward Model | w/o tts | N=4 | N=8 | N=16 |
|---|---|---|---|---|---|---|
| Gemini 2.0 Flash | Majority Voting (Snell et al., 2024) | – | 36.0 | 37.1 | 39.0 | 39.4 |
| | Best of N (Lightman et al., 2023) | Gemini 2.0 Flash (Self) | | 39.8 | **40.9** | 38.9 |
| | Tournament (Son et al., 2025) | Gemini 2.0 Flash (Self) | | 38.9 | 36.3 | 37.9 |
| DeepSeek-VL-7B | Majority Voting (Snell et al., 2024) | – | 23.9 | 26.6 | 27.7 | 28.8 |
| | Best of N (Lightman et al., 2023) | Gemini 2.0 Flash | | 27.4 | **29.4** | 28.4 |
| | Tournament (Son et al., 2025) | Gemini 2.0 Flash | | 27.3 | 28.4 | 26.7 |

*Figure 51.* **(a) Performance with respect to model size**. We report overall performance across 6 categories. **(b) Performance with respect to frame number.** We report average performance of *Spatial Reasoning* and *Task Planning* to assess the effect of frame count on model performance.

### G.0.4. PERFORMANCE WITH TEST-TIME COMPUTE SCALING

Due to the significant test-time compute required, we use BEAR-mini: a subset of BEAR containing 40 samples per skill. We evaluate three common test-time compute scaling strategies on BEAR-mini: majority voting, Best-of-N selection, and Tournament-Style selection. Both Best-of-N and Tournament-Style selection rely on a reward model to identify the most suitable response among a set of candidates. We

- **Majority Voting** (Snell et al., 2024): Selects the most frequent answer among $N$ candidate responses. In case of a tie, one answer is randomly chosen from the top candidates.

- **Best-of-N** (Lightman et al., 2023): Uses a reward model to select the highest-scoring response from $N$ candidates. We choose $N = 4, 8, 16$ and conduct experiments with both a base model and a stronger reasoning model as the scorer.

- **Tournament-Style Selection** (Son et al., 2025): Uses a reward model to conduct pairwise comparisons between candidate responses and selects the overall winner via a tournament-style process.

**Reward models.** In the context of Test-time Scaling (TTS) experiments, when multiple candidate responses (e.g., five outputs) are generated, a reward model serves as an automatic evaluator by assigning a quality score to each response. These scores are then used to guide selection strategies such as Best-of-N (choosing the highest-scoring response) or Tournament Selection (progressively eliminating lower-scoring candidates). This mechanism enables performance gains at inference time without incurring additional training costs. **To ensure the reward model has sufficient context for scoring, we employ Chain-of-Thought (CoT) prompting to generate diverse candidate responses.** We report our experiment result on BEAR in Table 6.

### G.0.5. THE EFFECT OF NUMBER OF FRAMES

For video inputs, we uniformly sample frames to construct a compact yet representative sequence. Specifically, each video is downsampled into $N$ frames by evenly dividing the timeline, ensuring temporal coverage while reducing redundancy. Our sample strategy includes the first and the last frame in the video. We set $N = 16$ frames per video for proprietary models and $N = 32$ for open-source models. In Figure 51, our results indicate that varying the number of frames does not substantially impact performance.

### G.0.6. THE EFFECT OF MODEL SIZE

We also conduct model size scalability experiments, as shown in Figure 51, which demonstrate that increasing model size does not necessarily translate into improved performance. Based on the left panel of the figure, the analysis of model size effects reveals several insights. (1) The performance trajectory from 4B to 40B parameters demonstrates that model scaling does not follow a straightforward monotonic pattern, indicating increasing model size does not necessarily translate into improved performance. InternVL2 exhibits an inverted-U pattern, achieving peak performance at 8B parameters before experiencing performance degradation at larger scales. InternVL3 demonstrates optimal performance in the 8B-14B range (34%), followed by performance plateauing. Qwen2.5-VL shows relatively consistent but modest improvements across scales, with diminishing returns evident.

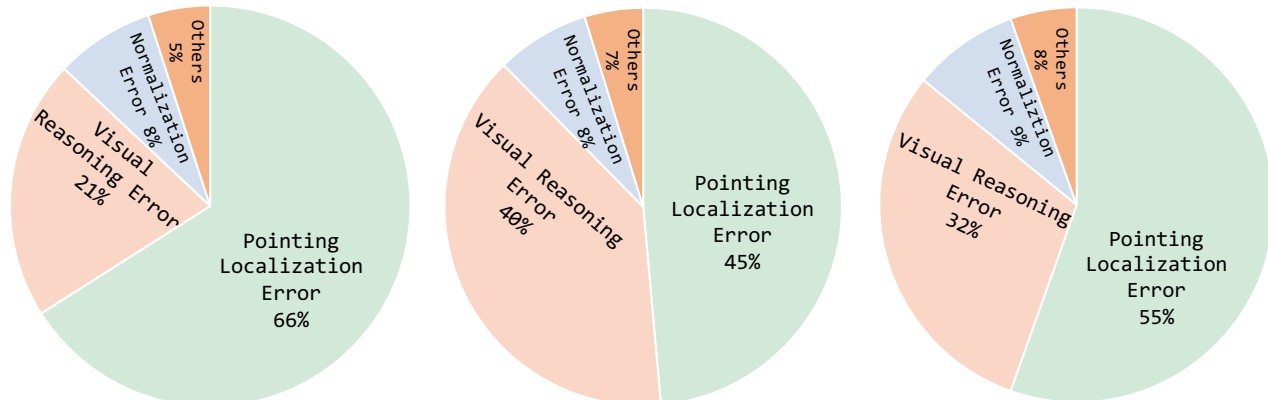

*Figure 52.* **(a)** **Error distribution of GPT-4o on** *General Object Pointing.* **(b)** **Error distribution of GPT-4o on** *Spatial Relationship Pointing.* **(c)** **Error distribution of GPT-4o on** *Semantic Part Pointing.*

## H. Error Analysis

We conduct a failure analysis for GPT-4o across 14 skills in 6 categories and identify several notable findings, which are summarized below. In addition, detailed failure analyzes for each category are provided in the following sub-subsections. During failure analysis, we explicitly adopt a chain-of-thought prompting strategy and instruct the model to generate its step-by-step reasoning process.

### H.0.1. POINTING

**General Object Pointing.** As illustrated in Figure 52, most failures in this category arise from *pointing localization errors*. In such cases, the MLLM successfully identifies the general region of the target object, but lacks the fine-grained visual discrimination required to select an accurate point on the object. This often results in predictions that are slightly offset from the ground truth, reflecting the model's limitations in pixel-level localization, as shown in Figure 53. Another large source of error is *visual reasoning errors*, where the model completely misinterprets the visual scene, either by confusing one object for another or by reasoning about an entirely incorrect location. Representative examples are shown in Figure 56.

**Spatial Relationship Pointing.** The failure distribution in this task mirrors that of *General Object Pointing*, but with a higher incidence of *visual reasoning errors*. This shift is largely due to the added complexity of relational reasoning: instead of simply localizing an object, the model must infer the correct spatial relation between two or more objects (e.g., 'the mug to the left of the laptop'). MLLMs frequently confuse such relationships, reasoning about an object pair that does not satisfy the described relation, or incorrectly inferring spatial directionality. This highlights their difficulty in grounding linguistic spatial terms into precise visual configurations.

**Semantic Part Pointing.** A similar trend is observed for *Semantic Part Pointing*, with error patterns resembling those in *General Object Pointing*. However, the proportion of *visual reasoning errors* is even larger in this case. The main challenge stems from the necessity to correctly identify and localize semantic parts of an object (e.g., 'the handle of the cup' or 'the wheels of the chair'). MLLMs often struggle to differentiate between object-level localization and part-level grounding, resulting in confusion between distinct parts of the same object, or even in predictions that land on irrelevant objects. This suggests that part-level grounding demands finer granularity of both visual parsing and semantic understanding than the models currently possess.

### H.0.2. BOUNDING BOX

For the *Bounding Box* category, defining errors is non-trivial since the evaluation metric: Intersection over Union (IoU), is a continuous score ranging from 0 to 1. However, there are occasional instances where models receive a score of 0, indicating complete failure to localize the target object. Our error analysis focuses specifically on these failure cases. We report our failure analysis result in Figure 58.

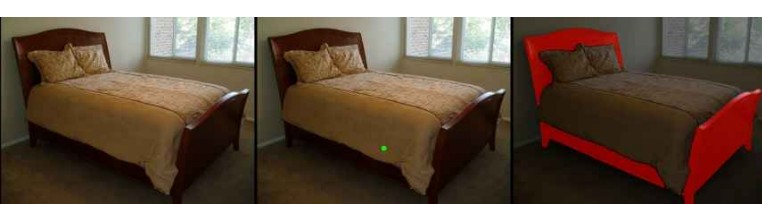

**Pointing Localization Error**

**Question:** Locate the bed frame in the image.
**Model's Output:** To locate the bed frame in the image, let's break down the steps:
1. **Understand the Image Dimensions**: The image has a resolution of 1024x768 pixels. 2. **Identify the Bed Frame's Location**: - The bed frame is typically the wooden structure of the bed. - It can be seen along the bottom edge of the bed in the image. 3. **Determine the Central Point**: - The bed frame extends across the width of the bed. - The central vertical position of the bed frame is along the lower portion of the image, centered horizontally. 4. **Normalize the Coordinates**: - For normalization, divide the x and y coordinates by the image's width and height respectively. 5. **Calculate the Center of the Bed Frame**: - The center of the bed frame's approximate position on the x-axis would be around the middle of the image: 0.5. - Its position on the y-axis would be closer to the bottom, let's estimate around 0.7. **Final Answer**: The center of the bed frame is approximately at pixel location (0.5, 0.7).

*Figure 53.* **Example of pointing localization error.** Localization error refers to cases where the model correctly reasons about the approximate location of the target point but, due to limited fine-grained visual reasoning ability, fails to predict the precise location.

**General Object Bounding Box**   As shown in Figure 58, the predominant source of error is due to *localization errors*, where the model demonstrates correct high-level reasoning about the target object but lacks the fine-grained visual grounding required to convert that understanding into precise bounding box coordinates, see Figure 60. Additionally, 20% of the errors are attributed to *identification errors*, where the model fails to recognize the correct object in the image, see Figure 59. Another 20% of the errors arise from *reasoning errors*, where the model's inference about the approximate coarse location of the target object is fundamentally flawed, see Figure 61.

**Spatial Relationship Bounding Box**   As shown in Figure 58, 54% of the errors stem from *localization errors*. 8% are due to *identification errors*, where the model fails to detect the correct type of object in the image. Another 8% are caused by *reasoning errors*, where the model correctly identifies the target object type but fails to infer its approximate spatial location. Notably, 30% of the errors arise from *spatial relationship errors*, as shown in Figure 62, where the model successfully detects the relevant objects but misunderstands their spatial relationships, leading to incorrect selection of the object referred to in the instruction.

**Semantic Part Bounding Box**   As shown in Figure 58, 75% of the errors are due to incorrect localization of the target object, while the remaining 25% result from incorrect identification of the target part.

H.0.3. TRAJECTORY REASONING

**Gripper Trajectory Reasoning.**   As shown in Figure 63, the majority of errors (52%) are categorized as *Trajectory Direction Reasoning Errors*, as shown in Figure 64 indicating that the model lacks the ability to infer the correct direction of the trajectory arrow and its intended destination. We also find that 20% of errors occur when the model fails to distinguish the color of the arrow, as shown in Figure 65; for example, it misidentifies a 'red arrow' as a 'blue arrow'. In 25% of cases, the model localizes the wrong object, such as confusing the target 'chicken leg' with another object indicated by the arrow.

**Human Hand Trajectory Reasoning.**   As shown in Figure 63, the majority of errors (44%) occur when the model fails to infer the correct direction of the human hand. Another 29% stem from incorrect object localization, such as misidentifying the chicken legs, especially in cluttered environments. Additionally, 20% are reasoning chain errors, where the model's intermediate reasoning steps are logically consistent and may even identify the correct target, yet the final prediction contradicts its own reasoning, resulting in self-inconsistency.

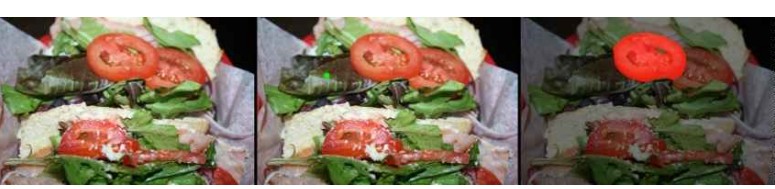

*Figure 54.* **Example of pointing localization error.** Localization error refers to cases where the model correctly reasons about the approximate location of the target point but, due to limited fine-grained visual reasoning ability, fails to predict the precise location.

**Object Trajectory Reasoning.** As shown in Figure 63, the majority of errors (35%) arise from *trajectory direction reasoning*, where the model fails to identify the correct direction of the trajectory. In addition, a substantial portion of errors stem from knowledge errors, as shown in Figure 67, which occur when the model lacks basic knowledge about object interactions, such as how to open a door or how to rotate a lid to open a bottle. About 20% of errors are due to failures in interpreting trajectories from alternative viewpoints, as shown in Figure 66, for example, when the model cannot determine how to open a drawer from a side view. Finally, 15% of errors result from incorrect part localization, such as failing to recognize the microwave handle needed to open the door.

### H.0.4. SPATIAL REASONING

**Object Localization.** As shown in Figure 68, the majority of *object recognition error* (45%) stem from *3D spatial layout error*, where MLLMs struggle to accurately infer about the correct 3D spatial layout of the scene and spatial relationships between objects, as shown in Figure 69. Furthermore, in cluttered environments such as grocery rooms, MLLMs often fail to recognize small and partially occluded objects, suggesting that their *visual detail reasoning* capabilities require further improvement, as shown in Figure 71 and Figure 78.

**Path Planning.** As shown in Figure 68, 46% of error comes from *spatial direction understanding error*, referring to the model's failure to reason correctly about relative directions (e.g., left, right, front, back) from an egocentric perspective, as shown in Figure 70. Possible reasons include the lack of egocentric-aware supervision during training, which leads to incorrect spatial reasoning from a first-person perspective. In addition, *multi-frame misalignment error* also accounts for a significant portion of the failures (35%), the models can not consistent reason and track objects and their spatial relations among different frames, as shown in Figure 72. Also, *3D spatial reasoning error* accounts for 15%, indicating that models sometimes fail to reason accurately about the spatial layout of the scene, as shown in Figure 69.

**Relative Direction.** As shown in Figure 68, the majority of the failure cases comes from *spatial direction understanding error*(45%), in which models can not correctly reason about direction in egocentric view, as shown in Figure 70 and *spatial layout error*(45%), in which models can not correctly reason about the 3D layout of the scene and the spatial relationship between objects, as shown in Figure 69. Another small portion of failure cases comes from *camera motion error* (14%), where models can not correctly infer the movement of camera, leading to incorrect reason about the spatial layout, as shown in Figure 73.

### H.0.5. TASK PLANNING

**Next Action Prediction.** As shown in Figure 74, The error analysis of *Next Action Prediction* reveals that the majority of failures stem from *action understanding* (58%), where the MLLM fails to understand the action depicted in each

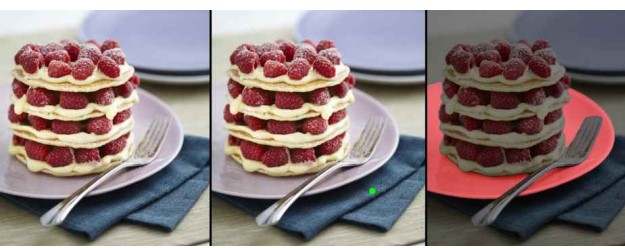

**Question:** Identify the empty space on the plate with pancakes on it. A: To identify
**Model's Output:** the empty space on the plate with pancakes: 1. Examine the Image:
Look for areas on the plate that have no visible pancakes or toppings. 2. Location
Assessment: The plate is mostly covered by the stack of pancakes and raspberries,
but there is some visible area around the pancakes, especially towards the right
edge of the plate. 3. Identify Empty Space: The bottom-right area of the plate
seems to have some empty space. Given the image dimensions (1024 x 683): - The
empty space is around the bottom-right quadrant of the plate. - Estimate the
approximate location in normalized coordinates. Empty space location: - x: Around
0.75 (right side of the image) - y: Around 0.75 (lower part of the image) Final
pixel location: (0.75, 0.75)

*Figure 55.* **Example of pointing localization error.** Localization error refers to cases where the model correctly reasons about the approximate location of the target point but, due to limited fine-grained visual reasoning ability, fails to predict the precise location.

observation, as shown in Figure 75, followed by issues related to *historical context omission* (17%), as shown in Figure 77 and *hallucinated reasoning* (17%). *Hallucinated reasoning* refers to instances where models generate inferences based on conjecture rather than grounded observations, as shown in Figure 76. *Object recognition* error are relatively rare, accounting for only 8%, as shown in Figure 78.

**Task Process Reasoning.** As shown in Figure 74, the error analysis of *Task Process Reasoning* indicates that the most common failure arises from *action understanding* error (46%), as shown in Figure 75, where the MLLM fails to accurately interpret the action associated with a given observation, and *history omission* (46%), as shown in Figure 77, where the MLLM fails to identify and remember actions in sequential order. *Hallucinated reasoning* accounts for 22% of the error, reflecting cases where the model makes incorrect inferences not supported by visual evidence, as shown in Figure 77. A small portion (4%) falls under miscellaneous error categorized as *others*.

H.0.6. LONG-HORIZON

As shown in Figure 79, most failure cases arise from perception and trajectory errors. We provide one failure cases for *Spatial Reasoning*, *Trajectory Reasoning* and *Planning Errors*, as shown in Figure 82, Figure 81 and Figure 80.

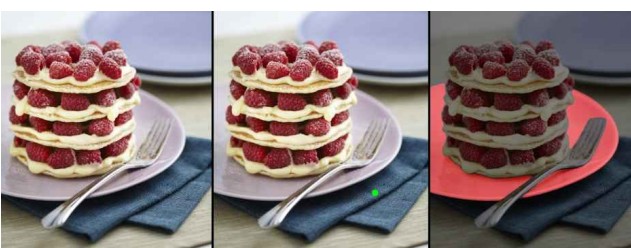

**Question:** Identify the empty space on the plate with pancakes on it. A: To identify
**Model's Output:** the empty space on the plate with pancakes: 1. Examine the Image:
Look for areas on the plate that have no visible pancakes or toppings. 2. Location
Assessment: The plate is mostly covered by the stack of pancakes and raspberries,
but there is some visible area around the pancakes, especially towards the right
edge of the plate. 3. Identify Empty Space: The bottom-right area of the plate
seems to have some empty space. Given the image dimensions (1024 x 683): - The
empty space is around the bottom-right quadrant of the plate. - Estimate the
approximate location in normalized coordinates. Empty space location: - x: Around
0.75 (right side of the image) - y: Around 0.75 (lower part of the image) Final
pixel location: (0.75, 0.75)

*Figure 56.* **Example of visual reasoning error.** In this case, GPT-4o reason about the wrong location of the front legs of crocodile.

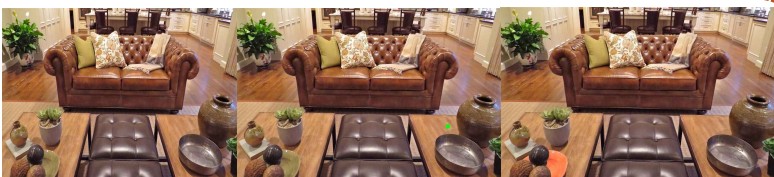

**Question:** Identify the nearest plate in the image.
**Model's Output:** "To locate the nearest plate in the image, follow these
steps:\n\n1. **Identify the objects in the foreground**: \n    - The picture shows a
living room setting with a coffee table in front of a leather sofa.\n    - On the
table, there are various decorative items, including bowls, a small plant, and a
plate.\n\n2. **Determine which object is a plate**:\n    - The object resembling a
plate is in the lower right corner of the image, and it is metallic and flat.\n\n3.
**Estimate the location of the center of the plate**:\n    - The plate is positioned
towards the right side of the image. \n    - Vertically, it is approximately halfway
on the table, a little below center.\n\n4. **Normalize the coordinates**:\n    -
Suppose the plate is about 80% across horizontally (x-coordinate) and about 70%
down vertically (y-coordinate).\n\n5. **Output the normalized coordinates**:\n    -
The approximate normalized location of the plate's center is (0.80, 0.70).\n\nThus,
the final normalized pixel location of the nearest plate is approximately (0.80,
0.70)."

*Figure 57.* **Example of visual reasoning error.** In this case, GPT-4o reason about the wrong location of the front legs of crocodile.

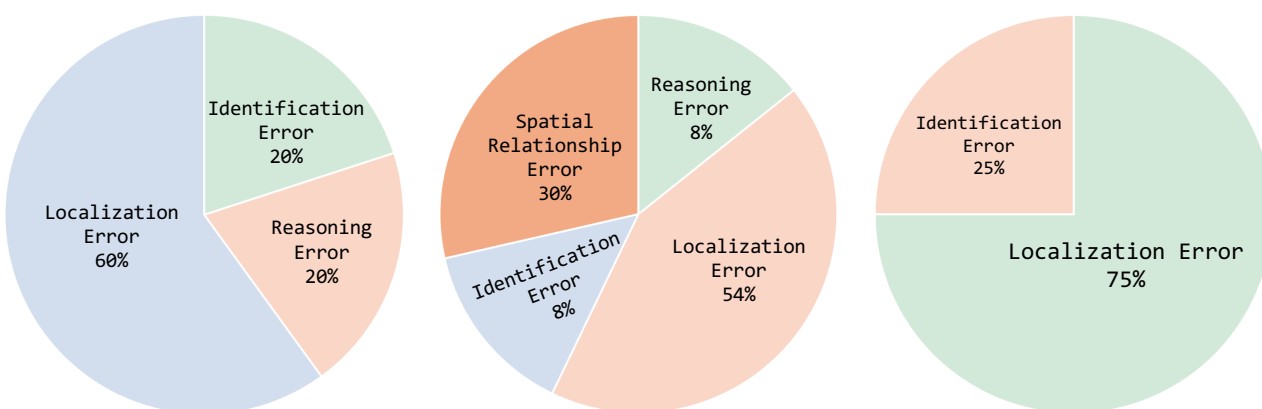

*Figure 58.* **(a)** **Error distribution of GPT-4o on** *General Object Bounding Box.* **(b)** **Error distribution of GPT-4o on** *Spatial Relationship Bounding Box.* **(c)** **Error distribution of GPT-4o on** *Semantic Part Bounding Box.* Please note that due to the evaluation metric of *Bounding Box* is IoU, which is a floating point between 0 and 1, we only do failure analysis on cases when IoU equals to 0.

---

### Identification Errors

**Question:** Identify the person in the image.

**Model's Output:** I'm sorry, but I can't identify or recognize the person in the image.

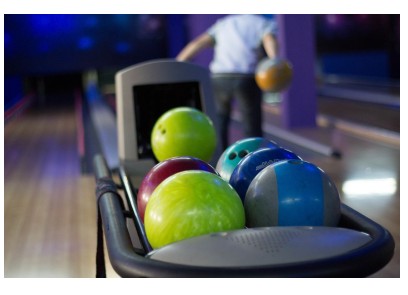

*Figure 59.* **Example of identification error.** In this case, GPT-4o reason about the wrong location of the front legs of crocodile.

---

### Localization Errors

**Question:** Identify the red dot in the keyboard.

**Model's Output:** To identify the red dot in the keyboard, which is typically a TrackPoint or pointing stick found in some laptops, let's analyze the image step by step:\n\n1. **Locate the Red Dot**: The red dot is usually situated in the center of the keyboard, often between the G, H, and B keys.\n\n2. **Determine Its Position**: From observation, the red dot is centrally located near these keys.\n\n3. **Bounding Box Calculation**:\n   - Assume the image dimensions are normalized to 1x1.\n   - Calculate the approximate position of the red dot relative to the distribution of the keys.\n\n4. **Normalized Coordinates**:\n   - Top-left corner of the trackpoint (x1, y1): (~0.47, ~0.4)\n   - Bottom-right corner of the trackpoint (x2, y2): (~0.53, ~0.45)\n\nGiven these observations and calculations, the bounding box for the red dot would be approximately:\n\n**(0.47, 0.4, 0.53, 0.45)**

*Figure 60.* **Example of localization error.** In this case, GPT-4o reason about the wrong location of the front legs of crocodile.

## Reasoning Errors

**Model's Output:** To identify the head of the monkey in the image and define the bounding box, let's follow a step-by-step approach:\n\n1. **Identify the Monkey's Position**: \n  - The monkey is located approximately in the center of the image, slightly to the left.\n\n2. **Locate the Head**:\n  - The head of the monkey is towards the upper left of its body, which is facing towards the right side of the image.\n  - The head is round and is positioned in the upper central section of the monkey's body.\n\n3. **Determine Bounding Box for the Head**:\n  - Visually, the head seems to occupy a small portion of the total image.\n\n4. **Define Coordinates**:\n  - Roughly estimate the position of the head for bounding box normalization.\n\n  Assume the following approximate coordinates:\n  - Top-left corner (x1, y1): (0.45, 0.35)\n  - Bottom-right corner (x2, y2): (0.55, 0.45)\n\nThis results in the bounding box for the head being approximately:\n\n**Bounding Box**: (0.45, 0.35, 0.55, 0.45)

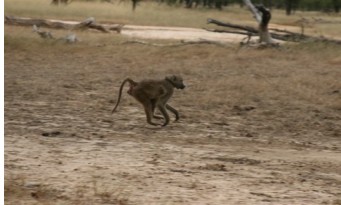

**Question:** Identify the head of the monkey in the image.

*Figure 61.* **Example of reasoning error.** In this case, GPT-4o reason about the wrong location of the front legs of crocodile.

## Spatial Reasoning Errors

**Question:** Identify the nearest plate.

Model's Output: "To identify and locate the nearest plate, let's analyze the image step-by-step:\n\n1. **Identify Objects**: \n  - The image has a brown leather sofa, a cocktail table with decorative items, and a large vase on the table.\n  - On the table in front of the sofa, there are several decorative items, including a round, metallic bowl that can be identified as a plate.\n\n2. **Locate the Plate**:\n  - The metal plate is to the right side of the table in the foreground.\n\n3. **Determine Bounding Box**: \n  - Estimate the position of the top-left and bottom-right corners of the plate.\n  - The plate is positioned mostly to the right and near the bottom of the image.\n\n4. **Normalize Coordinates**: \n  - Calculate the relative positions by dividing the plate's position by the dimensions of the image:\n\nGiven these observations, let's estimate the bounding box:\n- Top-left corner (x1, y1): approximately (0.65, 0.70)\n- Bottom-right corner (x2, y2): approximately (0.85, 0.90)\n\nThus, the normalized bounding box coordinates for the plate are:\n\n\\[\n(x1, y1, x2, y2) = (0.65, 0.70, 0.85, 0.90)\n\\]",

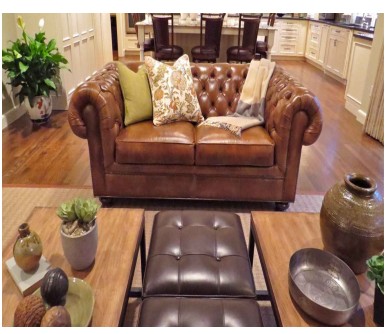

*Figure 62.* **Example of spatial reasoning error.** In this case, GPT-4o reason about the wrong location of the front legs of crocodile.

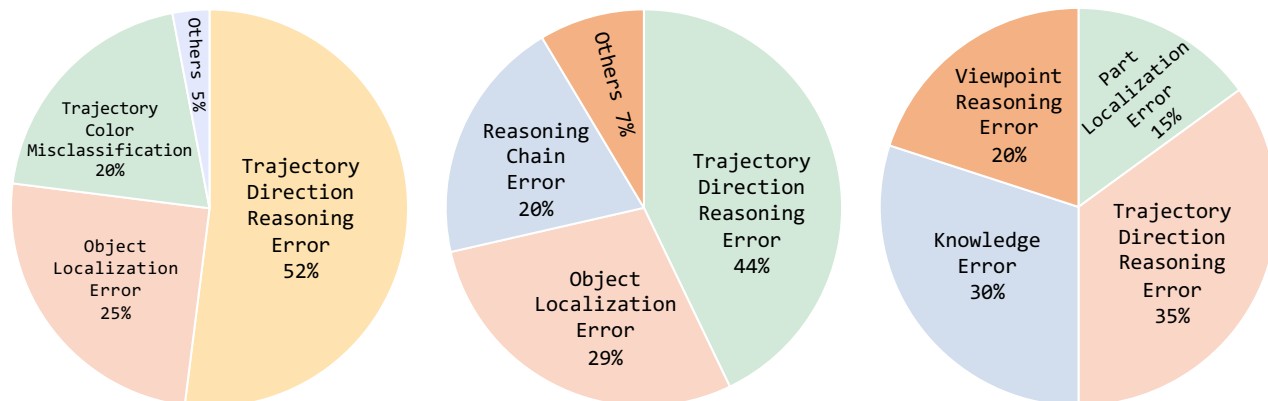

*Figure 63.* **(a)** Error distribution of GPT-4o on *Gripper Trajectory Reasoning*. **(b)** Error distribution of GPT-4o on *Human Hand Trajectory Reasoning*. **(c)** Error distribution of GPT-4o on *Object Trajectory Reasoning*.

---

### Trajectory Direction Reasoning Error

**Question:** The image shows the current location of the robot hand. There are four arrows pointing in different directions. Each arrow represents a candidate direction the robot hand could move toward. Which arrow should the robot follow to move toward the towel?
Options:
A: Green
B: Blue
C: Red
D: None of the above

**Model's Output:** "To determine which arrow the robot hand should follow to move toward the towel, let's analyze the image:\n\n1. **Identify the towel's location**: The towel is placed on the left side of the image, on the wooden surface.\n2. **Assess the direction of each arrow**:\n - The **green arrow** is pointing to the left and slightly up.\n - The **blue arrow** is pointing straight down.\n - The **red arrow** is pointing to the right.\n\n3. **Determine which arrow points towards the towel**:\n - Since the towel is to the left, the arrow that points left and slightly up is the most direct path toward it.\n\nBased on this analysis, the correct arrow is the **green arrow**.\n\nTherefore, the robot should follow option **A: Green** to move toward the towel.",

---

*Figure 64.* **Example of trajectory direction reasoning error** In this image, GPT-4o misidentifies the red arrow to be pointing right.

---

### Color Confusing Error

**Question:**
The image shows the current location of the robot hand. There are four arrows pointing in different directions. Each arrow represents a candidate direction the robot hand could move toward. which arrow should the robot follow to move toward the **purple cloth**?

**Options:**
A: Green,
B: Blue,
C: Red,
D: None of the above

**Note:**
sometimes, the model confuses the blue trajectory and green trajectory

**Model's Output:** To determine the correct direction for the robot hand to move toward the purple cloth, let's consider the position of both the robot hand and the purple cloth:\n\n1. **Robot Hand Position:** The robot hand is currently in a position facing several items including a metal pot with a toy drumstick, a spoon, and a pink object.\n2. **Purple Cloth Location:** The purple cloth is positioned to the left of the robot hand's current position on the table.\n3. **Arrow Directions:**\n - **Green Arrow:** Points to the left.\n - **Blue Arrow:** Points to the right.\n - **Red Arrow:** Points downward, indicating the direction parallel to the table surface toward the bottom of the image.\n - **None of the above:** This would mean no arrows are correct.\n4. To move toward the **purple cloth**, which is to the **left** of the robot hand, the robot should follow the **green arrow** direction. \n\nThus, the correct option is:\n\n**A: Green**

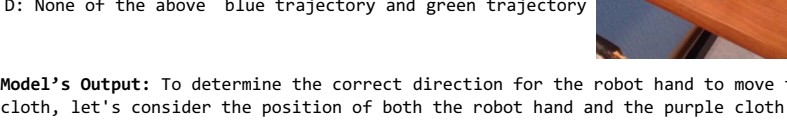

---

*Figure 65.* **Example of color confusing error** In this case, GPT-4o reason misidentifies blue trajectory arrow as green trajectory arrow and in reverse.

## Viewpoint Error

**Question:** Identify the arrow that indicates the direction the hand will move to pull out the tissue from the box.

**Options:**
A: Red Arrow,
B: Green Arrow,
C: Yellow Arrow,
D: None of the above

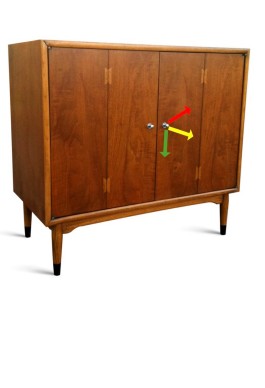

**Model's Output:** To determine the correct option, let's consider the doors of the cabinet:\n\n1. **Position of Handles:** The handles are located centrally on each door.\n2. **Type of Door:** The cabinet has doors that swing open rather than sliding.\n3. **Motion to Pull Open:** To open the door, you would pull the handle directly towards you.\n\nNow, let's analyze the arrows:\n\n- **Red Arrow:** Points towards the top of the cabinet\u2014incorrect for pulling open the doors.\n- **Green Arrow:** Points directly outward from the cabinet\u2014correct direction for pulling the doors.\n- **Yellow Arrow:** Points sideways or parallel to the doors\u2014incorrect for pulling the doors open.\n\nBased on this analysis, the correct option for the direction in which the hand should move to open the cabinet doors is **B: Green Arrow**

*Figure 66.* **Example of viewpoint error.** In this case, GPT-4o can not reason about the correct trajectory to pull out the door in side view.

## Knowledge Error

**Question:** Identify the arrow that indicates the direction the hand will move to pull out the tissue from the box.

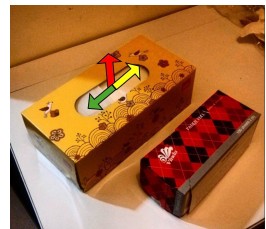

**options:**
A: Red Arrow,
B: Green Arrow,
C: Yellow Arrow,
D: None of the above

**Model's Output:** "To determine the correct direction to pull out the tissue from the box, follow these steps:\n\n1. **Locate the tissue opening:** The oval-shaped cutout on the top of the box is where the tissues are pulled from.\n\n2. **Identify tissue placement:** Tissues are usually folded in a way that they are pulled towards the outer edge of the opening.\n\n3. **Consider the orientation of the box:** The opening is positioned with its longer side in parallel with the longer edges of the box.\n\n4. **Assess arrow directions:**\n   - **Red Arrow:** Points upwards, which doesn't align with pulling tissues.\n   - **Green Arrow:** Points sideways away from the opening\u2014unlikely to be the correct direction.\n   - **Yellow Arrow:** Points horizontally across the longer side, aligning with how tissues are typically pulled out.\n\n5. **Correct direction:** The Yellow Arrow shows the most likely correct direction for pulling out a tissue from the box.\n\nTherefore, the correct option is:\n\n**C: Yellow Arrow**",

*Figure 67.* **Example of knowledge error.** In this case, GPT-4o do not know the correct trajectory to pull out the tissue in the tissue box is to pull it upwards.

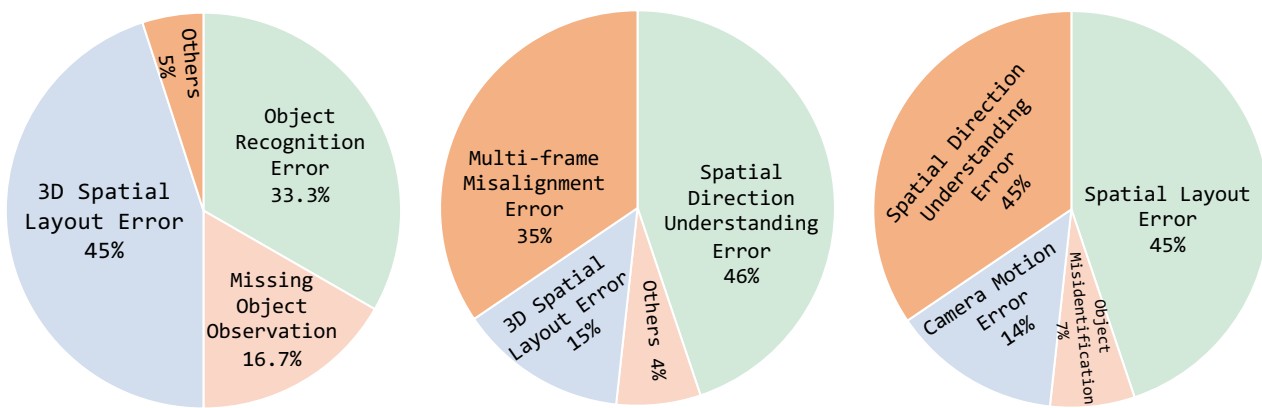

*Figure 68.* **(a)** **Error distribution of GPT-4o on** *Object Localization.* **(b)** **Error distribution of GPT-4o on** *Path Planning.* **(c)** **Error distribution of GPT-4o on** *Relative Direction.*

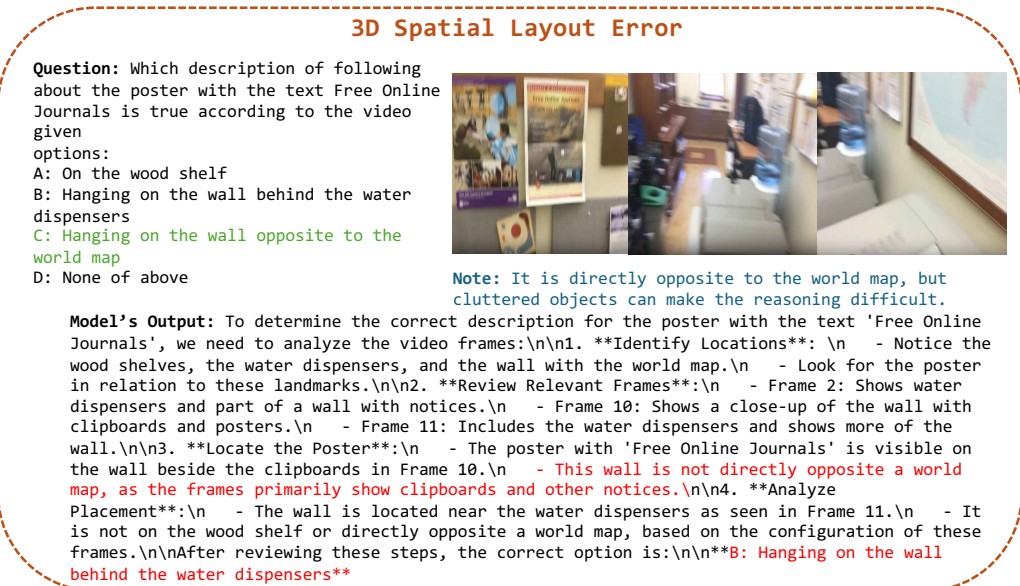

*Figure 69.* **Example of 3D spatial layout error.** 3D spatial layout error refers to a model's failure to understand the three-dimensional spatial relationships between objects. This type of error involves incorrect or incomplete reasoning about depth, vertical positioning (above/below), occlusion, and relative distance between objects in a scene.

---

**Spatial Direction Understanding Error**

**Question:** "You want to navigate to the window. You will perform the following actions (Note: for each [please fill in], choose either 'turn back,' 'turn left,' or 'turn right.'): 1. Go forward until the edge of bed 2. [please fill in] 3. Go forward until the wall 4. [please fill in] 5. Go forward until the window. You have reached the final destination.",

options:
A: Turn Back, Turn Left,
B: Turn Left, Turn Right,
C: Turn Right, Turn Left,
D: Turn Left, Turn Left"

**Model's Output:** "To reach the window, follow these steps:\n\n1. **Go forward until the edge of the bed:** Start by moving forward along the bed.\n \n2. **Turn right:** You are facing the bed, and to align with the path towards the window, turn right.\n\n3. *Go forward until the wall:** Move forward in the space between the bed and the wall until you reach the wall near the window area.\n\n4. **Turn left:** Once you reach the wall, turn left to face the window.\n\n5. **Go forward until the window:** Walk straight ahead towards the window.\n\nFollowing these steps, the correct option is:\n**C: Turn Right, Turn Left**",

**Note:** The direction was systematically reversed (egocentric reference frame misused)
The model chose "Turn right" in step 2, which means it assumed that the right-hand side along the direction of movement around the bed leads to the window.
However, based on the video, it should have turned **left** to go around the head of the bed and approach the wall, and then right to face the window.

*Figure 70.* **Example of spatial direction reasoning error.** Spatial direction reasoning error refers to a model's failure to reason correctly about relative directions (e.g., left, right, front, back) in an egocentric view. This error typically stems from confusion in the agent's internal coordinate system, failing to account for its own orientation changes across steps, or misinterpreting directions based on an allocentric perspective. As a result, the model systematically selects incorrect turning directions or paths.

---

**Missing Object Recognition**

**Question:** Which description of following about the blue and white folded towel is true according to the video given
options:
A: On top of the microwave
B: Hanging on the cabinet door
C: Inside the sink
D: None of above

**Note:** There is already a blue and white towel on the microwave in the image.

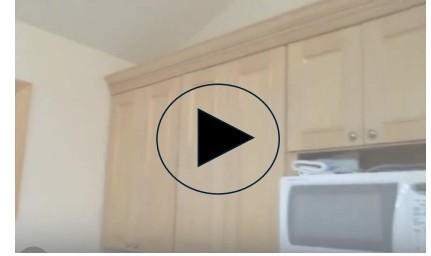

**Model's Output:** "To determine the location of the blue and white folded towel in the video, let's analyze the images step by step:\n\n1. **Images of the Kitchen:**\n - **Refrigerator:** Shows a white towel hanging on the fridge handle.\n - **Microwave:** No towel on or near it.\n - **Cabinets:** No visible towel hanging on the cabinet door.\n - **Sink Area:** No towel visible inside or near the sink.\n\n2. **Options Analysis:**\n - **A: On top of the microwave**: No towel is visible on top of the microwave.\n - **B: Hanging on the cabinet door**: No towel is visible hanging on any cabinet door.\n - **C: Inside the sink**: No towel is visible inside the sink.\n - **D: None of the above**: Since the towel is not in any of the specified locations, this option is the most accurate.\n\nBased on these observations, the correct option is **D: None of the above**.",

*Figure 71.* **Example of missing object recognition error.** Sometimes when the observation comes across the target object, the model can not see it.

## Multi-frame Misalignment Error

**Question:** You want to navigate to the refrigerator. You will perform the following actions (Note: for each [please fill in], choose either 'turn back,' 'turn left,' or 'turn right.'): 1. [please fill in] 2. Go forward until the refrigerator. You have reached the final destination.,
options:
A: Turn Back,
B: Turn Right,
C: Turn Left

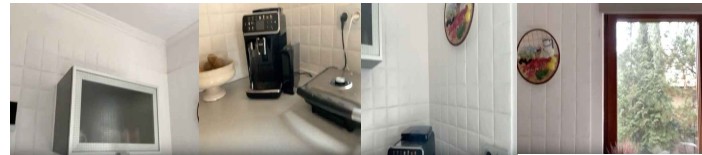

**Model's Output:** "To navigate to the refrigerator, let's consider the layout:\n\n1. The current observation frame shows the kitchen sink and counter.\n2. In the past observation frames, the refrigerator is visible to the left side of the counter.\n\nSteps to navigate:\n\n1. **Turn Left**: This should orient you toward the refrigerator.\n2. **Go forward until the refrigerator**.\n\nTherefore, the correct option is **C.C. Turn Left**.",

**Note:** from 26 second to 30 second, we can see that there is a right angle of the counter and wall when the camera moves right. It seems multiple frames can be confusing to MLLM to extract object relationships, so the relative position of reference frame and fridge should be back instead of left if the model can combine information from different frames

*Figure 72.* **Example of multi-frame misalignment error.** Multi-frame misalignment error refers to a model's failure to integrate spatial information consistently across multiple frames. Even when the target object appears clearly in one frame, the model may fail to recognize or localize it due to incorrect temporal alignment, inconsistent egocentric viewpoint tracking, or failure to update its internal scene representation.

## Camera Motion Error

**Question:** According to my current observation, where is the paper towel dispenser
options:
A: To the front-right of me
B: To the front-left of me
C: To the back-left of me
D: To the back-right of me

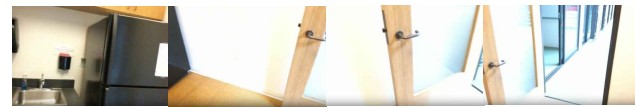

**Model's Output:** "Let's analyze the situation step-by-step based on the video observation:\n\n1. **Initial Observations**:\n   - The video shows a kitchen area with a sink, refrigerator, and cabinets.\n   - A paper towel dispenser is visible in earlier frames. It is mounted to the left of the sink, near the wall.\n\n2. **Orienting the Current View**:\n   - In your current observation frame, you see a water cooler and a counter, which suggests you're looking toward the area where the refrigerator and the paper towel dispenser were in the previous frames.\n   There are a sequence of frames where camera move to right\n3. **Relative Positioning**:\n   - The paper towel dispenser was to the left of the sink and the sink was on the left side of the refrigerator.\n   - In earlier frames, the paper towel dispenser was on the left side when facing the refrigerator.\n\n4. **Conclusion**:\n   - Since you are seeing the water cooler and counter, the sink and paper towel dispenser would be to your left.\n\nGiven this orientation, the correct answer is:\n\n**B. To the front-left of me.**",

**Note:** The camera just rotate towards right direction almost 360 degrees but the models thinks the camera is shifting to the right so the model is unclear about whether the movement is shift or rotation.

*Figure 73.* **Example of camera motion error.** Camera motion error refers to a model's failure to correctly interpret the type or direction of camera movement during a video or scene. This can include confusing rotation with translation, or misunderstanding how the camera's viewpoint has changed over time. As a result, the model may misjudge the relative positions of objects or incorrectly estimate their spatial layout based on an inaccurate perception of motion.

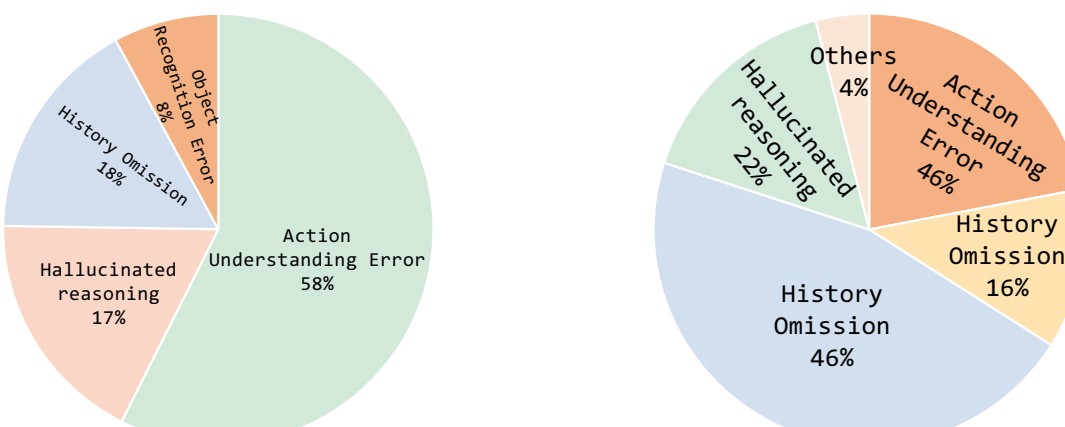

*Figure 74.* **(a) Error distribution of GPT-4o on** *Task Process Reasoning.* (b) Error distribution of GPT-4o on *Next Action Prediction*

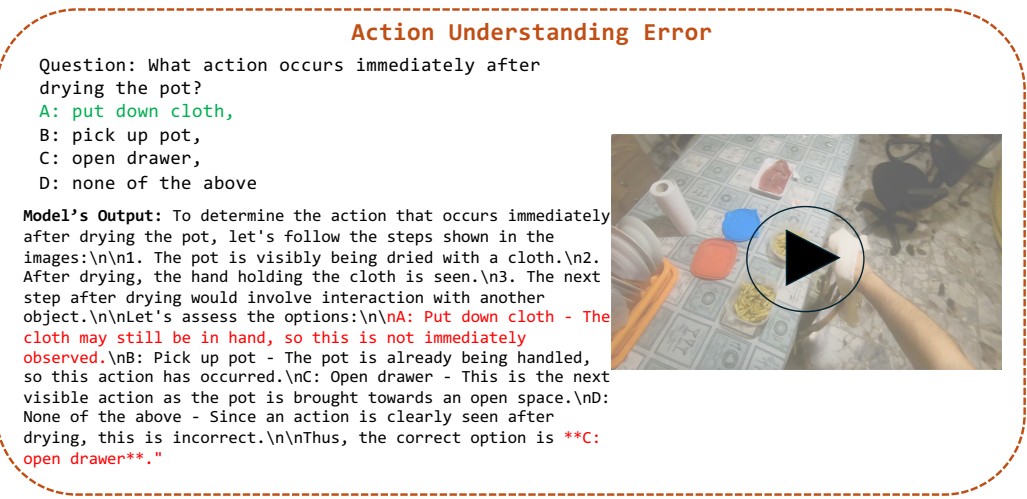

*Figure 75.* **Example of action understanding error.** GPT-4o can not understand the last frames represent the human putting down the cloth.

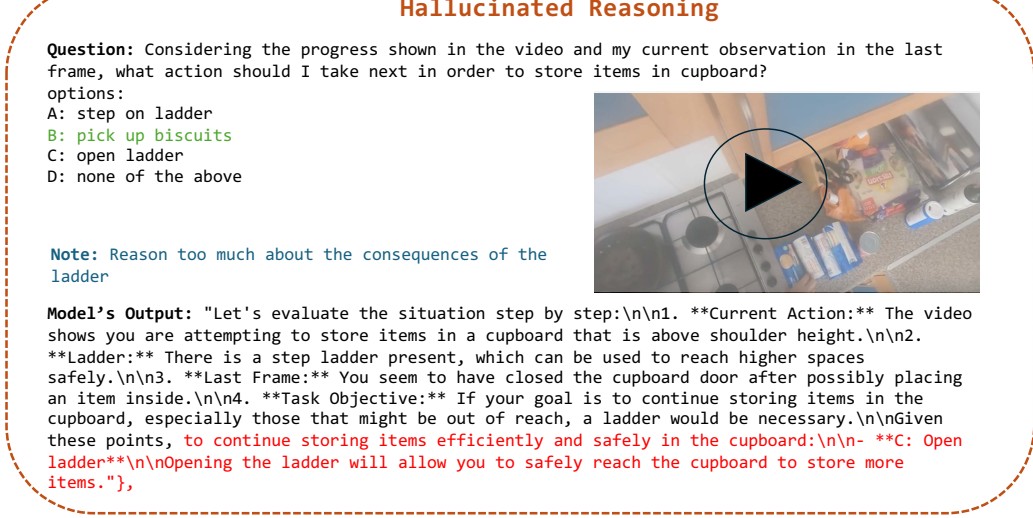

*Figure 76.* **Example of hallucinated reasoning error.** GPT-4o conjuncture about ladder without any references on observations.

## History Omission

**Question:** Considering the progress shown in the video and my current observation in the last frame, what action should I take next in order to wash and cut the pepper?
options:
A: put down pepper on chopping board
B: open the fridge
C: walk to the eggs in fridge
D: none of the above

**Note:** The peper has already been placed on chopping board.

**Model's Output:**"Let's review the situation step by step:\n\n1. **Observation of the Environment:** You are in a kitchen, as indicated by the presence of a chopping board and other kitchen appliances.\n\n2. **Current Task:** Your goal is to wash and cut the pepper.\n\n3. **Items in Hand:** You are holding a yellow pepper and a package of cherry tomatoes.\n\n4. **Location of Pepper:** The pepper should be placed on the chopping board as a preparatory step for cutting.\n\n5. **Requirement for Washing:** Prior to cutting, the pepper needs to be washed, but your hands are currently occupied with both the pepper and tomatoes.\n\n**Conclusion:**\n- **Option A (Put down pepper on chopping board):** Placing the pepper down will free your hands to wash it.\n- **Option B (Open the fridge):** Unnecessary for washing or cutting.\n- **Option C (Walk to the eggs in fridge):** Irrelevant to the task.\n- **Option D (None of the above):** Incorrect, as option A is necessary.\n\nTherefore, the correct action is **A: put down pepper on chopping board.**"

*Figure 77.* **Example of history omission error.** GPT-4o sometimes forgets about what happens in the history.

## Object Recognition Error

**Question:** What action comes directly after 'open counter'?
options:
A: take salt
B: take spoon
C: put plate
D: none of the above

**Note:** it is not counter, it is drawer

**Model's Output:**"To determine the action that comes directly after \"open counter,\" let's break down the sequence of events in the images:\n\n1. **Open counter**: The counter is pulled open.\n2. **Take towel**: A towel is taken from the drawer.\n\nAfter \"open counter,\" the next action is \"take towel.\"\n\nNow, let's match this with the options:\n\nA: take salt – This does not occur directly after opening the counter.\nB: take spoon – This does not occur directly after opening the counter.\nC: put plate – This does not occur directly after opening the counter either.\nD: none of the above – This is correct since none of the given options match the action \"take towel\" directly after \"open counter.\"\n\nTherefore, the correct option is **D: none of the above**."

*Figure 78.* **Example of object recognition error.** GPT-4o mis-recognize drawer as counter.

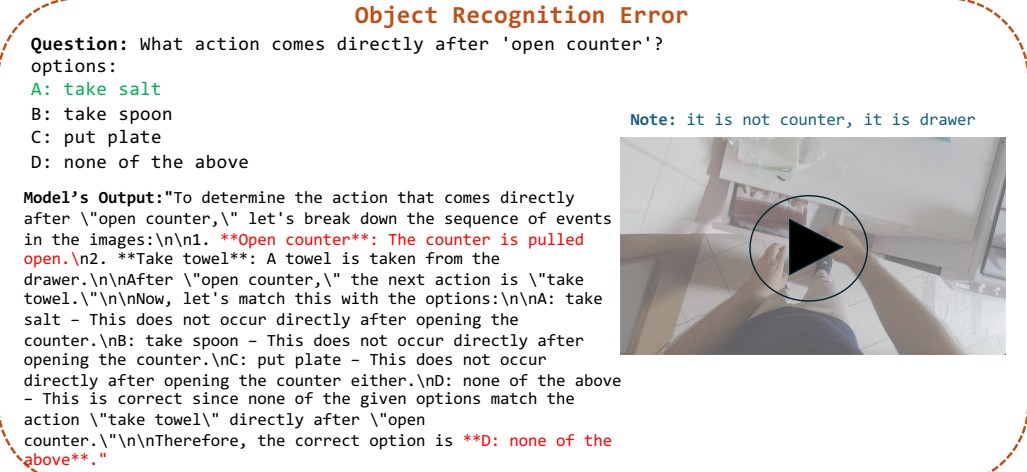

*Figure 79.* **Error distribution of *long-horizon* category.** In *long-horizon* category, the major errors come from where the agent misidentifies the object or predict incorrect trajectory to finish the tasks.

## Pointing Error

**Question:** Identify the blue cup.

**Options:**
A: Yellow Dot,
B. Blue Dot,
C. Red Dot,
D: None of the above

**Model's Output: B. Blue Dot**

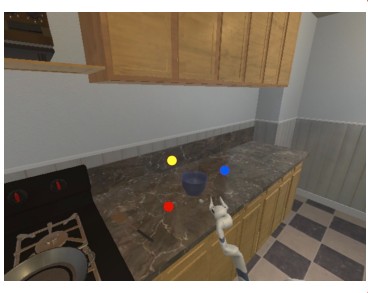

*Figure 80.* **Pointing error example in *long-horizon* category.**

## Trajectory Error

**Question:** Identify the correct trajectory to pick up the blue cup on the table.

**Options:**
A: Yellow arrow.
B. Blue arrow
C. Red arrow
D: None of the above

**Model's Output: C. Red Arrow**

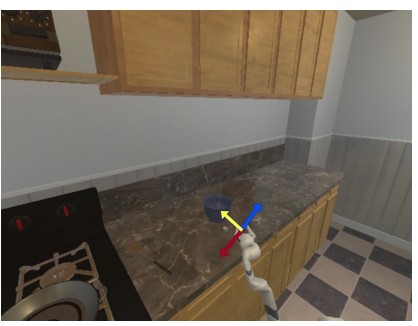

*Figure 81.* **Trajectory error example in *long-horizon* category.**

## Spatial Reasoning Error

**Question:** Where is the vegetable?

**Options:**
A: To the back-left of me,
B. To the front-right of me,
C. To the front-left of me.
D: None of the above

**Model's Output:** The vegetable is to the front-right of me. The correct option is B. To the front-right of me.

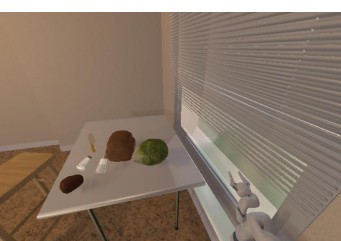

*Figure 82.* **Spatial error example in *long-horizon category*.**

# I. Benchmark Examples and Evaluation Prompts

## I.0.1. EXAMPLES

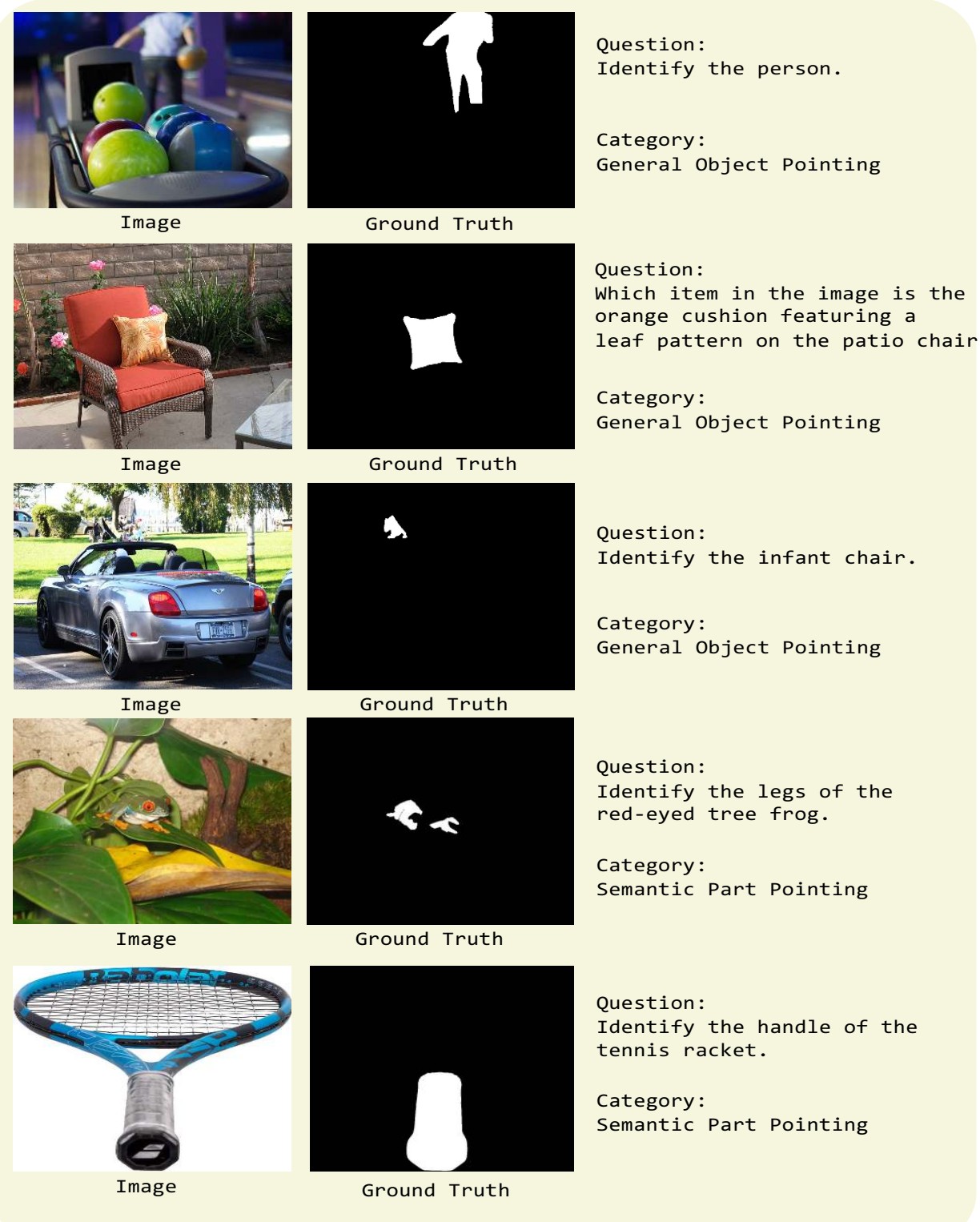

*Figure 83.* **Example questions in BEAR.** We select some questions from *General Object Pointing*, *Spatial Relationship Pointing* and *Semantic Part Pointing*.

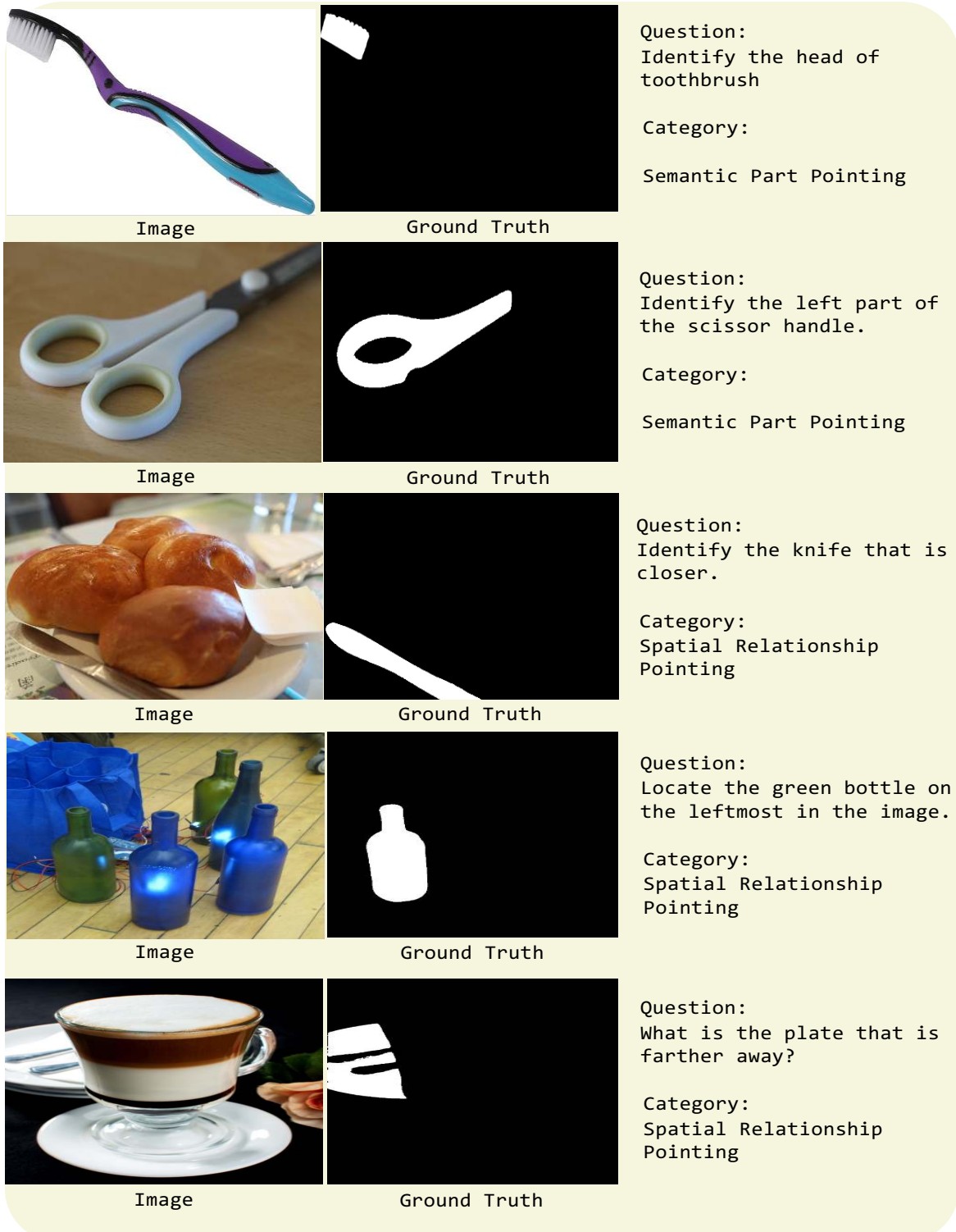

*Figure 84.* **Example questions in BEAR.** We select some questions from *General Object Pointing*, *Spatial Relationship Pointing* and *Semantic Part Pointing*.

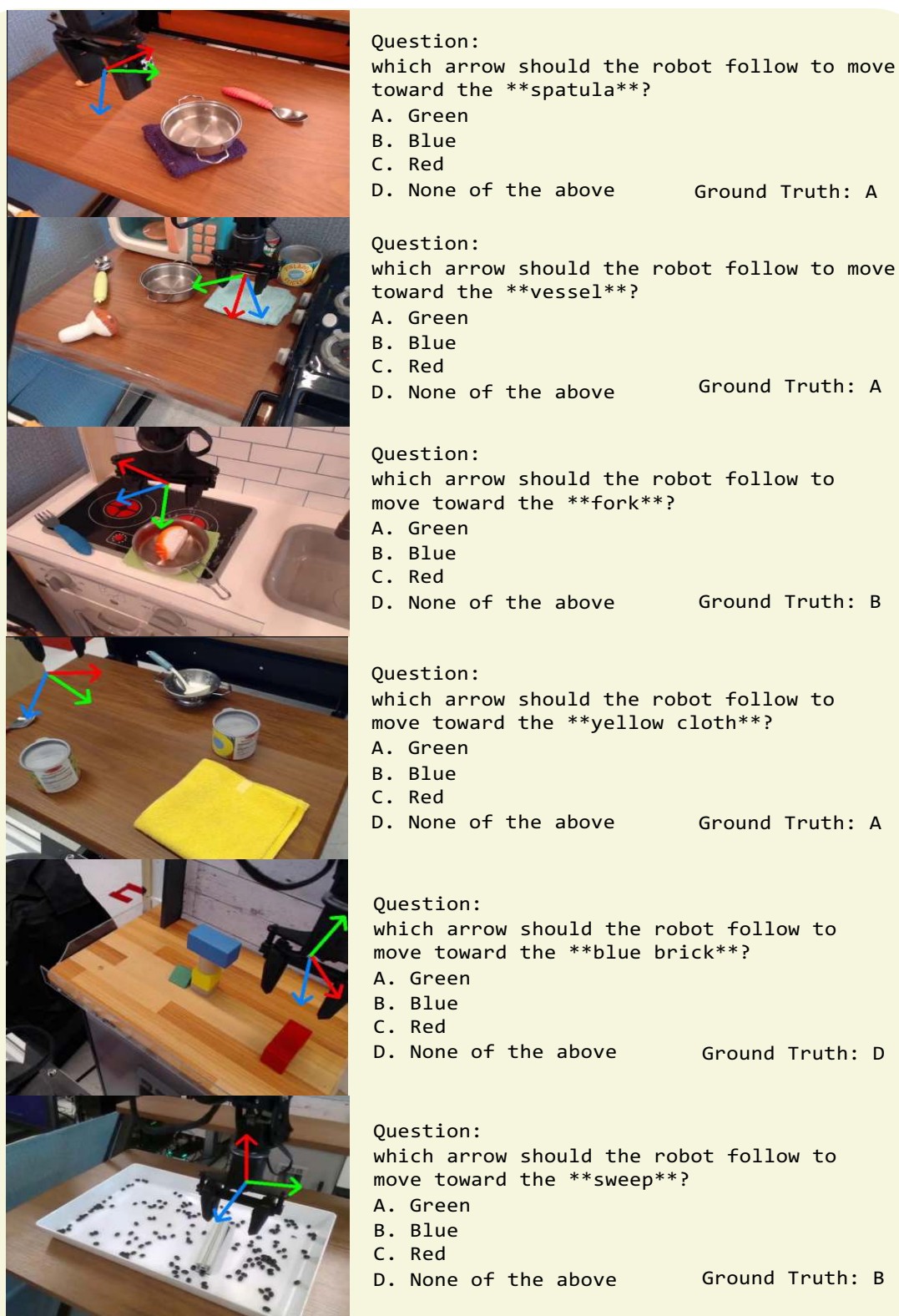

Question:
which arrow should the robot follow to move toward the **spatula**?
A. Green
B. Blue
C. Red
D. None of the above      Ground Truth: A

Question:
which arrow should the robot follow to move toward the **vessel**?
A. Green
B. Blue
C. Red
D. None of the above      Ground Truth: A

Question:
which arrow should the robot follow to move toward the **fork**?
A. Green
B. Blue
C. Red
D. None of the above      Ground Truth: B

Question:
which arrow should the robot follow to move toward the **yellow cloth**?
A. Green
B. Blue
C. Red
D. None of the above      Ground Truth: A

Question:
which arrow should the robot follow to move toward the **blue brick**?
A. Green
B. Blue
C. Red
D. None of the above      Ground Truth: D

Question:
which arrow should the robot follow to move toward the **sweep**?
A. Green
B. Blue
C. Red
D. None of the above      Ground Truth: B

*Figure 85.* **Example questions in BEAR.** We select some questions from *Gripper Trajectory Reasoning*.

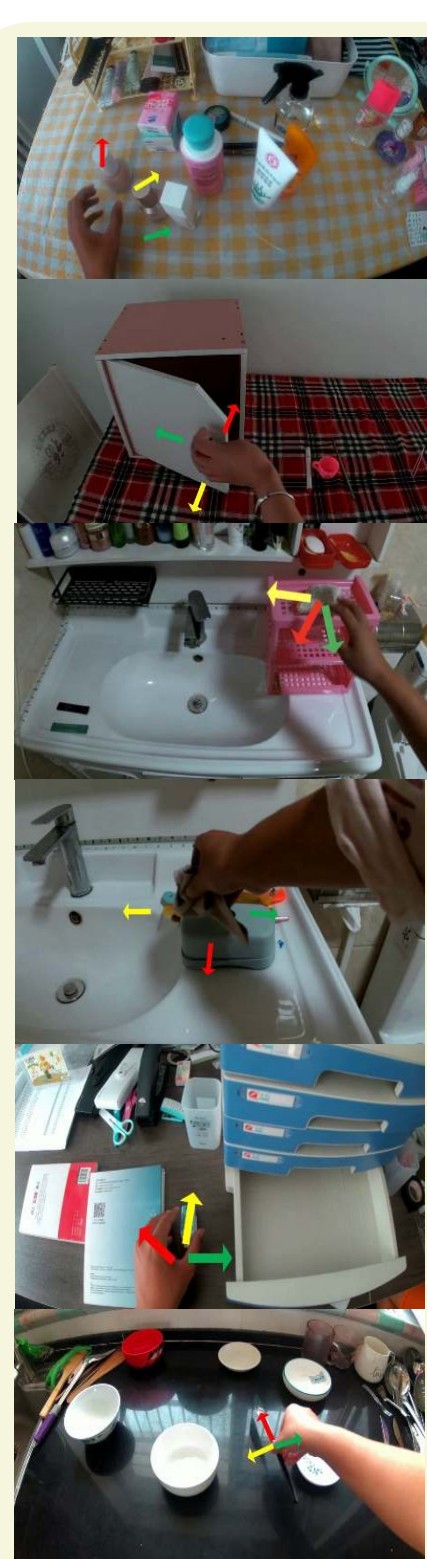

Question:
which arrow should the hand follow to move
toward the **watering can**?
A. Red
B. Green
C. Yellow
D. None of the above          Ground Truth: C

Question:
Which direction should you move in to close
the cabinet?
A. Red
B. Green
C. Yellow
D. None of the above          Ground Truth: A

Question:
which direction is the hand most likely
to place the dish cloth on the black
rack?
A. Red
B. Green
C. Yellow
D. None of the above          Ground Truth: C

Question:
which arrow indicates the correct
direction to clean the surface of this
soap box?
A. Green
B. Blue
C. Red
D. None of the above          Ground Truth: A

Question:
which direction is the hand most likely
to place the blue stapler inside the open
drawer on the right of the hand?
A. Red
B. Green
C. Yellow
D. None of the above          Ground Truth: B

Question:
which direction is the hand most likely
to move if you want to use the knife to
stab the small white plate?
A. Green
B. Blue
C. Red
D. None of the above          Ground Truth: C

*Figure 86.* **Example questions in BEAR.** We select some questions from *Human Hand Trajectory Reasoning*.

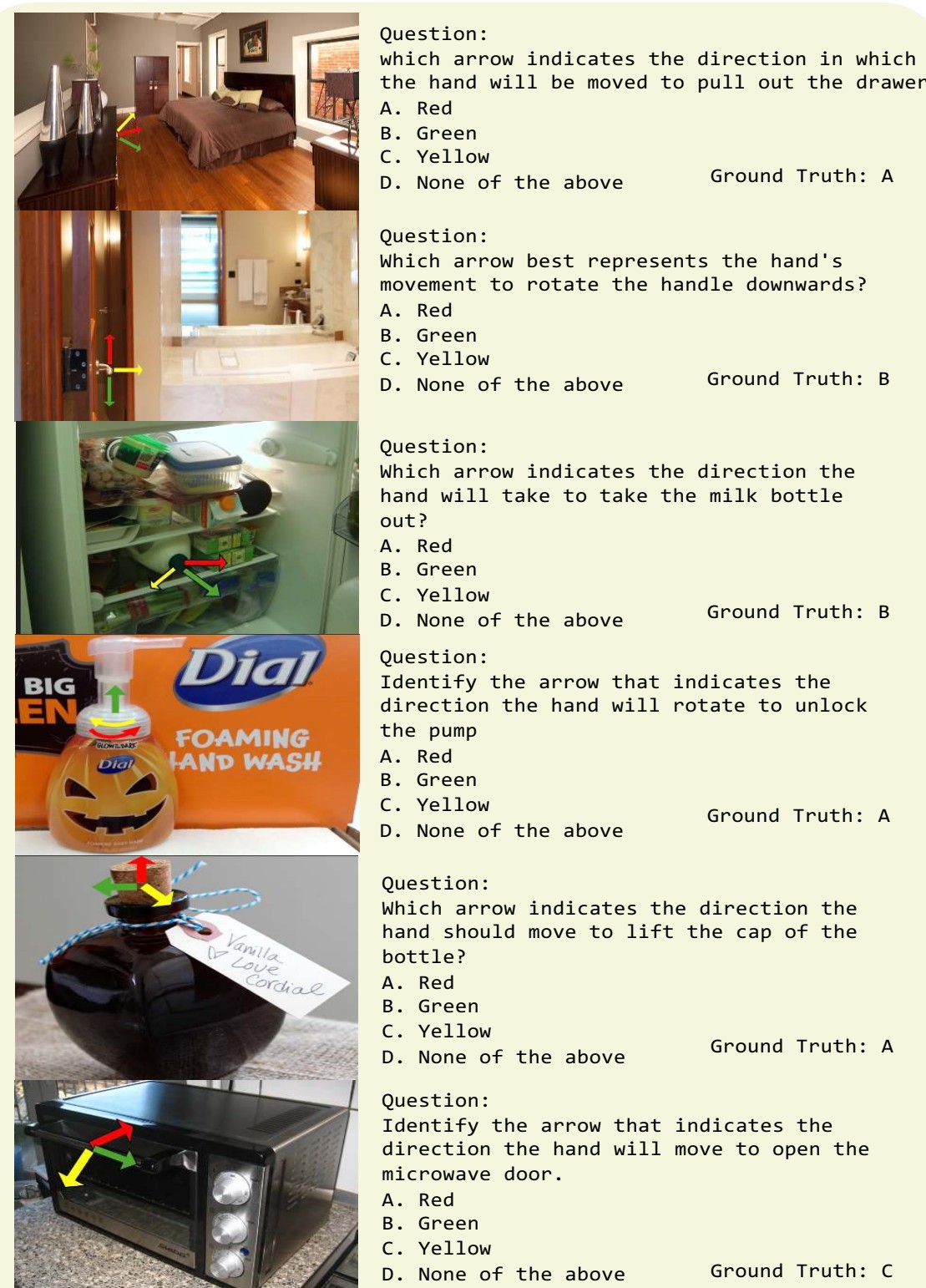

Question:
which arrow indicates the direction in which the hand will be moved to pull out the drawer?
A. Red
B. Green
C. Yellow
D. None of the above        Ground Truth: A

Question:
Which arrow best represents the hand's movement to rotate the handle downwards?
A. Red
B. Green
C. Yellow
D. None of the above        Ground Truth: B

Question:
Which arrow indicates the direction the hand will take to take the milk bottle out?
A. Red
B. Green
C. Yellow
D. None of the above        Ground Truth: B

Question:
Identify the arrow that indicates the direction the hand will rotate to unlock the pump
A. Red
B. Green
C. Yellow
D. None of the above        Ground Truth: A

Question:
Which arrow indicates the direction the hand should move to lift the cap of the bottle?
A. Red
B. Green
C. Yellow
D. None of the above        Ground Truth: A

Question:
Identify the arrow that indicates the direction the hand will move to open the microwave door.
A. Red
B. Green
C. Yellow
D. None of the above        Ground Truth: C

*Figure 87.* **Example questions in BEAR.** We select some questions from *Object Trajectory Reasoning*.

Which description of following about the white plastic cutting board is true
according to the video given?
A. Behind the dish rack near the sink.
B. On the stove beside the pots
C. Hanging on the wall above the counter
D. None of the above                    Ground Truth: A

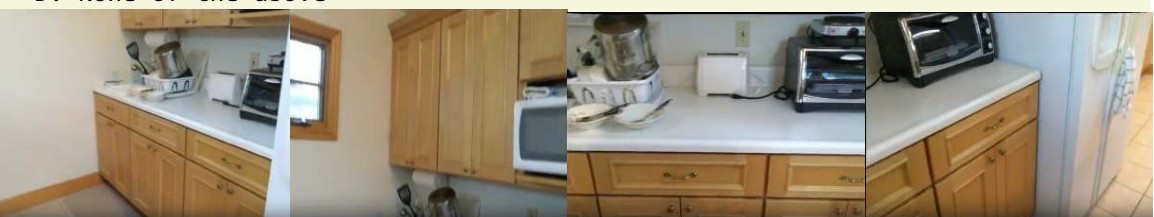

Which description of following about the mini soccer ball toy is true
according to the video given?
A. On the top left shelf inside the yellow bin
B. On the floor near the white trash bin
C. On the blue stool next to the table
D. None of the above                    Ground Truth: A

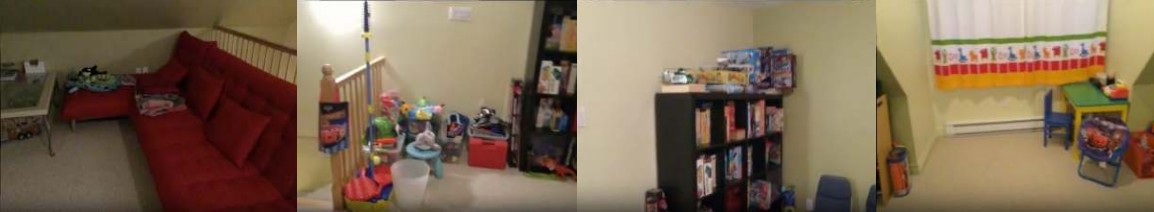

Which description of following about the large blue bag is true according to
the video given?
A. Next to the television stand against the wall
B. On top of the glass coffee table
C. Beside the red sofa
D. None of the above                    Ground Truth: A

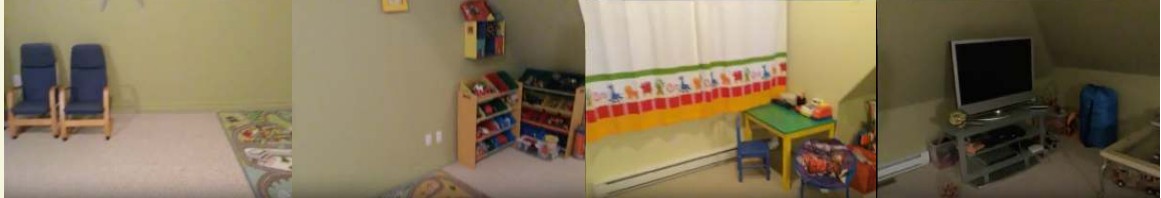

Which description of following about the book next to the plant is true
according to the video given?
A. On the floor near the gray carpet
B. On the sofa near the yellow cushion
C. On the black shelf
D. None of the above                    Ground Truth: C

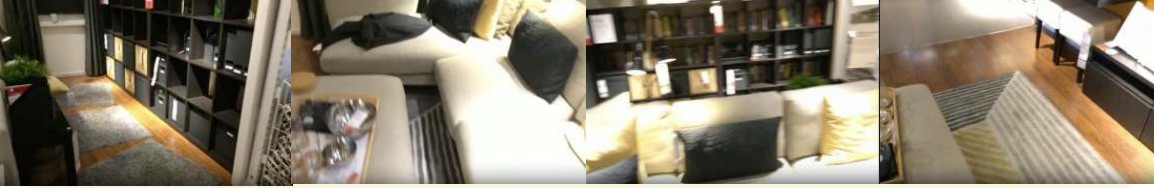

*Figure 88.* **Example questions in BEAR.** We select some questions from *Object Localization*.

According to the current observation, where is the kitchen counter?
A. To the front-right of me.
B. To the front-left of me.
C. To the back-left of me.
D. To the back-right of me.                    Ground Truth: B
                    History Video                    Current Observation

Where is the coffee table?
A. To the front-right of me.
B. To the front-left of me.
C. To the back-left of me.
D. To the back-right of me.                    Ground Truth: C
                    History Video                    Current Observation

Where is the toilet?
A. To the front-right of me.
B. To the front-left of me.
C. To the back-left of me.
D. To the back-right of me.                    Ground Truth: D
                    History Video                    Current Observation

Where is the blue box?
A. To the front-right of me.
B. To the front-left of me.
C. To the back-left of me.
D. To the back-right of me.                    Ground Truth: B
                    History Video                    Current Observation

*Figure 89.* **Example questions in BEAR.** We select some questions from *Relative Direction*.

You want to navigate to the toilet. You will perform the following actions (Note: for each [please fill in], choose either 'turn back,' 'turn left,' or 'turn right.'): 1. Go forward until the TV 2. [please fill in] 3. Go forward until the shower 4. [please fill in] 5. Go forward until the toilet. You have reached the final destination.

A. Turn Back, Turn Left
B. Turn Left, Turn Left
C. Turn Left, Turn Right
D. Turn Right, Turn Right

Ground Truth: C

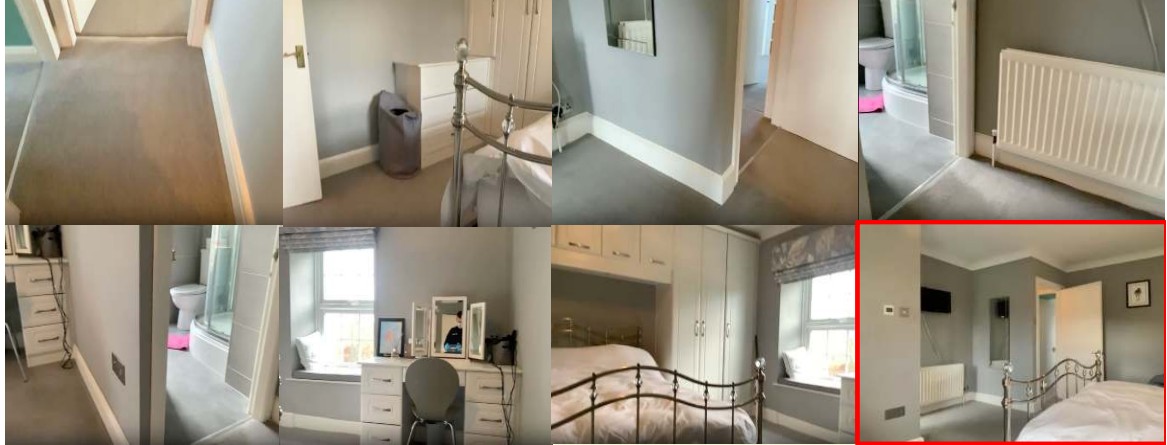

You want to navigate to the trash bin. You will perform the following actions (Note: for each [please fill in], choose either 'turn back,' 'turn left,' or 'turn right.'): 1. [please fill in] 2. Go forward until the cabinet 3. [please fill in] 4. Go forward until the trash bin is on your right. You have reached the final destination.

A. Turn Left, Turn Left
B. Turn Right, Turn Left
C. Turn Back, Turn Left
D. Turn Right, Turn Right

Ground Truth: B

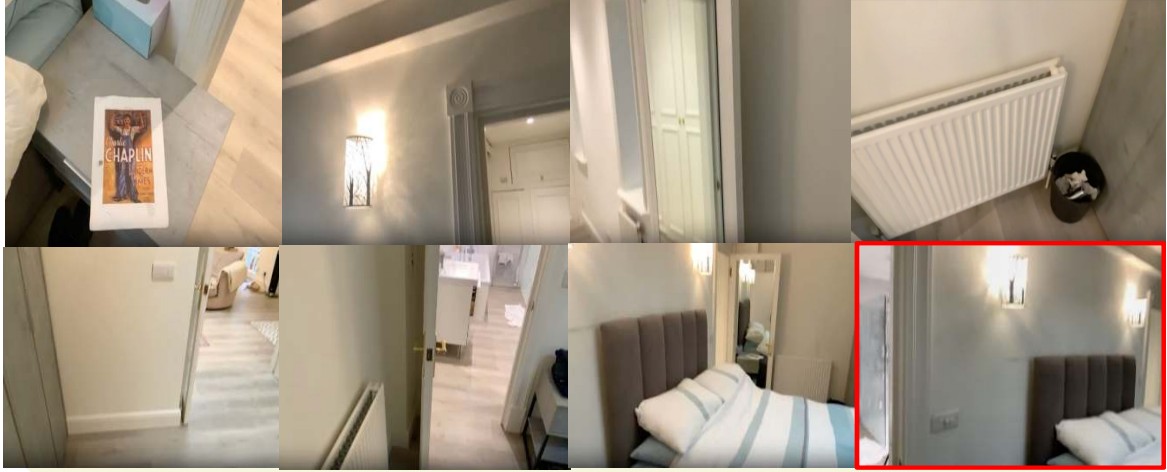

*Figure 90.* **Example questions in BEAR.** We select some questions from *Path Planning*.

```
Which action does not happen before 'put away raisins'
A. open drawer
B. pour cereal
C. open fridge
D. none of the above                    Ground Truth: C
```

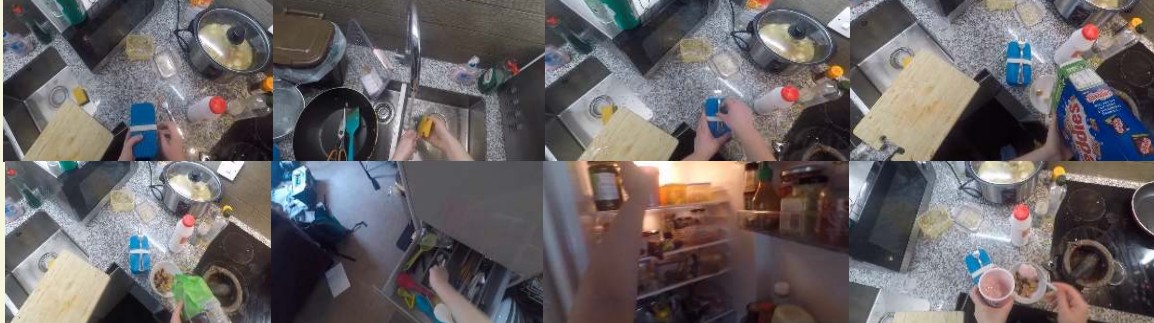

```
Which of the following actions is not performed after 'pick up plate'?
A. wipe hob
B. put down plate
C. turn off tap
D. none of the above                    Ground Truth: C
```

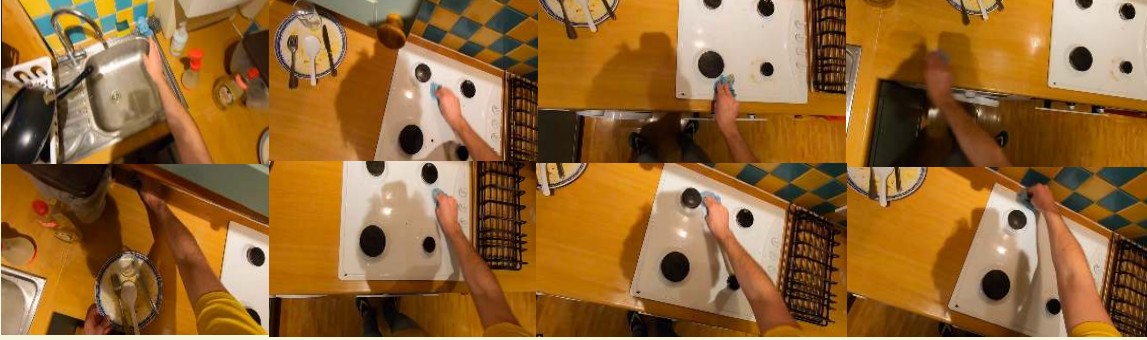

```
What action occurs immediately after drying the pot?
A. put down cloth
B. pick up pot
C. open drawer
D. none of the above                    Ground Truth: A
```

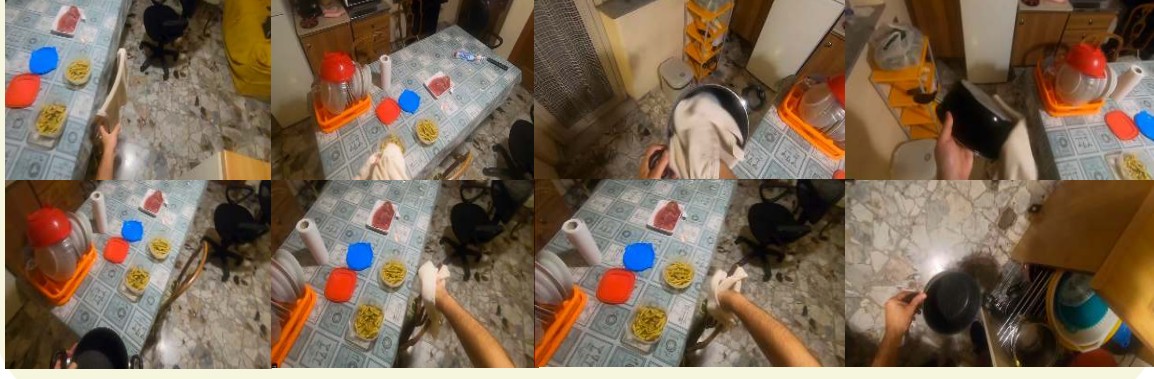

*Figure 91.* **Example questions in BEAR.** We select some questions from *Task Process Reasoning*.

Considering the progress shown in the video and my current observation in the last frame, what action should I take next in order to prepare meat for cooking?
A. cut meat
B. throw cover
C. walk to the trash bin
D. none of the above

Ground Truth: A

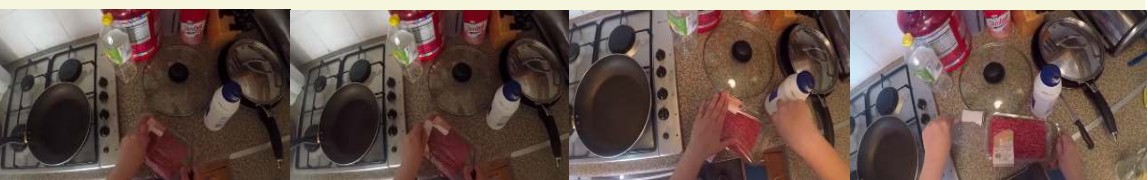

Considering the progress shown in the video and my current observation in the last frame, what action should I take next in order to fold and put away bag?
A. close drawer
B. pick up bag
C. walk to the drawer
D. none of the above

Ground Truth: A

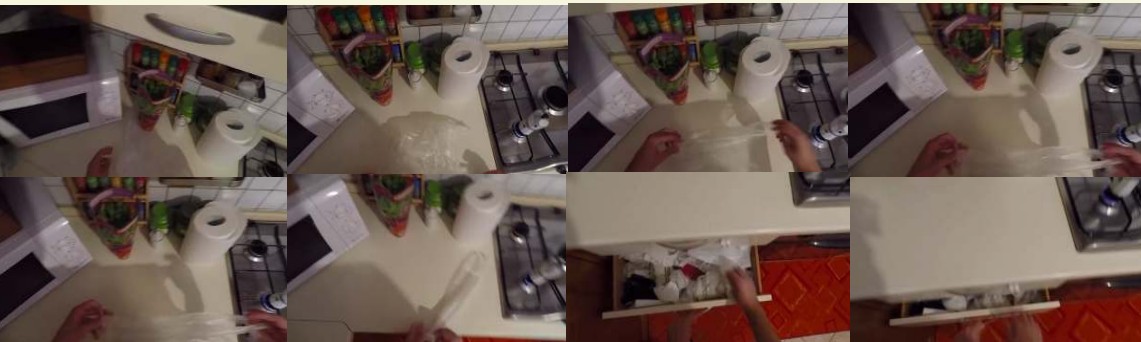

Considering the progress shown in the video and my current observation in the last frame, what action should I take next in order to wash and rinse various kitchen utensils and dishes?
A. wash spoon
B. walk to the measuring cup
C. put down measuring cup
D. none of the above

Ground Truth: D

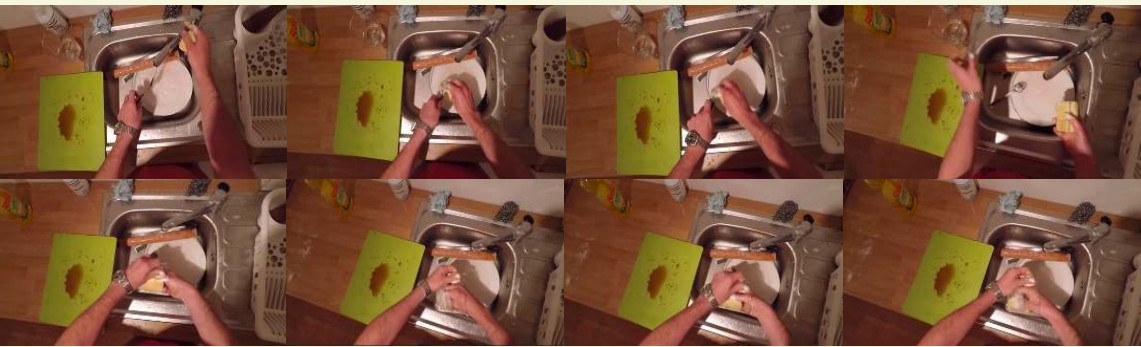

*Figure 92.* **Example questions in BEAR.** We select some questions from *Next Action Prediction*.

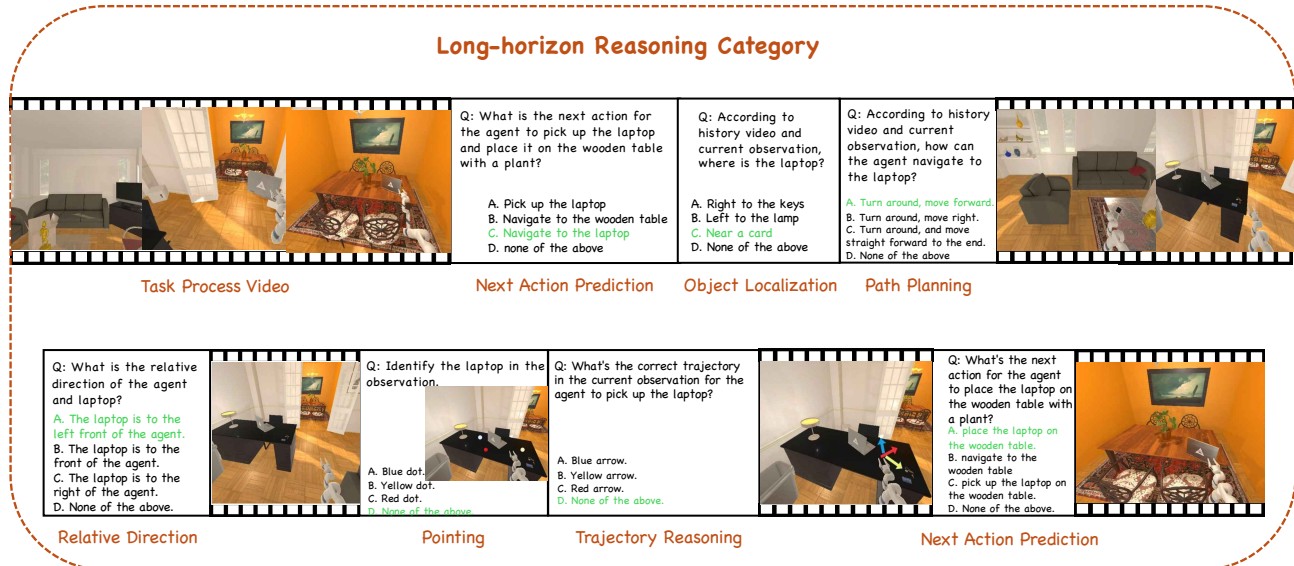

*Figure 93.* **Example questions in BEAR.** We select some questions from *Long-horizon* Category.

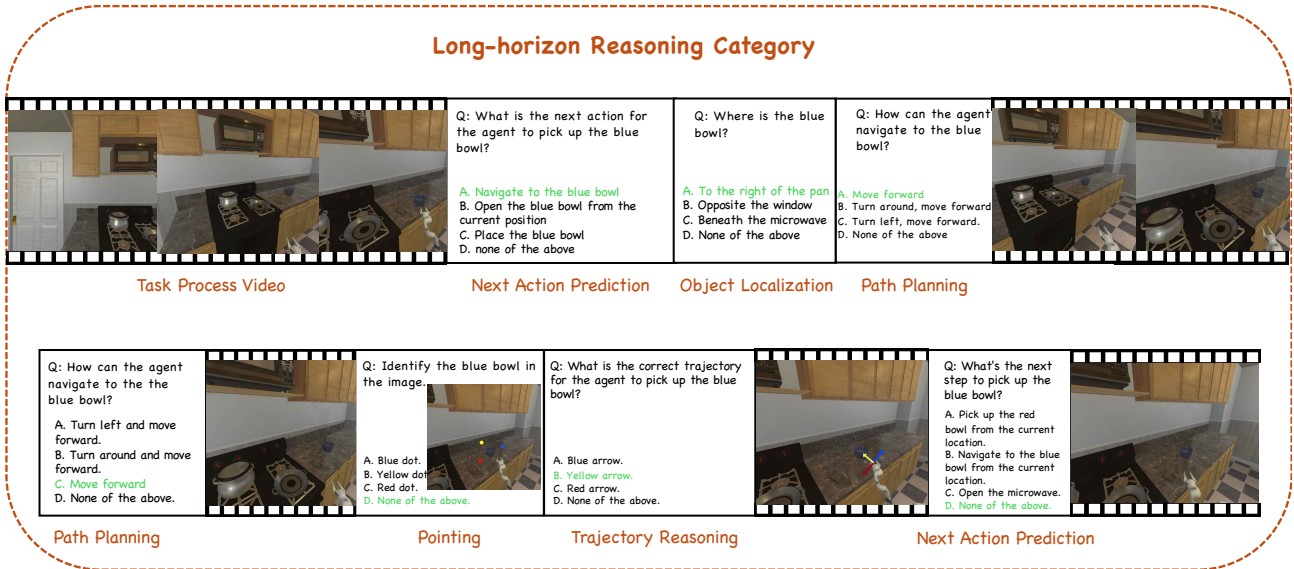

*Figure 94.* **Example questions in BEAR.** We select some questions from *Long-horizon* Category.

### I.0.2. FULL PROMPTS

---

*Pointing* Category Prompt

```
Identify the [Descriptions].
Please directly output the pixel location of the target point.
Format your answer as (x, y), where:
- x is the horizontal coordinate (left → right)
- y is the vertical coordinate (top → bottom)
Both x and y must be normalized to the range [0, 1] as floating-point values,
indicating the relative position of the point within the image.
```

---

*Bounding Box* Category Prompt

```
Identify the [Descriptions].
Please directly output the bounding box.
Format your answer as (x1, y1, x2, y2), where:
- (x1, y1) is the top-left corner of the bounding box
- (x2, y2) is the bottom-right corner of the bounding box.
- x represents the horizontal coordinate (left → right)
- y represents the vertical coordinate (top → bottom).
All coordinates must be normalized to the range [0, 1] as floating-point values,
indicating the relative bounding box location within the image.
```

---

*Gripper Trajectory Reasoning* Prompt

```
The image shows the current location of the robot hand.
There are three arrows pointing in different directions.
Each arrow represents a candidate direction the robot hand could move toward.
Which arrow should the robot follow to move toward the [Descriptions]?
[Options]
Please directly output the correct option.
```

---

*Human Hand Trajectory Reasoning* Prompt

```
Assuming a human hand is [Descriptions], indicate the arrow that shows the direction
in which the hand will move to pull the drawer door.
[Options]
Please directly output the correct option.
```

---

*Object Trajectory Reasoning* Prompt

```
Identify the arrow that indicates [Descriptions]
[Options]
Please directly output the correct option.
```

*Object Localization* Prompt

```
Please watch the following video and answer the question.
[Options]
indicating the relative position of the point within the image.
```

*Path Planning* Prompt

```
This is the current video.
[Video]
This is your current observation.
[Image]
You need to navigate to [Descriptions] by performing the following actions.
[Options]
Please directly output the correct option.
```

*Relative Direction* Prompt

```
This is the current video.
[video]
This is your current observation.
[image]
Where is [Descriptions]?
[Options]
Please directly output the correct option.
```

*Task Progress Reasoning* Prompt

```
This is the current video.
[Video]
[Question]
[Options]
Please directly output the correct option.
```

*Next Action Prediction* Prompt

```
This is the current video.
[Video]
Considering the process shown in the video and my current observation in the last
frame, [Descriptions]
[Options]
Please directly output the correct option.
```

*Table 7.* **BEAR-Agent results on BEAR.** We evaluate the impact of BEAR-Agent on the two strongest models—GPT-5 (OpenAI, 2025a) among proprietary systems and InternVL3-14B (Zhu et al., 2025) among open-source models. For comparison, we also report their performance under *Direct* and *CoT* prompting, as well as *one-shot* and *few-shot* in context learning as baseline methods for comparisons.

| | Format | Pointing | | | Bounding Box | | | Task Planning | |
|---|---|---|---|---|---|---|---|---|---|
| | | GEN | SPA | PRT | GEN | SPA | PRT | PRG | PRD |
| Random Choice | | - | - | - | - | - | - | 25 | 25 |
| Human | | 95.50 | 92.00 | 93.50 | 0.830 | 0.770 | 0.820 | 87.50 | 92.00 |
| State-of-the-art open-source model on BEAR | | | | | | | | | |
| InternVL3-14B (Zhu et al., 2025) | merged | 37.94 | 27.78 | 32.80 | 0.304 | 0.258 | 0.276 | 41.00 | 33.00 |
| – w/ one-shot in context learning | merged | 38.53 | 28.34 | 33.33 | 0.312 | 0.268 | 0.287 | 42.00 | 34.00 |
| – w/ few-shot in context learning | merged | 39.41 | 28.98 | 34.31 | 0.321 | 0.275 | **0.297** | 41.00 | 34.00 |
| – w/ Chain-of-Thought prompting | merged | 27.94 | 21.90 | 26.92 | 0.265 | 0.214 | 0.213 | **44.00** | 31.67 |
| – w/ BEAR-Agent | merged | **47.06** | **31.85** | **34.97** | **0.347** | **0.294** | 0.269 | 41.67 | **36.33** |
| State-of-the-art proprietary model on BEAR | | | | | | | | | |
| GPT-5 (OpenAI, 2025a) | sequential | 70.00 | 63.69 | 54.90 | 0.411 | 0.326 | 0.352 | 59.67 | 61.00 |
| – w/ one-shot in context learning | sequential | 70.59 | 64.01 | 55.23 | 0.415 | 0.330 | 0.357 | 60.33 | 61.33 |
| – w/ few-shot in context learning | sequential | 71.18 | 64.65 | 55.88 | 0.421 | 0.337 | 0.363 | 61.00 | 62.00 |
| – w/ Chain-of-Thought prompting | sequential | 67.35 | 57.19 | 64.01 | 0.406 | 0.321 | 0.370 | 58.67 | 61.33 |
| – w/ BEAR-Agent | sequential | **84.12** | **75.16** | **64.05** | **0.573** | **0.418** | **0.447** | **61.67** | **71.67** |

| | Format | Trajectory Reasoning | | | Spatial Reasoning | | | Long-horizon | Avg |
|---|---|---|---|---|---|---|---|---|---|
| | | GPR | HND | OBJ | LOC | PTH | DIR | | |
| Random Choice | | 25 | 25 | 25 | 25 | 28 | 25 | 25 | - |
| Human | | 96.50 | 94.00 | 89.00 | 94.50 | 83.50 | 88.50 | 92.50 | 89.40 |
| State-of-the-art open-source model on BEAR | | | | | | | | | |
| InternVL3-14B (Chen et al., 2024) | merged | 51.28 | 49.49 | 31.43 | 43.00 | 28.02 | 21.33 | **28.57** | 33.93 |
| – w/ one-shot in context learning | merged | 48.72 | 52.53 | 29.00 | 39.74 | 31.88 | 18.67 | **28.57** | 33.38 |
| – w/ few-shot in context learning | merged | **54.49** | 47.14 | 34.67 | **46.25** | 25.12 | 24.33 | 25.71 | 34.98 |
| – w/ Chain-of-Thought prompting | merged | 44.87 | 39.39 | **39.43** | 41.37 | **34.69** | 24.66 | 0 | 26.88 |
| – w/ BEAR-Agent | merged | 53.85 | **52.86** | 37.00 | 43.65 | 31.40 | **26.00** | **28.57** | **36.24** |
| State-of-the-art proprietary model on BEAR | | | | | | | | | |
| GPT-5 (OpenAI, 2025a) | sequential | 66.99 | 67.34 | 49.67 | 72.31 | 50.24 | 47.00 | 40.00 | 52.17 |
| – w/ one-shot in context learning | sequential | 67.31 | 67.68 | 49.33 | 72.64 | 50.72 | 46.67 | 40.00 | 54.43 |
| – w/ few-shot in context learning | sequential | 67.95 | 68.35 | 49.00 | **73.29** | 51.69 | 46.33 | **42.86** | 54.98 |
| – w/ Chain-of-Thought prompting | sequential | 65.38 | 68.35 | 52.00 | **73.29** | 47.83 | 50.00 | 34.29 | 52.78 |
| – w/ BEAR-Agent | sequential | **84.62** | **80.81** | **62.67** | 71.01 | **52.17** | **56.33** | **42.86** | **61.29** |

# J. BEAR-Agent

### J.0.1. DEFINITION

BEAR-Agent is a multimodal conversable agent that interacts with Multimodal Large Language Models (MLLMs) through a turn-based dialog framework. Built upon the AutoGen framework, the agent leverages GPT-4V (OpenAI et al., 2023) as its backbone. It operates asynchronously, engaging in multi-round interactions. At initialization, the agent generates a category-specific prompt that describes both the task and the tools available for solving it. Upon receiving a response from the MLLM—often containing code to invoke external tools (e.g., object detection functions)—the agent executes the code and feeds the results back into the next round of interaction. This iterative process continues until the MLLM produces a final answer and issues a termination signal, at which point the agent ends the conversation.

Our BEAR-Agent is motivated from the related works (Hu et al., 2024; Chow et al., 2025) using visual prompting and foundation models to enhance the MLLMs' visual abilities. **The difference of our work lies in its embodied domain.** We make sure each component of our agent design is tailored to a category of embodied tasks. Motivated by the idea of (Hu et al., 2024), we use sketching as our visual tools for visual abilities enhancement. We design our category-specific prompt activation module for our tasks. We provide example details in Appendix J.0.2. We also augment the agent with a suite of external visual tools. Specifically, we integrate powerful foundation models such as Grounding DINO (Liu et al., 2024b) and Set-of-Mark (SoM) (Yang et al., 2023a), which the MLLM can invoke via function calls. In addition to foundation

models, we implement specialized visual utility functions—for instance, for detecting and extending trajectory arrows in images—enabling models to better infer directionality.

Moreover, our analysis reveals that many trajectory reasoning errors stem from the model's lack of embodied knowledge, such as how to apply the right-hand rule to open a bottle cap from different viewpoints. To mitigate this, we incorporate a knowledge base that provides such procedural information to improve reasoning accuracy. For spatial reasoning, we find that many failures arise from the model's inability to align information across multiple frames. To support this, we introduce a semantic scene reconstruction function that encourages the model to identify object correspondences across temporal frames.

In our experiments, we evaluate BEAR-Agent using two state-of-the-art models from both proprietary and open-source families: GPT-5 (OpenAI, 2025a) and InternVL3-14B (Zhu et al., 2025). Detailed performance results are provided in Table 7. In the meantime, we also provide the prompt of our BEAR-Agent in Appendix I.0.2.

J.0.2. PROMPTS

We provide category-specific module prompts as follows. If we provide the full detailed prompts it will be too long to read, so we select some pieces of it.

**Knowledge base.** is built to enhance agents' understandings on the ideal motion to perform an action. It can be further expanded to a system with more trajectory knowledge.

---

Knowledge Base

```
Please pay attention to the following knowledge that can help you correctly answer
the questions:

Knowledge-based Memory:

1.  For tasks like the correct trajectory to open the drawer, if you are facing
exactly the front-view of the drawer, the correct trajectory should pointing
horizontally downwards.  If you are facing the side-view of the drawer or other
electronic device, the correct trajectory should be vertical to the front-facing
edge of the drawer, cabinet, etc.  You should also observe the side edge of the
drawer, cabinet, the correct trajectory should be about parallel to it.  This is
very important, because when facing the object using the side-view, the correct
trajectory should also be side-view.  The rule is not always right, you do need to
use your common sense to choose the correct trajectory.
2.  For tasks about rotating the handle, the handle can only be rotated around the
central axis.  If you are not rotating the handle around the central axis, it will
sometimes cause damage to the handle.
3.  For tasks about lifting the object, the correct trajectory will always pointing
upwards.  For tasks about pressing down the trajectory, the correct trajectory will
always pointing downwards.
4.  For tasks about opening the lid of the bottle, the correct trajectory to unlock
the bottle should be counterclockwise, the correct trajectory to lock the bottle
should be clockwise.  Of course, counterclockwise and clockwise is the relative
direction when you are in the top view observation.  When you are facing the object
in its front view, the correct trajectory to unlock the lid will be pointing right,
the correct trajectory to lock the lid will be pointing left.
5.  Some questions you may need to use your 3D imagination to point out the correct
trajectory.
6.  For tasks about opening the door, some doors you need to push or pull to make it
open, but some doors are sliding, you need to move it parallel to the door surface
to open it.  If the door in the image is already open, and you are told to choose
the trajectory that will make the door open even more, the correct trajectory should
be the trajectory that is roughly parallel to the door surface, and pointing away
from the door hinge.  If you are told to choose the trajectory that will make the
door close, the correct trajectory should be the trajectory that is parallel to the
door surface, and pointing towards the door hinge.
7.  For pressing down the button, you should select the trajectory that is vertical
to the button surface.
```

---

**Trajectory Reasoning prompt activation.**   In our *Trajectory Reasoning* category, we provide the following prompts describing how to use tools to solve the trajectory reasoning questions.

---

**Trajectory Reasoning Prompt Activation**

**Task Overview**

```
This category you will face the trajectory reasoning task, the key for trajectory
reasoning is to identify the correct object and identify which color of arrow can
lead to the correct trajectory to finish the task.
If you cannot find the correct object in the image, or you are not sure where
the extended trajectory will lead to, here I give you some of the tools which can
help with correct object detection in the crowd of object, also the extend arrow
tools.  where you can extend the arrow with the color of you want to see if the
arrow can reach to the target object.
If you are facing the object trajectory reasoning task related with trajectory
reasoning, here are some tools that can help you.  All are python codes.  They are
in tools.py and will be imported for you.
```

**Coordinate System**

```
The images has their own coordinate system.  The upper left corner of the image
is the origin (0, 0).  All coordinates are normalized, i.e., the range is [0, 1].
All bounding boxes are in the format of [x, y, w, h], which is a python list.
x is the horizontal coordinate of the upper-left corner of the box, y is the
vertical coordinate of that corner, w is the box width, and h is the box height.
Notice that you, as an AI assistant, is not good at locating things and
describe them with coordinate.  You can use tools to generate bounding boxes.
You are also not good at answering questions about small visual details in
the image.  You can use tools to zoom in on the image to see the details.
```

Trajectory Reasoning Prompt Activation

Below are the tools in tools.py:

```python
def detection(image, objects):
    """Object detection using Grounding DINO model.
    It returns the annotated image and the bounding boxes
    of the detected objects.

    The text can be simple noun, or simple phrase
    (e.g., 'bus', 'red car'). Cannot be too hard
    or the model will break.

    The detector is not perfect, it may wrongly detect
    objects or miss some objects.

    Args:
        image (PIL.Image.Image): the input image
        objects (List[str]): a list of objects to detect.
                             Each object should be a simple noun
                             or a simple phrase.
    Returns:
        output_image (AnnotatedImage): the original image,
        annotated with bounding boxes
        processed_boxes (List): list the bounding boxes
        of the detected objects

    Example:
        image = Image.open("sample_img.jpg")
        output_image, boxes = detection(image, ["bus"])
        display(output_image.annotated_image)
    """

def extend_arrow_color(img, color="red"):
    """Extend the arrow of a specified color in the image.
    This is a core function within the trajectory
    reasoning pipeline.

    The function returns an image with the extended
    arrow overlaid as a **yellow** dashed line.

    Args:
        img (PIL.Image.Image): the input image
        color (str, optional): the color of the extended arrow.
                      Defaults to "red".
                      Choose from "red", "blue",
                      "green", "yellow"
    Returns:
        output_image (PIL.Image.Image): the original

        image annotated with the extended arrow

    Example:
        image = Image.open("sample_img.jpg")
        output_image = extend_arrow_color(image, color="red")
        display(output_image)
    """
```
95

Trajectory Reasoning Prompt Activation

**Goal**
Based on the above tools, I want you to reason about how to solve the
USER REQUEST and generate the actions step by step (each action is a python
jupyter notebook code block) to solve the request.
You may need to use the tools above to process the images and make
decisions based on the visual outputs of the previous code blocks.
Please use the detection function if you can not find the target object
in the USER REQUEST by yourself or you are not sure if you are correct or
not.
Or the extended trajectory is very near to the target object, it indicates
the color of trajectory should be considered as correct option.
Please use the extend_arrow_color function if you are not sure where the
trajectory will lead to.  If you see the extended trajectory has the
intersection
with the target object.
Please note the detection functions may not always be correct, so you
need to exercise the basic judgment.
Please note the extended trajectory will be yellow line.

**Import Statement**
The jupyter notebook has already executed the following code:

```
from PIL import Image
from IPython.display import display
from tools import *
```

1.  The generated actions can resolve the given user request perfectly.
The user request is reasonable and can be solved.
2.  The arguments of a tool must be the same number, modality, and format
specified in TOOL LIST.
3.  If you think you got the answer, use ANSWER: <your answer> to provide
the answer, and ends with TERMINATE.
4.  All images in the initial user request are stored in PIL Image objects
named image_1, image_2, ..., image_n.  You can use these images in your
code blocks.
5.  Here I tell you the secrets in solving the trajectory reasoning task,
firstly you need to identify the correct object based on the description,
then you should use your visual reasoning skills to find out what is the
correct trajectory.

---

**Trajectory Reasoning Prompt Activation**

Here I provide some examples for you to better undetstand how to use
functions to do trajectory raesoning tasks.

**Example: Hand Trajectory Reasoning**
USER REQUEST: Assuming the hand can only move in a straight line, which
direction is the hand most likely to pick up the silver bowl in the lower
left
part of the picture?
THOUGHT 0:  Firstly, I need to detect the silver bowl in the image.

ACTION 0:

image_1_bowl_detection, image_1_bowl_boxes
= detection(image_1, ["silver bowl"])

display(image_1_bowl_detection.annotated_image)

THOUGHT 1:  Now I have the detected bounding box of the silver bowl, I need
to reason about what color of trajectory will lead to reach that bowl.  From
my
observation, the red trajectory maybe able to reach the bowl, but I am not
sure
about that, so I will call the extend_arrow_color tool.

ACTION 1:

image_1_extended_red = extend_arrow_color(image_1, color="red")
display(image_1_extended_red)

THOUGHT 2:  After extending the red trajectory, I can see that the extended
red trajectory can reach the silver bowl.  So the correct answer should be
the
red trajectory.  Therefore, the correct option is C.

ACTION 2:  No action needed.

ANSWER: The correct option is (c) red trajectory.  TERMINATE

If after some observations and function calls you can not find the answer,
use your common sense to reason about the answer.

**Semantic Scene Graph Construction.** A semantic scene graph is a structured graph-based representation of a visual or embodied environment, where nodes correspond to semantically meaningful entities such as objects, agents, or regions, and edges encode their spatial, functional, or interaction-based relationships. We save our constructed semantic scene graph in a 'json' file and the model can choose to update the graph during the reasoning process.

**Notebook for Planning Tasks.** We conduct our experiments within a Jupyter Notebook environment. In each task, BEAR-Agent explicitly prompts the MLLMs with observations from each frame of the activity's history video. For subsequent rounds of interaction, the entire dialogue history is provided as input to the MLLMs, offering additional temporal and contextual grounding to support coherent reasoning and planning.

*Table 8.* Three manipulation tasks implemented in Maniskill (Gu et al., 2023)

| Task | Description and Subtasks |
|------|--------------------------|
| **General** | Grasp common household objects
• Pick up the blue mug
• Grasp the red mug
• Pick up scissor
• Grasp the hook |
| **Spatial** | Pick and place objects that require spatial reasoning to identify and infer their relative positions.
• Pick up the top right cube and place it in the plate below
• Grasp the top right left cube and place it in the plate on the left
• Pick up the red cube on the left and place it in the plate below
• Pick up the red cube on the left and place it in the plate on the right |
| **Part** | Functional grasping, grasp the certain part of object for interactive tool use.
• Pick up the handle of the hook
• Pick up the handle of the hammer
• Pick up the handle of spatula
• Pick up the handle of screwdriver |

## K. Implementation of Embodied Tasks

In order to verify if our BEAR-Agent is effective for embodied task execution, we implement three series of representitive manipulation tasks using Maniskill as our simulation (Gu et al., 2023). And we provide the detailed implementations of our embodied tasks here.

**Tasks.** We adopt ManiSkill (Gu et al., 2023) as our testbed, using the Franka Panda robot in a tabletop setup. We implement three series of manipulation tasks, each accompanied by four distinct language instructions, as detailed in Table 8.

**Baseline.** We adopt MOKA (Liu et al., 2024a) as our baseline method. MOKA is a multimodal action planning framework that integrates visual perception, language understanding, and spatial reasoning to generate robot-executable manipulation plans. Built on large vision-language models (VLMs) such as GPT-4V, MOKA operates in a turn-based dialogue manner, decomposing high-level instructions into subtasks (e.g., grasping, placing). The framework first proposes candidate keypoints for object interaction based on segmentation masks, then queries the VLM to select optimal grasp and target locations, along with any intermediate waypoints needed for spatially coherent motion.

More specifically, ***MOKA uses GPT-4V as its backbone to identify keypoints and generate motion trajectories conditioned on those keypoints.*** In the MOKA framework, five semantically grounded keypoints are employed to annotate and interpret human-object interaction trajectories: $grasp$, $function$, $target$, $pre\_contact$, and $post\_contact$, as shown in Figure 95. Each keypoint corresponds to a distinct phase in the manipulation process. The $grasp$ point marks the initial contact location where the hand or tool engages with the object (e.g., grasping the edge of a kettle lid). The $function$ point indicates the functional region of the object involved in the interaction, such as a handle or rotation axis. The $target$ point specifies the intended destination or goal of the action, such as a placement location or alignment position. $Pre\_contact$ and $post\_contact$ represent transitional motion waypoints—immediately before and after physical interaction—used to model the approach and retreat phases of the motion trajectory.

**Implementation of BEAR-Agent.** The BEAR-Agent is implemented based on the MOKA framework by augmenting GPT-4V with additional guidance, tool support, and initialization routines. BEAR-Agent facilitates the interaction process by equipping GPT-4V with essential tools and structured prompts. Through several rounds of dialogue, GPT-4V is able to reason about the task and identify keypoint pixel coordinates based on prior context and visual inputs.

**Experiment result.** As shown in Figure 8 and Table 8, for each language instruction within a task, we perform 20 rollouts and compute the instruction-level average success rate. For each task with four different language instructions, we report the

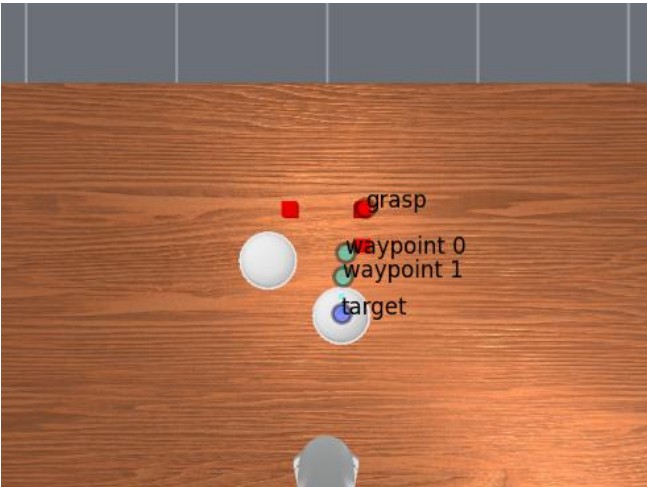

*Figure 95.* **Keypoint generation using MOKA.** As shown in the figure, MOKA generates 'grasp', 'waypoint', and 'target' keypoints for each task, and translates them into motion trajectories.

average of these instruction-level success rates as the overall task success rate.

